# Beyond the Lazy versus Rich Dichotomy: Geometry Insights in Feature Learning from Task-Relevant Manifold Untangling

## Abstract

The ability to integrate task-relevant information into neural representations is a fundamental aspect of both human and machine intelligence. Recent studies have explored the transition of neural networks from the *lazy* training regime (where the trained network is equivalent to a linear model of initial random features) to the *rich* feature learning regime (where the network learns task-relevant features). However, most approaches focus on weight matrices or neural tangent kernels, limiting their relevance for neuroscience due to the lack of representation-based methods to study feature learning. Furthermore, the simple lazy-versus-rich dichotomy overlooks the potential for richer subtypes of feature learning driven by variations in learning algorithms, network architectures, and data properties.

In this work, we present a framework based on representational geometry to study feature learning. The key idea is to use the untangling of task-relevant neural manifolds as a signature of rich learning. We employ manifold capacity—a representation-based measure—to quantify this untangling, along with geometric metrics to uncover structural differences in feature learning. Our contributions are threefold: First, we show both theoretically and empirically that task-relevant manifolds untangle during rich learning, and that manifold capacity quantifies the degree of richness. Second, we use manifold geometric measures to reveal distinct learning stages and strategies driven by network and data properties, demonstrating that feature learning is richer than the lazy-versus-rich dichotomy. Finally, we apply our method to problems in neuroscience and machine learning, providing geometric insights into structural inductive biases and out-of-distribution generalization. Our work introduces a novel perspective for understanding and quantifying feature learning through the lens of representational geometry.

## 1 Introduction

Learning induces changes in brain activity, whether it involves navigating a new city, adapting novel motor skills, or acquiring new cognitive tasks. These changes are reflected in the incorporation of task-relevant features into neural representations (Olshausen & Field, 1996; Poort et al., 2015; Niv, 2019; Reinert et al., 2021; Gurnani & Gajic, 2023). Similarly, the remarkable success of deep learning is often attributed to the ability of neural networks to learn problem-specific features[1]. For example, in deep neural networks (DNNs) (LeCun et al., 1998; Krizhevsky et al., 2012), the ability to learn rich feature hierarchies enables superior image classification performance (Girshick et al., 2014). Meanwhile, the seminal work of (Chizat et al., 2019) demonstrated that neural networks can perform well even when there are negligible changes in the weights of the networks. This observation had led to some research questions. Do neural networks always need to learn the feature relevant to the task? How can we evaluate the quality of the learned features?

To answer these questions, researchers in representation learning have developed several methods to determine whether a neural network operates in the *lazy* regime (learning without changing internal features) or the *rich* regime (learning task-relevant features)[2]. These methods include measuring

---

[1]In this paper, *features* to refer to measurable properties or characteristics of patterns in data/input.
[2]These two regimes are also known as *kernel regime* and *feature learning regime* .

changes in the weights of the network, tracking activated neurons, and assessing differences in the linearized model (also known as the neural tangent kernel, NTK (Jacot et al., 2018)). Factors such as initial weight norm, learning rate, and readout weight have been found to play a role in whether a network is lazy or rich (Chizat et al., 2019). Moreover, recent theoretical evidence has suggested that networks could perform better in the rich regime compared to the lazy regime (Yang & Hu, 2021; Shi et al., 2022; Karp et al., 2021; Damian et al., 2022; Ba et al., 2022).

However, feature learning is much *richer* than the lazy versus rich dichotomy. For example, changes in representations are not always beneficial as they can lead to issues such as catastrophic forgetting (Kirkpatrick et al., 2017). Moreover, different network architectures, training procedures, and objective functions, initializations, can result in different inductive biases for feature learning (Chizat et al., 2019; Bordelon & Pehlevan, 2022; Ba et al., 2022; Damian et al., 2022), yet all of these scenarios could fall under the broad category of rich learning. Lastly, current limitations in neuroscience technology for precisely tracking synaptic weight changes in neural circuits necessitate a framework based on neural representations rather than network weights or neural tangent kernel.

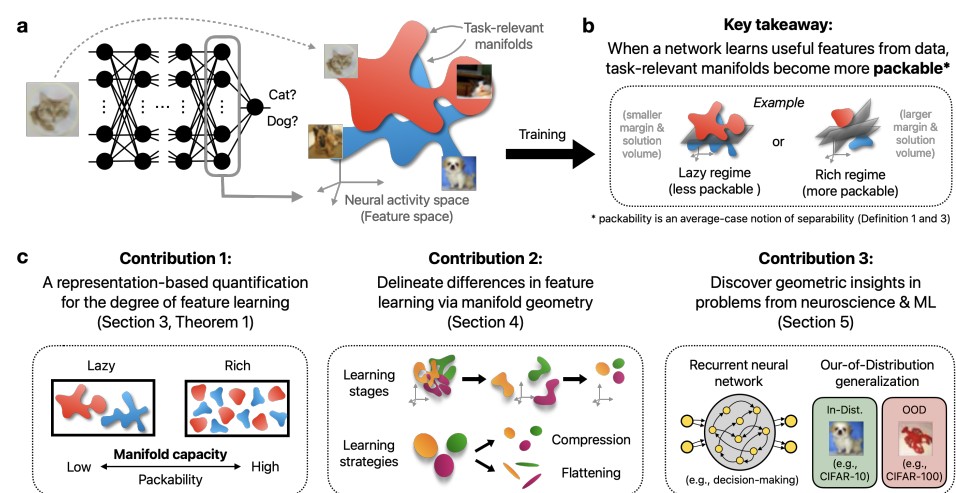

Figure 1: Schematic illustration. **a**, We propose to investigate feature learning using representational geometry and task-relevant manifolds. **b**, Specifically, using the packability of manifolds (Definition 1 and Definition 3) to quantify the degree of richness in feature learning. **c**, Three main contributions of this paper. More details in the corresponding section.

## 1.1 CONTRIBUTIONS

We study feature learning through the angle of *task-relevant manifolds*. Here, task-relevant manifolds refer to point clouds in the neural activity space that are related to the tasks. For example, in a classification task, a manifold could be the point cloud of neural activations corresponding to stimuli in a given category (e.g., the cat and dog manifolds in Figure 1a, left). In other domains, a manifold could correspond to a context in a neuroscience experiment or to a concept in a language model.

In a network that does not learn task-relevant features (e.g., lazy learning, random features, Figure 1b, left), the manifolds are poorly organized, making them harder to distinguish (e.g., smaller margin, smaller solution volume). In contrast, when a network learns task-relevant features (e.g., rich learning, neural collapse Figure 1b, right), the manifolds become well-organized and easier to separate (e.g., larger margin, larger solution volume). From this perspective, feature learning can be viewed as a process of *untangling* task-relevant manifolds—structuring the neural representational space to improve separation among manifolds.

To make this intuition concrete and quantitative, we propose the usage of *manifold capacity* (Chung et al., 2018; Chou et al., 2024) to quantify the degree of richness in feature learning (Figure 1c, left). Spcifically, manifold capacity (Definition 1 and Definition 3) quantifies the degree of manifold

untangling via an average-case notion of how separable the manifolds are: manifold packability[3]. Additionally, manifold capacity is analytically connected to a collection of geometric measures, which provide mechanistic descriptors to explain how these manifolds untangle.

To demonstrate our proposed method, we examine problems in neuroscience and machine learning and find insights that have not been reported. Our contributions and results are summarized below.

- Manifold capacity as a representation-based method to quantify feature learning (Section 3).
    - We theoretically and empirically show that manifold capacity tracks the degree of feature learning in a wide range of settings.
    - We demonstrate that capacity is better than conventional measures in quantifying the degree of feature learning.
- Manifold geometry reveals previously unreported subtypes of feature learning (Section 4).
    - We find that the training process of neural networks undergoes various *learning stages* as shown by the dynamics of different geometric measures.
    - We discover emergent learning strategies as networks transition between lazy and rich regimes. These strategies involve trade-offs among different geometric measures.
- New geometric insights in problems from neuroscience and machine learning (Section 5).
    - In recurrent neural networks (RNNs) trained on common neuroscience tasks, we find that different network structures can result in different manifold geometry, even when achieving the same accuracy and feature learning level.
    - In an out-of-distribution (OOD) generalization task on deep neural networks (DNNs), we find manifold-geometric correlates to the failure of generalization. This finding opens up new avenues for future research in mitigating issues in OOD generalization.

## 1.2 RELATED WORK

Feature learning has been a fundamental research problem in various domains, including neuroscience and machine learning. In neuroscience, understanding the relationship between neural representations and task performance is a central focus (Gao & Ganguli, 2015). Representational geometry (Chung & Abbott, 2021) has emerged as a promising approach to investigate how different organizations of features can lead to better task performance (Bernardi et al., 2020; Flesch et al., 2022; Gurnani & Gajic, 2023). There were also works that attempted to infer the underlying learning rules of a neural network using representational geometry (Cao et al., 2020; Sorscher et al., 2022) and low-order statistics (Nayebi et al., 2020). In machine learning, *visualization techniques* (Zeiler & Fergus, 2014) have been widely used to gain intuitive insights into learned representations, often supplemented with specialized measures to quantify specific properties. On the theoretical front, the *kernel method* (Jacot et al., 2018; Lee et al., 2019) has been a leading approach to analytically characterize the behavior of neural networks, particularly in terms of their deviation from the corresponding kernel. This line of research includes studies on the distinction between lazy and rich regimes (Chizat et al., 2019; Geiger et al., 2020) and identifying problem settings where neural networks with feature learning outperform kernel methods (Ba et al., 2022; Dandi et al., 2023; Yang & Hu, 2021). For a more comprehensive overview of related work, see Appendix A.

## 2 METHOD AND SETUP

### 2.1 MANIFOLD CAPACITY THEORY

Manifold capacity theory (MCT) (Chung et al., 2018; Chou et al., 2024) was originally developed to study the *untangling hypothesis*[4] of invariant object recognition in vision neuroscience (DiCarlo & Cox, 2007). MCT (Chung et al., 2018; Chou et al., 2024) extends the classic notion of storage

---

[3]We remark that the margin in support vector machine (SVM) theory quantifies the degree of separability in the worst-case setting. Here the manifold capacity theory is average-case in the sense of the random projection in Definition 1 and the random up-lifting in Definition 3.

[4]In computational neuroscience, the "untangling hypothesis" posits that the brain transforms complex, entangled sensory inputs into more linearly separable representations, facilitating efficient object recognition.

capacity of points (Cover, 1965; Gardner & Derrida, 1988; Gardner, 1988) to object manifolds, i.e., the collection of neural representations that are invariant to the same input category (Figure 11, left). Let $P$ be the number of classes and $N$ be the number of neurons (in the layer of interest). The $i$-th class manifold is modeled as the convex set[5] $\mathcal{M}_i = \text{conv}(\{\Phi(x) : x \in \mathcal{X}_i\})$ where $\mathcal{X}_i$ is the collection of inputs in the $i$-th class, $\Phi(x)$ is the representation for $x$, and $\text{conv}(\cdot)$ denotes the convex hull of a set. A *simulated* version of manifold capacity is defined as follows.

**Definition 1** (Simulated manifold capacity (Chung et al., 2018)). *Let $P, N \in \mathbb{N}$ and $\mathcal{M}_i \subseteq \mathbb{R}^N$ be convex sets for each $i \in [P] = \{1, \ldots, P\}$. For each $n \in [N]$, define*

$$p_n := \Pr_{\mathbf{y}, \Pi_n} [\exists \theta \in \mathbb{R}^n : y_i \langle \theta, \mathbf{s} \rangle \geq 0, \forall i \in [P], \mathbf{s} \in \mathcal{M}_i]$$

*where $\mathbf{y}$ is a random dichotomy sampled from $\{\pm 1\}^P$ and $\Pi_n$ is a random projection operator from $\mathbb{R}^N$ to $\mathbb{R}^n$. Suppose $p_N = 1$, the simulated capacity of $\{\mathcal{M}_i\}_{i \in [P]}$ is defined as*

$$\alpha_{sim} := \frac{P}{\min_{n \,:\, p_n \geq 0.5}\{n\}}.$$

Intuitively, the simulated manifold capacity measures the *packability* (Chung et al., 2018) of manifolds by determining the smallest dimensional subspace needed to ensure that the manifolds can be separated. Namely, manifolds that are more packable (i.e., separable when projected to smaller dimensional subspaces) exhibit higher manifold capacity. While Definition 1 provides a quantitative description for *packability*, it is computationally expensive to estimate and is not analytically trackable. In (Chung et al., 2018; Chou et al., 2024), the authors resolved these issues by considering a mean-field version of the manifold capacity (formal definition deferred to Definition 3 for simplicity), denoted as $\alpha_{\text{mf}}$, which is analytically trackable and has the property that $|\alpha_{\text{sim}} - \alpha_{\text{mf}}| = O(1/N)$. In particular, (Chou et al., 2024) derived that

$$\alpha_{\text{mf}}^{-1} = \frac{1}{P} \mathop{\mathbb{E}}_{\substack{\mathbf{y} \sim \{\pm 1\}^P \\ T \sim \mathcal{N}(0, I_N)}} \left[ \max_{\mathbf{s}_i \in \mathcal{M}_i} \left\{ \|\text{proj}_{\text{cone}(\{y_i \mathbf{s}_i\})} T\|_2^2 \right\} \right] \tag{1}$$

where $\mathcal{N}(\mu, \Sigma)$ denotes the multivariate Gaussian distribution with mean $\mu$ and covariance $\Sigma$ and $\text{cone}(\cdot)$ is the convex cone spanned by the vectors, i.e., $\text{cone}(\{y_i \mathbf{s}_i\}) = \{\sum_i \lambda_i y_i \mathbf{s}_i : \lambda_i \geq 0\}$.

## 2.2 EFFECTIVE GEOMETRIC MEASURES FROM MANIFOLD CAPACITY THEORY

The advantages of mean-field manifold capacity are: (i) $\alpha_{\text{mf}}$ can be estimated via solving a quadratic program (Algorithm 1) and (ii) Equation 1 connects manifold capacity to the structure of the manifolds $\{\mathcal{M}_i\}$. Specifically, for each $\mathbf{y}, T$, define $\{\mathbf{s}_i(\mathbf{y}, T)\} = y_i \cdot \arg\max_{\{\mathbf{s}_i\}} \|\text{proj}_{\text{cone}(\{y_i \mathbf{s}_i\})} T\|_2^2$ as the *anchor points* with respect to $\mathbf{y}$ and $T$. Intuitively, these anchor points are the support vectors with respect to some random projection and dichotomy as in Definition 1. Specifically, these anchor points are analytically linked to manifold capacity via Equation 1 and are distributed over the manifolds $\{\mathcal{M}_i\}$. This connection inspired the previous work (Chung et al., 2018; Chou et al., 2024) to define the following effective manifold geometric measures that capture the structure of manifolds while being analytically connected to capacity (see Figure 2c and Appendix B).

For each $i \in [P]$, define $\mathbf{s}_i^0 := \mathbb{E}_{\mathbf{y}, T}[\mathbf{s}_i(\mathbf{y}, T)]$ as the **center** of the $i$-th manifold and define $\mathbf{s}_i^1(\mathbf{y}, T) := \mathbf{s}_i(\mathbf{y}, T) - \mathbf{s}_i^0$ to be the **axis** part of $\mathbf{s}_i(\mathbf{y}, T)$ for each pair of $(\mathbf{y}, T)$.

- **Manifold dimension** captures the degree of freedom of the noises/variations within the manifolds. Formally, it is defined as $D_{\text{mf}} := \mathbb{E}_{\mathbf{y}, T}[\|\text{proj}_{\text{cone}(\{\mathbf{s}_i^1(\mathbf{y}, T)\}_i)} T\|_2^2]$.

- **Manifold radius** captures the noise-to-signal ratio of the manifolds. Formally, it is defiend as

$$R_{\text{mf}} := \sqrt{\mathbb{E}_{\mathbf{y}, T} \left[ \frac{\|\text{proj}_{\text{cone}(\{\mathbf{s}_i(\mathbf{y}, T)\}_i)} T\|^2}{\|\text{proj}_{\text{cone}(\{\mathbf{s}_i^1(\mathbf{y}, T)\}_i)} T\|^2 - \|\text{proj}_{\text{cone}(\{\mathbf{s}_i(\mathbf{y}, T)\}_i)} T\|^2} \right]}.$$

- **Center alignment** captures the correlation between the center of different manifolds. Formally, it is defined as $\rho_{\text{mf}}^c := \frac{1}{P(P-1)} \sum_{i \neq j} |\langle \mathbf{s}_i^0, \mathbf{s}_j^0 \rangle|$.

---

[5]In the context of linear classification, it is mathematically equivalent to study the convex hull of a manifold.

- **Axis alignment** captures the correlation between the axis of different manifolds. Formally, it is defined as $\rho_{\mathsf{mf}}^a := \frac{1}{P(P-1)} \sum_{i \neq j} \mathbb{E}_{\mathbf{y},T}[|\langle \mathbf{s}_i^1(\mathbf{y},T), \mathbf{s}_j^1(\mathbf{y},T) \rangle|]$.
- **Center-axis alignment** captures the correlation between the center and axis of different manifolds. Formally, it is defined as $\psi_{\mathsf{mf}} := \frac{1}{P(P-1)} \sum_{i \neq j} \mathbb{E}_{T}[|\langle \mathbf{s}_i, \mathbf{s}_j^1(\mathbf{y},T) \rangle|]$.

Previous work Chung et al. (2018); Chou et al. (2024) showed that the the changes in manifold capacity can be explained by the changes of these geometric measures. For example, the decrease of manifold radius and dimension makes the capacity higher (see Figure 2c, Section B.4).

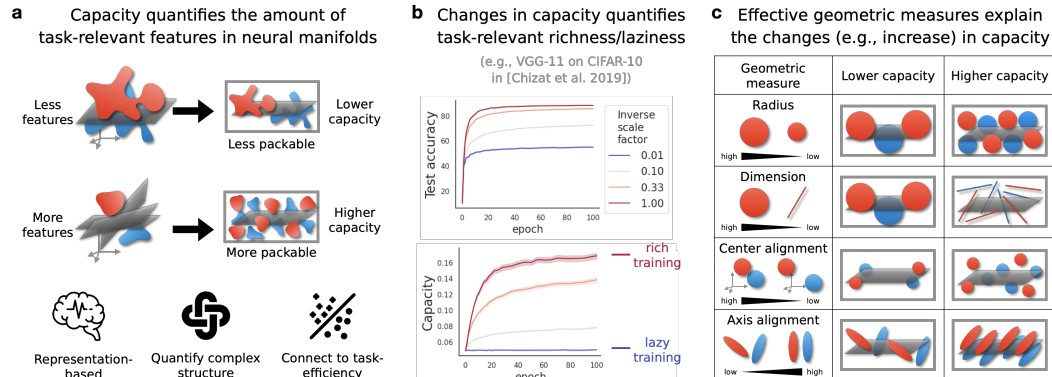

Figure 2: Our methods. **a**, Higher capacity means that the neural representational space can pack more manifolds (Definition 1). **b**, We propose to use changes of capacity across training to study task-relevant richness/laziness in feature learning (Section 3). Top: we consider the setting in (Chizat et al., 2019) where VGG-11 was trained on CIFAR-10. An inverse scale factor was introduced to interpolate between lazy and rich training, where smaller value (blue) corresponds to lazier learning and larger value correponds to richer learning (red). Bottom: we show that the changes in capacity faithfully tracks the degree of richness in feature learning. **c**, Effective geometric measures drive the capacity value, providing mechanistic descriptors to study representational changes in feature learning. Center-axis alignment has a more complex relationship with capacity, discussed Section B.4.

## 2.3 RICH AND LAZY LEARNING IN NEURAL NETWORKS

We studied rich versus lazy learning in two standard settings: 2-layer non-linear neural networks on synthetic data and feedforward deep neural networks on real image classification datasets (Chizat et al., 2019). All analyses were performed on the test data representations in the last layer.

**A scale factor for interpolating between rich and lazy regime.** In all experiments, we use the *inverse scale factor* $\bar{\eta}$ as a tunable ground truth for the degree of feature learning. In particular, $\bar{\eta}$ controls the magnitude of the output of the network as in (Chizat et al., 2019). Intuitively, a larger $\bar{\eta}$ indicates that the learning rate of intermediate layers is faster compared to that of the readout weights, resulting in a richer learning process. See Appendix D and E for more details.

**2-Layer non-linear neural networks.** We considered standard 2-layer neural networks with non-linear activation functions and trained with gradient descent. We also considered a data model to generate random point clouds as input manifolds. This setting serves as a well-curated testbed for testing the proposed methodology and showcasing intuitions. See Appendix D for more details.

**Deep neural networks.** The goal of this work is to develop a framework to understand neural representations rather than pushing the benchmark. Therefore, we focused on models and settings that are large enough to see interesting phenomena, while the computational cost is still reasonable. Specifically, we considered feedforward DNN architectures such as VGG-11 (Simonyan & Zisserman, 2015) and ResNet-18 (He et al., 2016) and datasets CIFAR-10 (Krizhevsky & Hinton, 2009), CIFAR-100 (Krizhevsky & Hinton, 2009), CIFAR-10C (Hendrycks & Dietterich, 2018). This setting illustrates the applicability of our methodology to DNNs. See Appendix E for more details.

# 3 MANIFOLD CAPACITY QUANTIFIES THE DEGREE OF FEATURE LEARNING

In this section, we provide both empirical and theoretical justifications for using the increase in capacity during training as a measure to quantify the degree of richness (or the amount of task-relevant features) in feature learning. Furthermore, we compare our method with conventional approaches in the study of lazy versus rich learning, highlighting the new insights uncovered by our approach.

## 3.1 JUSTIFICATIONS OF CAPACITY FOR QUANTIFYING THE LAZY VERSUS RICH DICHOTOMY

**Theoretical justification on 2-layer non-linear neural networks.** We built on previous work from Ba et al. (2022) and Montanari et al. (2019) to analytically characterize the connection between capacity, prediction error, and the effective degree of richness in a well-studied theoretical model. Concretely, we consider the training of a fully-connected 2-layer network of the form $f(\mathbf{x}) = \frac{1}{\sqrt{N}}\mathbf{a}^\top\sigma(W^\top\mathbf{x})$, where $\mathbf{x} \in \mathbb{R}^d$ is an input, $W \in \mathbb{R}^{N\times d}$ is the hidden layer matrix, $\mathbf{a} \in \mathbb{R}^N$ is the readout weight, and $\sigma : \mathbb{R} \to \mathbb{R}$ is the (non-linear) activation function. To study feature learning in this setting, it is common to consider $W$ to be randomly initialized (i.e., random feature model (Rahimi & Recht, 2007)) and update via gradient descent with squared loss. Meanwhile, the readout weight $\mathbf{a}$ is randomly initialized and fixed to avoid lazy learning (where the network minimally adjusts the hidden layer and focuses on learning a good readout weight) as well as enable mathematical analysis (Ba et al., 2022). Input data and label $(\mathbf{x}_1, y_1), \ldots, (\mathbf{x}_{P_{\text{train}}}, y_{P_{\text{train}}})$ were randomly generated by a teacher-student setting, where there is a hidden signal direction $\beta^*$ that correlates with the label (see Setting 1 for the full setting). As previously proved in Ba et al. (2022) (see Proposition 1), in the proportional asymptotic limit (i.e., $P_{\text{train}}, d, N \to \infty$ at the same rate), the first-step gradient update can be approximated by a rank-1 matrix that contains label information, resulting in the updated weight to be more aligned with the hidden signal $\beta^*$. Hence, in this setting, the learning rate $\eta$ can be used as the ground-truth to measure the amount of task-relevant information (i.e., richness in learning) in the model representation after gradient updates.

We extend the previous results in (Ba et al., 2022) from a regression setting to a classification setting. Specifically, We prove that capacity correctly tracks the effective degree of richness after one gradient step[6]. Moreover, we derive a monotone connection between capacity and prediction accuracy. Here, we provide an informal statement of our results and leave the formal version and proof in Appendix C.

**Theorem 1.** *Given Assumption 1 and Setting 1. Let $0 < \eta < \infty$ be the learning rate of a one-step gradient descent with squared loss and $\psi_1 = \frac{N}{d}, \psi_2 = \frac{P_{\text{train}}}{d}$ where $P_{\text{train}}$ is the number of training points, $d$ is the input dimension, and $N$ is the number of hidden neurons. Let $\alpha_{P_{\text{train}},d,N}(\eta)$ be the capacity and let $\mathsf{Acc}_{P_{\text{train}},d,N}(\eta)$ be the prediction accuracy after a gradient step with learning rate $\eta$. We have*

1. *(Capacity tracks the degree of richness) $\alpha_{P_{\text{train}},d,N}(\eta) \xrightarrow{P_{\text{train}},d,N\to\infty} \alpha(\eta, \psi_1, \psi_2)$ where $\alpha(\cdot, \cdot, \cdot)$ is defined in Theorem 2. Specifically, $\alpha(\eta, \psi_1, \psi_2) < \alpha(\eta', \psi_1, \psi_2)$ for every $0 < \eta < \eta'$.*

2. *(Capacity links to prediction accuracy) $\mathsf{Acc}_{P_{\text{train}},d,N}(\eta) \xrightarrow{P_{\text{train}},d,N\to\infty} \mathsf{Acc}(\eta, \psi_1, \psi_2)$ where $\mathsf{Acc}(\eta, \psi_1, \psi_2)$ is formally defined in Theorem 2. In particular, there exists an increasing and invertible function $h_{\psi_1,\psi_2} : \mathbb{R}_+ \to [0, 1]$ such that $\mathsf{Acc}(\eta, \psi_1, \psi_2) = h_{\psi_1,\psi_2}(\alpha(\eta, \psi_1, \psi_2))$.*

The above theorem justifies the usage of capacity as a measure for the degree of richness in feature learning within a well-studied theoretical setting. We remark that our proof requires substantial technical improvements from (Ba et al., 2022) due to the difference between regression and classification (e.g., analyzing the margin of the Gaussian equivalent model after one-step gradient using tools from (Montanari et al., 2019), Proposition 2).

**Empirical justification in standard settings.** Next, we empirically justify the use of capacity to quantify the degree of feature learning. A classic result in the literature of lazy versus rich training is to train a lazy network where the test accuracy improves, but the weight matrices (or kernels) do not change much before and after training. We consider two settings in (Chizat et al., 2019), one

---

[6]Here we follow the convention in (Ba et al., 2022) and study only the first gradient step as the key Gaussian equivalence step might not hold for more steps as remarked in footnote 2 of (Ba et al., 2022).

is feedforward DNNs (VGG-11 and ResNet-18) trained on CIFAR-10 (Figure 2b), and the other is 2-layer non-linear NNs trained on random point clouds (Figure 3a). In both cases, we observe that the manifolds are more untangled when training is richer and capacity correctly tracks the degree of feature learning (the ground truth being the scale parameter $\bar{\eta}$). This provides empirical justification for the use of capacity as well as evidence for manifold untangling in the rich learning regime.

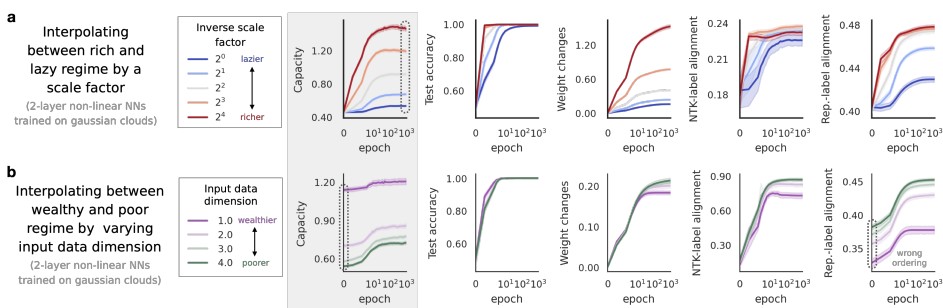

Figure 3: Capacity as a measure for the degree of feature learning. See Section D.1 for the experimental setup. **a**, We interpolated between lazy and rich regime in 2-layer NNs trained to classify Gaussian clouds. We found that capacity could tell the difference between the underlying scale parameter better than the other conventional methods. **b**, We fixed a scale parameter and initialized the input Gaussian clouds with different dimensions (the higher the poorer the initial representations are for each class). We found that capacity could tell the difference in the amount of tasks-relevant features at initialization than other conventional methods. Specifically, the representation-label alignment would characterize the wrong ordering of wealthiness in initial features.

## 3.2 COMPARISON WITH CONVENTIONAL FEATURE LEARNING MEASURES.

Here we compare the capacity with several common measures for feature learning: accuracy curves, weight changes, and alignment methods. Concretely, weight changes at the $t$-th epoch is defined as $\|W_t - W_0\|_F / \|W_0\|_F$ where $W_t$ is the weight matrix at the $t$-th epoch. NTK-label alignment and representation-label alignment at the $t$-th epoch are defined as $\mathrm{CKA}(K_t^{\mathrm{NTK}}, \mathbf{yy}^\top)$ and $\mathrm{CKA}(X_t X_t^\top, \mathbf{yy}^\top)$ respectively, where $\mathbf{y}$ is the label vector, $\mathrm{CKA}(\cdot, \cdot)$ is the center kernel alignment measure (Kornblith et al., 2019), $K_t^{\mathrm{NTK}}$ is the neural tangent kernel and $X_t$ is the representational matrix at the $t$-th epoch. See Appendix A for a detailed introduction to these methods and Appendix D for more experimental details.

**Capacity can detect task-relevant features in the presence of complex structures in data.** In Figure 3a, we consider 2-layer NNs trained on random Gaussian clouds with gradient descent. We vary the scale parameter of the network to interpolate between lazy and rich regimes as done in (Chizat et al., 2019). We find that capacity is better at telling the difference of effective richness (i.e., the scale parameter) of the training than other conventional measures (Figure 3a). In particular, when the training is richer, we expect the representations to exhibit more complex structures.

**Capacity can quantify the differences in task-relevant features at initialization.** When comparing two networks with different initializations, focusing solely on network changes can overlook differences in features present at initialization. Here, we use the capacity value at initialization to determine whether a network is in a wealthy regime (i.e., possessing more task-relevant features) or a poor regime (i.e., possessing less task-relevant features), as shown in (Figure 3b). The wealthy versus poor distinction provides insight into the network's initial state, allowing for a more comprehensive comparison of different settings (see 'Section 5.1 for an example).

## 4 MANIFOLD GEOMETRY REVEALS SUBTYPES OF FEATURE LEARNING

In this section, we demonstrate that feature learning is much richer than the lazy versus rich dichotomy. In particular, we use manifold geometric measures (Figure 2c, and Appendix B for details) to delineate the differences in the learned features (learning strategies) of neural networks and representational changes throughout training (learning stages). The key takeaway from this section is the ability of our method to reveal task-relevant changes in neural representations.

### 4.1 GEOMETRIC DIFFERENCES IN LEARNED FEATURES: LEARNING STRATEGIES

To increase capacity, a network can shrink the radius or compress the dimension of neural manifolds (Figure 2c). We demonstrate in 2-layer NNs the emergence of distinct learning strategies driven by different factors. In Figure 4a, we consider the setting in Figure 3a where we interpolate the degree of richness in feature learning via an inverse scale factor. As training moves from the lazy to a richer regime (blue to gray), the network compresses both the radius and dimension to increase capacity. Interestingly, in an even richer regime (gray to red), the network sacrifices radius to further reduce dimension. In Figure 4b, we consider the setting in Figure 3b where we interpolate the wealth of initialization by varying input data dimension. For the wealthiest initialization (purple), the network primarily compresses radius. For poorer initialization (green), both radius and dimension are compressed in lazier training, while in the richer regime (e.g., inverse scale factor $2^4$), the network sacrifices radius for further dimension compression. In summary, varying degrees of richness in feature learning can exhibit different learning mechanisms, as captured by manifold geometry.

### 4.2 MANIFOLD GEOMETRY CHANGES THROUGHOUT TRAINING: LEARNING STAGES

Neural networks learn in a highly non-monotonic manner throughout the training period. Examples include double descent (Belkin et al., 2019; Nakkiran et al., 2021; Mei & Montanari, 2022) and grokking (Power et al., 2022; Liu et al., 2022; Nanda et al., 2023; Kumar et al., 2024). Previous works have analytically or empirically described the different stages/phases such as comprehension, grokking, memorization, and confusion (Liu et al., 2022) through the trajectory of accuracy curves.

From Figure 4a,b we observe distinct stages of manifold geometry evolution during training in 2-layer networks. In the very rich regime, the network initially compresses both radius and dimension, then increases radius to further reduce dimension. In Figure 4c, we examine a standard setting where VGG-11 is trained on CIFAR-10. Despite the rapid saturation of training and test accuracy, at least four stages of geometric changes are evident (see Figure 2c for analytical connections between geometric measures and capacity): a *clustering stage* (initial manifold compression), followed by a *structuring stage* (increasing alignment), a *separating stage* (decreasing alignment to push manifolds apart), and a final *stabilizing stage* (further reducing center alignment).

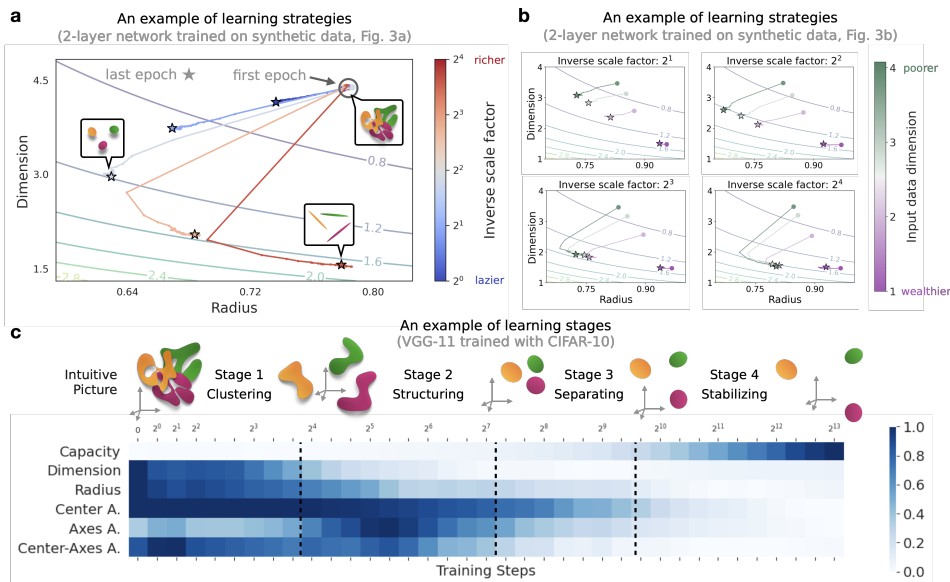

Figure 4: Manifold geometry characterizes learning strategies and learning stages. **a**, Capacity contour plot of the example from Figure 3a. The x-axis is the average manifold radius $R_{\mathsf{mf}}$, the y-axis is the average manifold dimension $D_{\mathsf{mf}}$, and the contour is the geometric approximation of capacity, i.e., $\alpha_{\mathsf{mf}} \approx (1 + R_{\mathsf{mf}}^{-2})/D_{\mathsf{mf}}$ (see Appendix B for details). **b**, Capacity contour plot of the example from Figure 3b. **c**, Normalized manifold geometry dynamics plot of VGG-11 trained with CIFAR-10. The values in each row are rescaled so that the max value is 1 and the min value is 0.

## 5 APPLICATIONS TO NEUROSCIENCE AND MACHINE LEARNING PROBLEMS

In previous sections, we used capacity to quantify the degree of feature learning and delineate the learning stages and strategies through effective geometry. In this section, we apply our framework to find geometric insights in problems from neuroscience and machine learning.

### 5.1 STRUCTURAL INDUCTIVE BIASES IN NEURAL CIRCUITS

We study recurrent neural networks (RNNs) that are trained on standard neuroscience tasks such as perceptual decision making (Britten et al., 1992) (Figure 5a). We adopt the setting from previous work (Liu et al., 2024) on investigating how differences in connectivity initialization affect the learning process. In particular, previous work used the weight changes of RNNs before and after training as a measure to quantify if a network is in rich or lazy training regimes (Figure 5b). Here, we use our methods of capacity and its effective geometry to study such structural biases of neural circuits in a data-driven way (i.e., from neural activity instead of weight matrix).

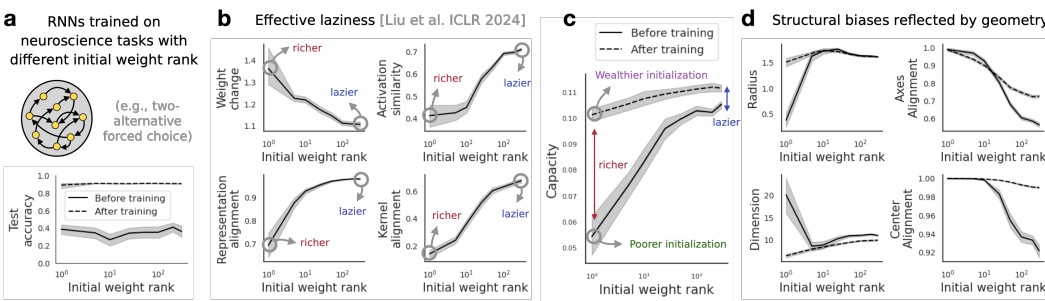

Figure 5: Structural inductive biases in neural circuits. **a**, We consider RNNs trained on standard neuroscience tasks. **b**, Previous work (Liu et al., 2024) found that the initial weight rank of the recurrent connectivity matrix leads to an inductive bias toward effectively richer or lazier training. **c**, We find that RNNs trained with different initial weight rank reach the same capacity value at final epoch. It is the difference in capacity at initialization that makes RNNs with small initial weight rank richer in training. **d**, Despite having the same capacity at final epoch, RNNs with different initial weight rank exhibit different manifold geometry.

**Experimental setup.** We use the `neurogym` package (Molano-Mazon et al., 2022) to simulate common cognitive tasks, including perceptual decision making, delayed matching, etc. To study how connectivity structure impacts learning strategies, we initialize recurrent neural networks (RNN) weights with varying ranks (low-rank weight has lower connectivity and higher initial bias and vice versa) via Singular Value Decomposition (similar setup used in (Liu et al., 2024)). The RNN have 300 hidden units, 1 layer, with ReLU activations, and are trained for 10000 iterations using `SGD` optimizer. (more details can be found in the Appendix section F). Manifold capacity and effective geometric measures are computed using representations from the hidden states.

**Our findings.** First, we study the training dynamics of capacity value in RNNs with various initial weight rank (Figure 5c). In agreement with the previous finding in (Liu et al., 2024) using weight changes, we find that the capacity changes of the small initial weight rank RNNs are higher than those of the large initial weight rank RNNs. Interestingly, the capacity values at the final epoch are about the same for RNNs with different initial weight rank. It is the difference in capacity value at initialization that distinguishes the learning dynamics of RNNs with different initial weight rank. Namely, small initial weight rank RNNs are in the poorer-richer feature learning regime, while large initial weight rank RNNs are in the wealthier-lazier feature learning (Figure 5c).

Next, although the capacity values of RNNs at the final epoch are about the same for different initial weight ranks, we find that their geometric organizations are quite different (Figure 5d). For example, poorer-richer learning (i.e., small initial weight rank) ends up with a larger radius but smaller dimension, while it is the opposite for wealthier-lazier learning (i.e., large initial weight rank). This finding suggests that there are structural biases in RNNs at the manifold geometry level.

## 5.2 OUT-OF-DISTRIBUTION GENERALIZATION

Out-of-distribution (OOD) generalization refers to the scenario when the training distribution $(\mathbf{x}, y) \sim \mathcal{D}_{\text{train}}$ is different from the test distribution $(\mathbf{x}, y) \sim \mathcal{D}_{\text{test}}$. Here we focus on the case where the label set in $\mathcal{D}_{\text{test}}$ is different from that in $\mathcal{D}_{\text{train}}$.

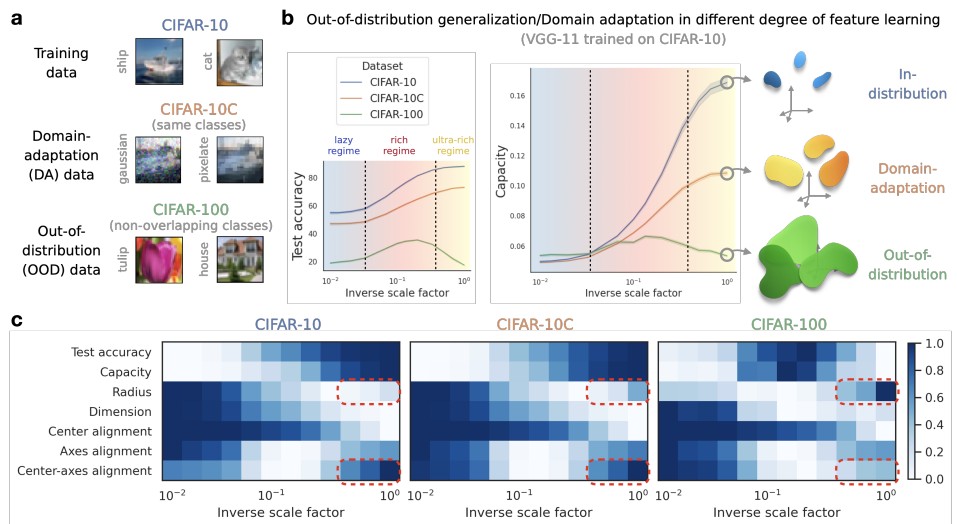

Figure 6: Out of distribution generalization. **a**, CIFAR-10c as a domain adaptation (DA) dataset and CIFAR-100 as an OOD dataset. **b**, Test accuracy improves for CIFAR-10 and CIFAR-10C as the training becomes richer and richer while the linear probe accuracy for CIFAR-100 would drastically drop in the ultra-rich training regime. **c**, Effective manifold geometry of CIFAR-100 reveals that the expansion of manifold radius and the increase of center-axis alignment explain the failure of OOD generalization in the ultra-rich regime. The color is normalized for each row respectively.

**Experimental setup.** For each model pre-trained on CIFAR-10, we train a linear classifier (i.e., linear probe (Alain & Bengio, 2016)) on top of the last-layer representation with CIFAR-100 train set, and then evaluate the linear probe's performance on CIFAR-100 test set (see more details in Appendix E.4). We also consider a corrupted version of CIFAR-10, the CIFAR-10C dataset (Hendrycks & Dietterich, 2018) as an example of domain adaptation (DA) task. Finally, we compute the manifold capacity and effective geometric measures on these last-layer representations.

**Our findings.** We see that the test accuracy of the OOD dataset increases when the network enters the rich learning regime ($\bar{\eta}$ around 0.1) but decreases drastically when the degree of feature learning is too rich ($\bar{\eta}$ around 1.0). The failure in such *ultra-rich* feature learning regime is different from the test accuracy of both CIFAR-10 and CIFAR-10C ( Figure 6b). Looking at the capacity and effective geometry ( Figure 6c), we first see strong correlations between the capacity and test accuracy, which warrants the use of effective geometry. Next, we find that the expansion of manifold radius and the increase of center-axis alignment in the ultra-rich regime explain the drop of capacity. Interestingly, we also see an architectural difference where it is the increment in dimension in the ultra-rich regime explaining the drop of capacity in ResNet-18 (Figure 21). We leave it as a future direction to extend our study, applying these geometric insights to improve OOD generalization performance.

## 6 CONCLUSION AND DISCUSSION

The primary contribution of this work is to demonstrate how the perspective of task-relevant manifold untangling (quantified by manifold capacity and delineated by manifold geometric measures) can enhance our understanding of feature learning at an intermediate level. We propose several promising future directions, including extending the theoretical analysis to more realistic settings, exploring applications in other types of DNN (e.g., recurrent networks, transformers) and addressing relevant scientific inquiries in neuroscience, such as inferring plasticity mechanisms from observed learning dynamics in neural data, and predicting learning-induced changes across brain regions.

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

## A MORE ON RELATED WORK

**Visualization.** Due to the black-box and complex nature of deep neural networks, various visualization techniques have been developed to attempt to characterize the features that models learn during training (*feature visualization*) and identify which input pixel and / or feature activation in the hidden layers contribute significantly to the final model outputs (*feature attribution*). *Feature visualization* techniques visualize features (e.g convolutional filter in the case of CNNs) by generating the input sample that maximizes the activation of that given feature via gradient descent (Olah et al., 2017) (Erhan et al., 2009) (Zeiler & Fergus, 2014). With its vivid visualization, *feature visualization* provide good intuition about the qualitative characteristics of the features that DNNs learn across layers (Zeiler & Fergus, 2014) as well as different types of models (e.g, standard vs adversarially robust (Engstrom et al., 2019)). *Feature attribution* techniques generally identify how much each input and/or hidden features contribute to the final model prediction by computing the gradient of that input/hidden features to the output (some example techniques include saliency map (Simonyan et al., 2013), Grad-cam (Selvaraju et al., 2017), integrated gradient (Sundararajan et al., 2017)). Although both *feature visualization* and *feature attribution* offer intuitive understanding about the model's feature characteristics, the qualitative nature of visualization makes it difficult to quantify the degree of relevance of the learned features to a given task.

**Kernel dynamics.** Kernel methods (Hofmann et al., 2008) have been classic machine learning techniques, where the primary goal is to design an effective embedding that maps inputs to a feature space, thus facilitating efficient algorithms to find good solutions (e.g., linear classifier). While neural networks are inherently complex, seminal works (Jacot et al., 2018; Lee et al., 2019) have shown that in the infinite width limit, a network can be linearized by its *neural tangent kernel (NTK)*. Thus, studying the NTK of a network allows an analytical understanding of various properties of neural networks, such as convergence to global minima (Du et al., 2018; 2019), generalization performance (Allen-Zhu et al., 2019; Arora et al., 2019), implicit bias (Bordelon et al., 2020; Canatar et al., 2021), and neural scaling laws (Bahri et al., 2021).

When a network is properly initialized (Chizat et al., 2019), gradient descent can converge to the NTK of the random initialization, a setting known as the *kernel regime* (a.k.a., *lazy training* or *random feature regime*). On the other hand, a network can also enter what is known as the *feature learning regime* (a.k.a., *rich training* or *mean-field limit*), where it deviates from the NTK of the initialization (Geiger et al., 2020). Extensive research has been conducted to characterize lazy versus rich regimes (Geiger et al., 2020) and to demonstrate instances where feature learning outperforms lazy training (Yang & Hu, 2021; Ba et al., 2022; Dandi et al., 2023). It is important to note that even when a network undergoes feature learning, the NTK can still be defined at each epoch. Previous works also analytically characterized the dynamics of kernel in simpler models (Bordelon et al., 2020). Studying such kernel dynamics also provides a lens for exploring questions related to feature learning, such as grokking (Kumar et al., 2024).

**Representational geometry.** The visualization approaches mentioned above focus on studying the geometric properties of the feature map itself. Another fruitful direction is to examine the geometric properties of the neural representations of inputs (i.e., embedding vectors) and their connections to performance (Chung & Abbott, 2021; Gurnani & Gajic, 2023). Various dimensionality reduction methods (e.g., principal components analysis (PCA), Isomap, t-SNE, MDS, and UMAP) have been proposed to build intuitions about the organization of high-dimensional feature spaces. In addition, there are approaches that study lower-order statistics of embedding vectors, such as representational similarity (Kriegeskorte & Kievit, 2013) and spectral methods (Rahaman et al., 2019; Bahri et al., 2021; Ghosh et al., 2022). Methods for extracting higher-level geometric properties (e.g., dimension) have also been proposed (Chung et al., 2018; Cohen et al., 2020; Chou et al., 2024; Ansuini et al., 2019), with wide applications in both machine learning (e.g., memorization (Stephenson et al., 2021), grokking of modular arithmetic (Liu et al., 2022; Nanda et al., 2023)) and neuroscience (e.g., perceptual untangling in object categorization (Chung et al., 2018), abstraction (Bernardi et al., 2020), few-shot learning (Sorscher et al., 2022), social learning (Paraouty et al., 2023)).

---

[7]See Figure 3 for examples of how NTK-label alignment and representation-label alignment could fail at quantifying the amount task-relevant features.

| | Our approach (manifold geometry) | Accuracy | Weight changes | NTK-label alignment | Representation-label alignment |
|---|:---:|:---:|:---:|:---:|:---:|
| Detect the changes in features | ✔ | ✗ | ✔ | ✔ | ✔ |
| Quantify the amount of task-relevant features | ✔ | ✗ | ✗ | ✗[7] | ✗[7] |
| Representation-based | ✔ | ✗ | ✗ | ✗ | ✔ |
| Delineate subtypes of feature learning | ✔ | ✗ | ✗ | ✗ | ✗ |

Table 1: Comparison with conventional measures used in lazy versus rich learning.

## A.1 PREVIOUS WORK ON STORAGE CAPACITY

Storage capacity is defined as the information load for linear readouts and has been studied in several communities, including learning theory (Cover, 1965) and statistical physics of neural networks (Gardner & Derrida, 1988; Gardner, 1988). To enable a mathematical treatment, we focus on the proportional limit (a.k.a. the high-dimensional limit, the thermodynamic limit), i.e., $N, P \to \infty$ and $\lim_{N,P\to\infty} N/P = O(1)$. For a given network and input data, we denote the representation of the $i$-th input $x_i$ as $\Phi(x_i) \in \mathbb{R}^N$ where $\Phi$ is the (non-linear) feature map. The storage capacity of $\Phi$ is defined as.

$$\alpha(\Phi) := \lim_{N\to\infty} \max_{P} \left\{ \frac{P}{N} \; : \; \Pr_{\mathbf{y}} \left[ \exists \theta \in \mathbb{R}^N, \; \forall i \in [P], \; y_i \langle \theta, \Phi(x_i) \rangle \geq 0 \right] \geq 1 - o_N(1) \right\} \quad (2)$$

where $\mathbf{y} \in \{\pm 1\}^P$ is uniformly random sampled, $\theta$ is the linear classifier, and $o_N(1)$ denotes vanishing terms (i.e., $o_N(1) \to 0$ as $N \to \infty$). One can also consider the setting where the distribution of $\mathbf{y}$ is biased toward some task direction (Montanari et al., 2019). Intuitively, $\alpha(\Phi)$ quantifies the number of patterns per neuron that a network can store and decode with linear readouts.

Recall that storage capacity is defined as the critical ratio between the number of stored patterns and the number of neurons (Equation 2). Cover's theorem (Cover, 1965) shows that the success probability of having a linear classifier for $P$ points with random binary labels in general position [8] is $p(N, P) = 2^{1-P} \sum_{k=0}^{N-1} \binom{P-1}{k}$. In particular, for $P/N < 2$ we have $\lim_{N\to\infty} p(N, P) = 0$ and for $P/N > 2$ we have $\lim_{N\to\infty} p(N, P) = 1$. Namely, the storage capacity of points in general position with random binary label is 2. See also Figure 7 for finite-size and numerical examples.

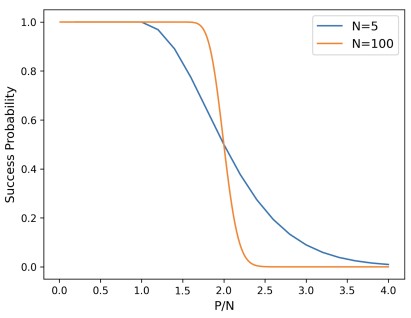 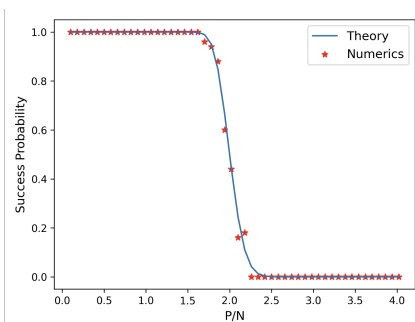

Figure 7: Storage capacity of random points and labels. Storage capacity is defined as the critical ration $P/N = 2$ where the success probability undergoes a phase transition. Left: finite size success probability curves proved in Cover's theorem. Right: a numerical check for Cover's theorem.

In the seminal works of Gardner and Derrida (Gardner & Derrida, 1988; Gardner, 1988), the storage capacity for random points with non-zero margin is analytically characterized using replica method. In the context of associative memory, the storage capacity of Hopfield networks (Hopfield, 1982) is calculated by (Amit et al., 1987).

---

[8]Meaning that every $N' \leq N$ points are linearly independent. Note that random points are in general position with probability $1 - o(1)$.

# B    MANIFOLD CAPACITY THEORY AND EFFECTIVE GEOMETRY

Manifold capacity theory (MCT)(Chung et al., 2018; Chung & Abbott, 2021; Wakhloo et al., 2023; Chou et al., 2024) was originally developed for the study of manifold untangling (DiCarlo & Cox, 2007) in theoretical/computational neuroscience. Intuitively, manifold untangling refers to the increased separation of high-dimensional manifolds (e.g., point cloud manifolds) in the eyes of a downstream readout. MCT quantifies this intuition via modeling a downstream neuron as a linear classifier, and uses the *packing efficiency* of the neural representational space to evaluate the degree of manifold untangling. Mathematically, such packing efficiency coincides with support vector machine (SVM) in an average-case setting.

## B.1   NEURAL MANIFOLDS AS CONVEX HULLS OF PRE-READOUT REPRESENTATIONS

As we are studying feature learning, we are interested in the neural representations that correspond to activations obtained from the pre-linear readout layer neurons. The readers can refer to Appendix D and Appendix E for details on activation extraction. Notation wise, let $N$ be the number of neurons. Therefore, all neural representations live in $\mathbb{R}^N$ space.

Next, we group neural representations by their category labels assigned during training to obtain $P$ data manifolds. For $i \in \{1, \ldots, P\}$, the $i$-th data manifold, denoted as $\mathcal{M}_i$, is a convex set in $\mathbb{R}^N$. To ensure convexity in practice, we take $M_i$ to be the convex hull of a collection of vectors $\mathcal{M}_i = \{\mathbf{x}_1^i, \ldots, \mathbf{x}_{M_i}^i\}$ where $M_i$ is the number of points in the $i$-th manifold.

Notice that the each data manifold lives in its own subspace of dimension $D_i \leq N$. Therefore, we can rewrite each data manifold in its own coordinate system:

$$\mathcal{M}_i = \left\{ \mathbf{u}_0^i + \sum_{j=1}^{D_i} s_j \mathbf{u}_j^i \ \middle| \ \mathbf{s} = (s_1, \ldots, s_{D_i}) \in \mathcal{S}_i \right\} \tag{3}$$

Here, $\mathbf{u}_0^i$ is the center of the $i$-th manifold and $\{\mathbf{u}_j^i\}_{j=1}^{D_i}$ is an orthonormal basis. The shape set $\mathcal{S}_i \subset \mathbb{R}^{D_i}$ is a convex set denoting coordinates of the manifold points in its subspace. In practice, the manifold axes and shape sets $\mathcal{S}_i$ are completely data driven.

## B.2   A SIMULATION DEFINITION FOR MANIFOLD CAPACITY

Recall from Section 2 that the simulation version of manifold capacity is defined as follows.

**Definition 1** (Simulated manifold capacity (Chung et al., 2018)). *Let $P, N \in \mathbb{N}$ and $\mathcal{M}_i \subseteq \mathbb{R}^N$ be convex sets for each $i \in [P] = \{1, \ldots, P\}$. For each $n \in [N]$, define*

$$p_n := \Pr_{\mathbf{y}, \Pi_n} \left[ \exists \theta \in \mathbb{R}^n \ : \ y_i \langle \theta, \mathbf{s} \rangle \geq 0, \ \forall i \in [P], \ \mathbf{s} \in \mathcal{M}_i \right]$$

*where $\mathbf{y}$ is a random dichotomy sampled from $\{\pm 1\}^P$ and $\Pi_n$ is a random projection operator from $\mathbb{R}^N$ to $\mathbb{R}^n$. Suppose $p_N = 1$, the simulated capacity of $\{\mathcal{M}_i\}_{i \in [P]}$ is defined as*

$$\alpha_{sim} := \frac{P}{\min_{n \, : \, p_n \geq 0.5}\{n\}} .$$

Intuitively, the simulated manifold capacity measures the *packability* (Chung et al., 2018) of manifolds by determining the smallest dimensional subspace needed to ensure they can be separated. Namely, manifolds that are more packable[9] (i.e., separable when projected to smaller dimensional subspaces) exhibit higher manifold capacity. Note that the simulated capacity can be estimated from data by empirically estimate $p_n$ and perform binary search to find the critical dimension $\min_{p_n \geq 0.5}\{n\}$. This procedure is computationally expensive and requires some choices of hyperparameters (which makes the definition a little ad hoc). Nevertheless, Definition 1 provides good intuition on how to think about manifold capacity (and its connection to packing).

---

[9]The reason why this is called "packing" is that projecting manifolds into smaller dimensional subspace is like packing them into a smaller neural representational space.

### B.3 A MEAN-FIELD DEFINITION FOR MANIFOLD CAPACITY

To overcome the above-mentioned drawbacks of simulated manifold capacity, previous work (Chung et al., 2018; Wakhloo et al., 2023; Chou et al., 2024) defined some *mean-field models* to enable a nice mathematical definition of manifold capacity while still being a good approximation to the simulated manifold capacity.

**Mean-field model from (Chou et al., 2024).** Given a collection of (finite) data manifolds $\{\mathcal{M}_i\}_{\mu=1}^P$. A mean-field model is to generate infinitely many ($P_{\mathsf{mf}}$) manifolds in an infinite-dimensional ($N_{\mathsf{mf}}$) space and characterizing the largest possible $P_{\mathsf{mf}}/N_{\mathsf{mf}}$ such that these "mean-field" manifolds are separable. The key idea is that if this generating process nicely preserve the structure in the data manifolds, then the packing property of these mean-field manifolds will be very similar

**Definition 2** (Mean-field model from (Chou et al., 2024)). *Let $\{\mathcal{M}_i\}_{i\in[P]}$ be a collection of data manifolds in $\mathbb{R}^N$ as defined in Equation 3. Let $\alpha \in \mathbb{R}_{\geq 0}$ and $P_{\mathsf{mf}}, N_{\mathsf{mf}}$ be integers with the following properties: (i) $P_{\mathsf{mf}}, N_{\mathsf{mf}} \to \infty$ and (ii) $P_{\mathsf{mf}}/N_{\mathsf{mf}} = \alpha < \infty$, and $P_{\mathsf{mf}}$ be divisible by $P$. We define the mean-field manifolds $\mathcal{M}_{\mathsf{mf}}(P_{\mathsf{mf}}, N_{\mathsf{mf}}) = \{\mathcal{M}_{\mathsf{mf}}^{a,i}\}_{a\in[P_{\mathsf{mf}}/P], i\in[P]}$ as follows.*

- *First, find an orthogonal basis $\{\mathbf{e}_k\}_{k=1}^N$ in $\mathbb{R}^N$ for the basis vectors of all the data manifolds. Namely, for each $i \in [P]$, there exists a linear transformation $Q^i \in \mathbb{R}^{(D_i+1)\times N}$ such that $\mathbf{u}_j^i = \sum_k Q_k^{i,j} \mathbf{e}_k$ for each $j \in \{0, 1, \ldots, D_i\}$.*

- *Next, for each $a \in [P_{\mathsf{mf}}/P]$, generate $\mathbf{v}_1^a, \ldots, \mathbf{v}_N^a \sim \mathcal{N}(0, I_{N_{\mathsf{mf}}})$ independently and let $\mathbf{V}^a$ be the $N_{\mathsf{mf}} \times N$ matrix with $\mathbf{v}_j^a$ on its columns.*

- *Define $M_{\mathsf{mf}}^{a,i} = \left\{(\mathbf{V}^a Q^i)_0 + \sum_{j=1}^{D_i} s_j(\mathbf{V}^a Q^i)_j : \mathbf{s} = (s_1, \ldots, s_{D_i}) \in \mathcal{S}_i\right\}$ as the $i$-th manifold in the $a$-th cloud where $(\mathbf{V}^a Q^i)_i = \sum_k \mathbf{v}_k^a Q_k^{i,j}$ for every $a \in [P_{\mathsf{mf}}/P]$ and $i \in [P]$.*

Now, we are ready to formally define the mean-field version of manifold capacity.

**Definition 3** (Mean-field manifold capacity Chung et al. (2018); Chou et al. (2024)). *Let $\{\mathcal{M}_i\}_{i\in[P]}$ be a collection of data manifolds in $\mathbb{R}^N$ as defined in Equation 3. The manifold capacity of $\{\mathcal{M}_i\}_{i\in[P]}$ is defined as*

$$\alpha_{\mathsf{mf}} := \lim_{N_{\mathsf{mf}}\to\infty} \max_{P_{\mathsf{mf}}} \left\{ \frac{P_{\mathsf{mf}}}{N_{\mathsf{mf}}} : \Pr_{\mathbf{y}, \mathcal{M}_{\mathsf{mf}}(P_{\mathsf{mf}}, N_{\mathsf{mf}})} \left[ \begin{matrix} \exists \theta\in\mathbb{R}^{N_{\mathsf{mf}}}, \forall a\in[P_{\mathsf{mf}}/P], i\in[P], \\ \min_{\mathbf{s}\in\mathcal{M}_{\mathsf{mf}}^{a,i}} y_i \langle\theta, \mathbf{s}\rangle \geq 0 \end{matrix} \right] \geq 1 - o_{N_{\mathsf{mf}}}(1) \right\}$$

*where and $o_{N_{\mathsf{mf}}}(1) \to 0$ as $N_{\mathsf{mf}} \to \infty$.*

Finally, previous work (Chung et al., 2018; Chou et al., 2024) derived a formula for mean-field manifold capacity as follows.

$$\alpha_{\mathsf{mf}}^{-1} = \frac{1}{P} \mathop{\mathbb{E}}_{\substack{\mathbf{y}\sim\{\pm 1\}^P \\ T\sim\mathcal{N}(0,I_N)}} \left[ \max_{\mathbf{s}_i\in\mathcal{M}_i} \left\{ \|\mathsf{proj}_{\mathsf{cone}(\{y_i\mathbf{s}_i\})} T\|_2^2 \right\} \right] \tag{4}$$

$$= \frac{1}{P} \mathop{\mathbb{E}}_{\substack{\mathbf{y}\sim\{\pm 1\}^P \\ T\sim\mathcal{N}(0,I_N)}} \left[ \max_{\substack{\mathbf{s}_i\in\mathcal{M}_i \\ \lambda_i\geq 0}} \left\{ \left( \frac{-T \cdot \sum_i \lambda_i y_i \mathbf{s}_i}{\|\sum_i \lambda_i y_i \mathbf{s}_i\|_2} \right)_+^2 \right\} \right]$$

where $\mathcal{N}(\mu, \Sigma)$ denotes the multivariate Gaussian distribution with mean $\mu$ and covariance $\Sigma$ and $\mathsf{cone}(\cdot)$ is the convex cone spanned by the vectors, i.e., $\mathsf{cone}(\{y_i\mathbf{s}_i\}) = \{\sum_i \lambda_i y_i \mathbf{s}_i : \lambda_i \geq 0\}$.

### B.4 EFFECTIVE GEOMETRIC MEASURES FROM CAPACITY FORMULA

The advantages of mean-field manifold capacity are: (i) $\alpha_{\mathsf{mf}}$ can be estimated via solving a quadratic program (Algorithm 1) and (ii) Equation 1 connects manifold capacity to the structure of the manifolds $\{\mathcal{M}_i\}$. Specifically, for each $\mathbf{y}, T$, define $\{\mathbf{s}_i(\mathbf{y}, T)\} = y_i \cdot \arg\max_{\{\mathbf{s}_i\}} \|\mathsf{proj}_{\mathsf{cone}(\{y_i\mathbf{s}_i\})} T\|_2^2$

as the *anchor points* with respect to $\mathbf{y}$ and $T$. Intuitively, these anchor points are the support vectors with respect to some random projection and dichotomy as in Definition 1. Specifically, these anchor points are analytically linked to manifold capacity via Equation 1 and are distributed over the manifolds $\{\mathcal{M}_i\}$. This connection inspired the previous work (Chung et al., 2018; Chou et al., 2024) to define the following effective manifold geometric measures that capture the structure of manifolds while being analytically connected to capacity.

The first key idea of defining effective geometric measure is the segregation of anchor points into their *center part* and their *axis part*. Concretely, for each $i \in [P]$, define $\mathbf{s}_i^0 := \mathbb{E}_{\mathbf{y},T}[\mathbf{s}_i(\mathbf{y},T)]$ as the center of the $i$-th manifold and define $\mathbf{s}_i^1(\mathbf{y},T) := \mathbf{s}_i(\mathbf{y},T) - \mathbf{s}_i^0$ to be the axis part of $\mathbf{s}_i(\mathbf{y},T)$ for each pair of $(\mathbf{y},T)$.

Next, (Chung et al., 2018) used an identity: $a = \frac{b}{1+\frac{b-a}{a}}$, and set $a = \|\mathsf{proj}_{\mathsf{cone}(\{\mathbf{s}_i(\mathbf{y},T)\}_i)} T\|_2^2$ and $b = \|\mathsf{proj}_{\mathsf{cone}(\{\mathbf{s}_i^1(\mathbf{y},T)\}_i)} T\|_2^2$ to rewrite the capacity formula (Equation 4) as follows.

$$\alpha_{\mathsf{mf}}^{-1} = \frac{1}{P} \mathbb{E}_{\mathbf{y},T} \left[ \|\mathsf{proj}_{\mathsf{cone}(\{\mathbf{s}_i(\mathbf{y},T)\}_i)} T\|_2^2 \right]$$

$$= \frac{1}{P} \mathbb{E}_{\mathbf{y},T} \left[ \frac{\|\mathsf{proj}_{\mathsf{cone}(\{\mathbf{s}_i^1(\mathbf{y},T)\}_i)} T\|_2^2}{1 + \frac{\|\mathsf{proj}_{\mathsf{cone}(\{\mathbf{s}_i^1(\mathbf{y},T)\}_i)} T\|_2^2 - \|\mathsf{proj}_{\mathsf{cone}(\{\mathbf{s}_i(\mathbf{y},T)\}_i)} T\|_2^2}{\|\mathsf{proj}_{\mathsf{cone}(\{\mathbf{s}_i(\mathbf{y},T)\}_i)} T\|_2^2}} \right].$$

Then, they proceeded with the following approximation.

$$\approx \frac{\frac{1}{P} \mathbb{E}_{\mathbf{y},T} \left[ \|\mathsf{proj}_{\mathsf{cone}(\{\mathbf{s}_i^1(\mathbf{y},T)\}_i)} T\|_2^2 \right]}{\mathbb{E}_{\mathbf{y},T} \left[ 1 + \frac{\|\mathsf{proj}_{\mathsf{cone}(\{\mathbf{s}_i^1(\mathbf{y},T)\}_i)} T\|_2^2 - \|\mathsf{proj}_{\mathsf{cone}(\{\mathbf{s}_i(\mathbf{y},T)\}_i)} T\|_2^2}{\|\mathsf{proj}_{\mathsf{cone}(\{\mathbf{s}_i(\mathbf{y},T)\}_i)} T\|_2^2} \right]}. \tag{5}$$

(Chung et al., 2018; Chou et al., 2024) found that the above approximation empirically performs well. Furthermore, as the numerator mimics the notion of Gaussian width of a convex body and the denominator behaves like (normalized) radius of a sphere, they defined effective manifold dimension and radius as follows.

- **Manifold dimension** captures the degree of freedom of the noises/variations within the manifolds. Formally, it is defined as $D_{\mathsf{mf}} := \mathbb{E}_{\mathbf{y},T}[\|\mathsf{proj}_{\mathsf{cone}(\{\mathbf{s}_i^1(\mathbf{y},T)\}_i)} T\|_2^2]$.
- **Manifold radius** captures the noise-to-signal ratio of the manifolds. Formally, it is defiend as

$$R_{\mathsf{mf}} := \sqrt{\mathbb{E}_{\mathbf{y},T} \left[ \frac{\|\mathsf{proj}_{\mathsf{cone}(\{\mathbf{s}_i(\mathbf{y},T)\}_i)} T\|^2}{\|\mathsf{proj}_{\mathsf{cone}(\{\mathbf{s}_i^1(\mathbf{y},T)\}_i)} T\|^2 - \|\mathsf{proj}_{\mathsf{cone}(\{\mathbf{s}_i(\mathbf{y},T)\}_i)} T\|^2} \right]}.$$

While (Chung et al., 2018) focusing on the cases where there are no correlations between manifolds, (Chou et al., 2024) extended the theory to incorporate manifold correlations. Hence, they further defined the following metrics for measuring the alignment between manifolds.

- **Center alignment** captures the correlation between the center of different manifolds. Formally, it is defined as $\rho_{\mathsf{mf}}^c := \frac{1}{P(P-1)} \sum_{i \neq j} |\langle \mathbf{s}_i^0, \mathbf{s}_j^0 \rangle|$.
- **Axis alignment** captures the correlation between the axis of different manifolds. Formally, it is defined as $\rho_{\mathsf{mf}}^a := \frac{1}{P(P-1)} \sum_{i \neq j} \mathbb{E}_{\mathbf{y},T}[|\langle \mathbf{s}_i^1(\mathbf{y},T), \mathbf{s}_j^1(\mathbf{y},T) \rangle|]$.
- **Center-axis alignment** captures the correlation between the center and axis of different manifolds. Formally, it is defined as $\psi_{\mathsf{mf}} := \frac{1}{P(P-1)} \sum_{i \neq j} \mathbb{E}_{\mathbf{y},T}[|\langle \mathbf{s}_i, \mathbf{s}_j^1(\mathbf{y},T) \rangle|]$.

**A capacity approximation formula by dimension and radius.** Recall that in Equation 5 previous work (Chung et al., 2018) used the identity $a = \frac{b}{1+\frac{b-a}{a}}$ to approximate the manifold capacity. After defining manifold dimension and radius, one can then plug them back to Equation 5 and get the following approximation of manifold capacity via effective manifold dimension and radius.

$$\alpha_{\mathsf{mf}} \approx \frac{1 + R_{\mathsf{mf}}^{-2}}{D_{\mathsf{mf}}}. \tag{6}$$

### B.5 CONNECTIONS BETWEEN MANIFOLD CAPACITY AND ITS EFFECTIVE GEOMETRIC MEASURES

Here, we demonstrate the connections between manifold capacity and its effective geometric measures by synthetic manifolds. In particular, we consider isotropic Gaussian clouds parametrized by a set of *ground truth* latent parameters: dimension $D_{\text{ground}}$, radius $R_{\text{ground}}$, center correlations $\rho^c\text{ground}$, axis correlations $\rho^a_{\text{ground}}$, and center-axis correlations $\psi_{\text{ground}}$. See Section D.1.1 for more details on the generative process. In this section, we focus on showing that the effective geometric measures $D_{\text{mf}}, R_{\text{mf}}, \rho^c_{\text{mf}}, \rho^a_{\text{mf}}, \psi_{\text{mf}}$ capture the corresponding ground truth parameter.

**Effective manifold dimension and radius.** We first set all the manifold correlations to be zero and vary the ground truth radius and dimension. Here we pick $N = 1000$ neurons, $P = 2$ manifold, $M = 200$ points per manifold, varying the underlying dimension from 2 to 10, and varying the underlying radius from 0.8 to 2. In Figure 8, we vary the ground truth dimension in the x-axis, and in Figure 9, we vary the ground truth radius in the x-axis.

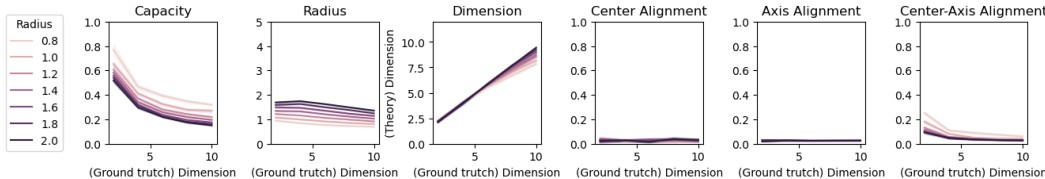

Figure 8: Effective manifold dimension tracks the ground truth dimension of uncorrelated isotropic Gaussian clouds. Note that the higher the dimension, the smaller capacity, as discussed in Figure 2c.

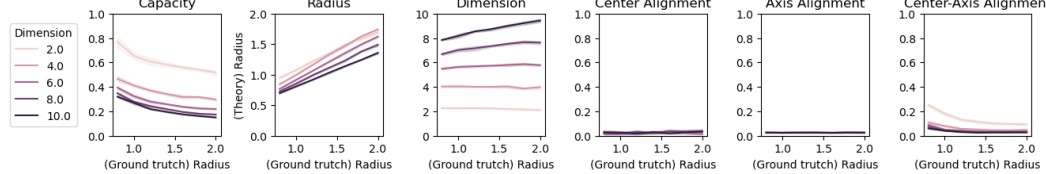

Figure 9: Effective manifold radius tracks the ground truth radius of uncorrelated isotropic Gaussian clouds. Note that the higher the radius, the smaller capacity, as discussed in Figure 2c.

**Effective alignment measures.** Next, we fix the ground truth dimension to be $D_{\text{ground}} = 4$ and radius to be $R_{\text{ground}} = 1$ and vary $\rho^c\text{ground}, \rho^a\text{ground}, \psi_{\text{ground}}$ from 0 to 0.8. In Figure 10, we vary the center correlations, and in Figure 11, we vary the axis correlations.

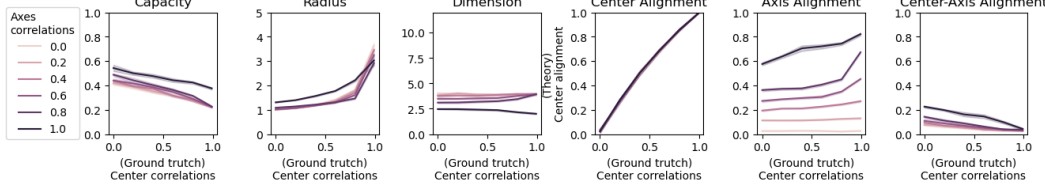

Figure 10: Effective manifold center alignment tracks the ground truth center correlations of isotropic Gaussian clouds. Note that the higher the center alignment, the smaller capacity, as discussed in Figure 2c. Also, in the large center correlations regime, the effective radius increases.

### B.6 ALGORITHMS FOR ESTIMATING MANIFOLD CAPACITY AND EFFECTIVE GEOMETRIC MEASURE

We provide pseudocodes for estimating manifold capacity and effective geometric measure in Algorithm 1.

---

**Algorithm 1** Estimate manifold capacity and effective geometric measures

---

**Input:** $\{\mathcal{M}_i\}$: $P$ point clouds, each containing $M$ points in an $N$-dimensional ambient space; $n_t$: number of samples for estimating the expectation.
**Output:** $\alpha_{\mathsf{mf}}$: Manifold capacity; $D_{\mathsf{mf}}$: Effective dimension; $R_{\mathsf{mf}}$: Effective radius; $\rho_{\mathsf{mf}}^a$: Effective axis alignment; $\rho_{\mathsf{mf}}^c$: Effective center alignment; $\psi_{\mathsf{mf}}$: Effective center-axis alignment.

  % Step 1: Sample anchor points.
  **for** $k$ from 1 to $n_t$ **do**
     $T_k \leftarrow$ a vector sampled from isotropic $N$-dimensional Gaussian distribution.
     $\mathbf{y} \leftarrow$ a random dichotomy vector from $\{\pm 1\}^P$.
     $\mathbf{A} \leftarrow I_N; \mathbf{q} \leftarrow -T_k; \mathbf{h} \leftarrow \mathbf{0}_N$.
     $\mathbf{G} \leftarrow (y \odot \{\mathcal{M}_i\}_{i=1}^P)$.              $\triangleright \mathbf{G}_{i,j} = y_i \mathbf{s}$ is a row vector where $\mathbf{s}$ is the $j$-th point in $\mathcal{M}_i$.
     $\mathsf{output} \leftarrow qp(\mathbf{A}, \mathbf{q}, \mathbf{G}, \mathbf{h})$.           $\triangleright \min_{\mathbf{x}} \frac{1}{2}\mathbf{x}^\top \mathbf{A}\mathbf{x} + \mathbf{q}^\top \mathbf{x}$ s.t. $\mathbf{G}\mathbf{x} \leq \mathbf{h}$.
     $\mathbf{z}_{\mathsf{dual}} \leftarrow \mathsf{output}[\text{"dual"}]$               $\triangleright$ The support vectors
     **for** $i$ from 1 to $P$ **do**
        $\mathbf{s}_i[k] \leftarrow \sum_j (\mathbf{z}_{\mathsf{dual}})_{i,j}^\top \mathbf{G} / \sum_j (\mathbf{z}_{\mathsf{dual}})_{i,j}$
     **end for**
  **end for**

  % Step 2: Estimate (anchor) manifold centers.
  **for** $i$ from 1 to $P$ **do**
     $\mathbf{s}_i^0 \leftarrow \frac{1}{n_t} \sum_{k=1}^{n_t} \mathbf{s}_i[k])$.
  **end for**
  $\mathbf{G}^0 \leftarrow \sum_i \mathbf{s}_i^0 (\mathbf{s}_i^0)^\top$.                          $\triangleright$ Anchor center gram matrix.

  % Step 3: Separate the center and axis part of anchor points.
  **for** $k$ from 1 to $n_t$ **do**
     **for** $i$ from 1 to $P$ **do**
        $\mathbf{s}_i^1[k] \leftarrow \mathbf{s}_i[k] - \mathbf{s}_i^0$.         $\triangleright$ The axis part of the anchor poitn in the $i$-th manifold.
     **end for**
     $T^1[k] \leftarrow \sum_i \mathbf{s}_i^1[k] T_k$.
     $\mathbf{G}^1[k] \leftarrow \sum_i \mathbf{s}_i^1[k](\mathbf{s}_i^1[k])^\top$.           $\triangleright$ Anchor axis gram matrix.
  **end for**

  % Step 4: Estimate manifold capacity and effective geometric measures.
  $\alpha_{\mathsf{mf}} \leftarrow (\frac{1}{n_t P} \sum_{k=1}^{n_t} (\mathbf{s}_i[k]T_k)^\top (\mathbf{s}_i[k](\mathbf{s}_i[k]^\top)^\dagger (\mathbf{s}_i[k]T_k))^{-1}$.
  $D_{\mathsf{mf}} \leftarrow \frac{1}{n_t P} \sum_{k=1}^{n_t} T^1[k]^\top \mathbf{G}^1[k]^\dagger T^1[k]$.
  $R_{\mathsf{mf}} \leftarrow \sqrt{\frac{1}{n_t} \sum_{k=1}^{n_t} \frac{T^1[k]^\top (\mathbf{G}^1[k]+\mathbf{G}^0)^\dagger T_1[k]}{T_1[k]^\top (\mathbf{G}^1[k]+\mathbf{G}^1[k](\mathbf{G}^0)^\dagger \mathbf{G}^1[k])^\dagger T^1[k]}}$.   $\triangleright$ Equivalent to the definition of radius after applying the Woodbury formula for numerical stabiltiy.
  $\rho_{\mathsf{mf}}^c \leftarrow \frac{1}{P(P-1)} \sum_{i=1}^P \sum_{i \neq j} \frac{(\mathbf{s}_i^0)^\top \mathbf{s}_j^0}{\|\mathbf{s}_i^0\|_2 \cdot \|\mathbf{s}_j^0\|_2}$.
  $\rho_{\mathsf{mf}}^a \leftarrow \frac{1}{P(P-1)} \sum_{i=1}^P \sum_{j \neq i} \frac{1}{n_k} \sum_{k=1}^{n_k} \frac{\mathbf{s}_i^1[k]^\top \mathbf{s}_j^1[k]}{\|\mathbf{s}_i^1[k]\|_2 \cdot \|\mathbf{s}_j^1[k]\|_2}$.
  $\psi_{\mathsf{mf}} \leftarrow \frac{1}{P(P-1)} \sum_{i=1}^P \sum_{j \neq i} \frac{1}{n_k} \sum_{k=1}^{n_k} \frac{(\mathbf{s}_i^0)^\top \mathbf{s}_j^1[k]}{\|\mathbf{s}_i^0\|_2 \cdot \|\mathbf{s}_j^1[k]\|_2}$.
  **return** $\alpha_{\mathsf{mf}}, D_{\mathsf{mf}}, R_{\mathsf{mf}}, \rho_{\mathsf{mf}}^a, \rho_{\mathsf{mf}}^c, \psi_{\mathsf{mf}}$.

---

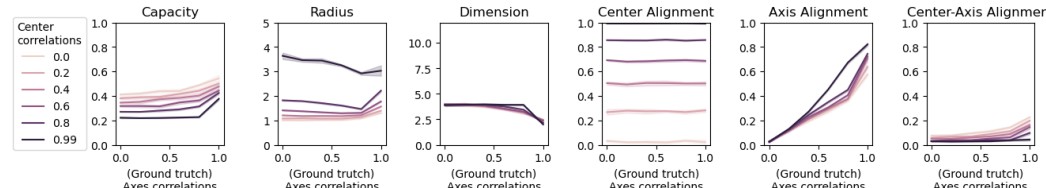

Figure 11: Effective manifold axis alignment tracks the ground truth axis correlations of isotropic Gaussian clouds. Note that the higher the axis alignment, the higher capacity, as discussed in Figure 2c. Also, in the large axis correlations regime, the effective dimension decreases.

## C  THEORETICAL RESULTS

### C.1  FORMAL STATEMENT OF THEOREM 1

Let $d \in \mathbb{N}$ be the input dimension and $N \in \mathbb{N}$ be the number of hidden units. Let $W_0 \in \mathbb{R}^{N \times d}$ be the weight matrix of a fully connected 2-layer neural network. The feature of an input vector is defined as $\Phi_0(\mathbf{x}) = \sigma(W_0\mathbf{x})$ where $\sigma(\cdot) : \mathbb{R} \to \mathbb{R}$ is a non-linear activation function, e.g., ReLU or tanh. The readout weight is denoted as $\mathbf{a} \in \mathbb{R}^N$. Finally, the output of the 2-layer NN is the sign of the readout, i.e., $f(\mathbf{x}) = \mathsf{sgn}(\mathbf{a}^\top \Phi(\mathbf{x}))$.

Let $\{(\mathbf{x}_i, y_i)\}_{i \in [P_{\text{train}}]}$ be the collection of training data. We consider gradient descent over the mean square error (MSE) of the 2-layer NN, i.e., $\mathcal{L}(f) = \frac{1}{P_{\text{train}}} \sum_{i \in [P_{\text{train}}]} \ell(f(\mathbf{x}_i), y_i)$ where $\ell(z_i, y_i) = \frac{1}{2}(z - y)^2$. The gradient update with learning rate $\eta > 0$ is $W_{t+1} = W_t + \eta G_t$ where

$$G_t = \frac{1}{P_{\text{train}}} \sum_{i \in [P_{\text{train}}]} \left[ (y_i - \mathbf{a}^\top \sigma(W_t \mathbf{x}_i)) \mathbf{a} \odot \sigma'(W_t \mathbf{x}_i) \right] \mathbf{x}_i^\top$$

and $\sigma'(\cdot)$ denotes the first order derivative of $\sigma(\cdot)$.

**Assumption 1.** *We adopt the following assumptions used in (Montanari et al., 2019; Ba et al., 2022).*

1. *(Proportional limit) $P_{train}, d, N \to \infty$ with $\psi_1 = N/d$, $\psi_2 = P_{train}/d$, and $0 < \psi_1, \psi_2 < \infty$.*

2. *(Gaussian initialization) $[W_0]_{kj} \sim \mathcal{N}(0, 1/N)$ for each $k \in [N]$ and $j \in [d]$.*

3. *(Gaussian readout) $a_k \sim \mathcal{N}(0, 1/N)$ for each $k \in [N]$.*

4. *(Normalized activation) The non-linear activation function $\sigma(\cdot)$ has $O(1)$-bounded first three derivatives almost surely. In addition, $\mathbb{E}[\sigma(G)] = 0$ and $\mathbb{E}[G\sigma(G)] \neq 0$ for $G \sim \mathcal{N}(0, 1)$.*

5. *(Non-degenerate label function) Let $F : \mathbb{R} \to [0, 1]$ be a continuous function satisfying*

$$inf\{x : \Pr[T < x] > 0\} = -\infty \ \text{ and } \ sup\{x : \Pr[T > x] > 0\} = \infty$$

*where $T = YG$, $G \sim \mathcal{N}(0, 1)$, and $\Pr[Y = 1 \,|\, G] = 1 - \Pr[Y = -1 \,|\, G] = F(G)$.*

**Setting 1.** *We consider the following data generation process. Let $F : \mathbb{R} \to [0, 1]$ be a function satisfying Assumption 1. Let $\beta_* \in \mathbb{R}^d$ be a hidden vector with $\|\beta_*\|_2 = 1$. The data distribution $\mathcal{D}_F(\beta_*)$ is defined by the following two steps: (i) sample $\mathbf{x} \sim \mathcal{N}(0, I_d)$, and (ii) sample $y$ with $\Pr[y = 1] = 1 - \Pr[y = -1] = F(\langle \beta_*, \mathbf{x} \rangle)$. Finally, the prediction accuracy of a network is defined as the expected accuracy of a fresh sample, i.e., $\Pr_{(\mathbf{x},y) \sim \mathcal{D}_F(\beta_*)}[yf(\mathbf{x}) \geq 0]$.*

**Parameter 1.** *Given $\psi_1, \psi_2, F, \beta_*$ from Assumption 1 and Setting 1. We define the following parameters.*

$$\gamma_1 = \mathop{\mathbb{E}}_{G \sim \mathcal{N}(0,1)} [G\sigma(G)]$$

$$\gamma_2^2 = \mathop{\mathbb{E}}_{G \sim \mathcal{N}(0,1)} [\sigma(G)^2] - \mathop{\mathbb{E}}_{G \sim \mathcal{N}(0,1)} [G\sigma(G)]^2$$

$$\theta_1 = \mathop{\mathbb{E}}_{X \sim \mu_{\psi_1}} \left[ \frac{\gamma_1^2}{\gamma_1^2 X + \gamma_2^2} \right]$$

$$\theta_2 = \psi_1 \mathop{\mathbb{E}}_{X \sim \mu_{\psi_1}} \left[ \frac{\gamma_1^2 X}{\gamma_1^2 X + \gamma_2^2} \right]$$

$$\theta_3 = \mathop{\mathbb{E}}_{(G,Y) \sim \mathcal{D}_F} [YG]$$

$$\theta_4 = \left( \frac{1}{\psi_2} + \mathop{\mathbb{E}}_{(G,Y),(G',Y') \overset{i.i.d.}{\sim} \mathcal{D}_F} [YY'GG'] \right)$$

*where $\mu_{\psi_1}$ is the Marchenko-Pastur distribution with the ratio parameter being $\psi_1$ and $(G, Y) \sim \mathcal{D}_F$ is defined as the sampling process: $G \sim \mathcal{N}(0,1)$ and $\Pr[Y = 1] = 1 - \Pr[Y = -1] = F(G)$.*

**Theorem 2.** *Given Assumption 1 and consider $0 < \psi_1, \psi_2, \eta < \infty$.*

1. *(Capacity tracks the degree of feature learning) The storage capacity of 2-layer network trained with synthetic data defined in Setting 1 after one gradient step is $\alpha_{P_{train}, d, N}(\psi_1, \psi_2, \eta)$ and*

$$\alpha_{P_{train}, d, N}(\psi_1, \psi_2, \eta) \xrightarrow{P_{train}, d, N \to \infty} \alpha(\psi_1, \psi_2, \eta)$$

   *Here the function $\alpha(\cdot)$ is defined as*

$$\alpha(\psi_1, \psi_2, \eta) = \left( \min_{c \in \mathbb{R}} \mathop{\mathbb{E}}_{(Z,G,Y) \sim \mathcal{D}_{\psi_1, \psi_2, \eta}} \left[ (-cYG - Z)_+^2 \right] \right)^{-1}$$

   *where $(Z, G, Y) \sim \mathcal{D}_{\psi_1, \psi_2, \eta}$ is defined as the following sampling process*

$$Z \sim \mathcal{N}(0,1), \ G \sim \mathcal{N}(0,1), \ \Pr[Y = 1] = 1 - \Pr[Y = -1] = f_{\tau(\psi_1, \psi_2, \eta)}(G)$$

   *and the scalar function $f_\tau(\cdot)$ and $\tau(\psi_1, \psi_2, \eta)$ are defined as*

$$f_\tau(G) = \mathop{\mathbb{E}}_{G' \sim \mathcal{N}(0,1)} \left[ F(\sqrt{1 - \tau^2} G + \tau G') \right]$$

   *and*

$$\tau = \tau(\psi_1, \psi_2, \eta) = \sqrt{\tau_0(\psi_1, \psi_2)^2 - \tau_\Delta(\psi_1, \psi_2, \eta)^2}$$

   *where $\tau_0(\cdot)$ and $\tau_\Delta(\cdot)$ are scalar functions defined as*

$$\tau_0(\psi_1, \psi_2)^2 = 1 - \theta_2$$

   *and*

$$\tau_\Delta(\psi_1, \psi_2, \eta)^2 = \frac{\eta^2 \theta_1 (1 - \theta_2)^2 \theta_3^2}{1 + \eta^2 \theta_1 (1 - \theta_2) \theta_4}$$

   *where the parameters $\theta_i$'s are defined in Parameter 1. In particular, $0 < \alpha(\psi_1, \psi_2, \eta) < \alpha(\psi_1, \psi_2, \eta')$ for all $0 < \eta < \eta'$.*

2. *(Capacity analytically links to prediction accuracy) The prediction accuracy of 2-layer network trained with synthetic data defined in Setting 1 after one gradient step is $\mathsf{Acc}_{P_{train}, d, N}(\psi_1, \psi_2, \eta)$ and*

$$\mathsf{Acc}_{P_{train}, d, N}(\psi_1, \psi_2, \eta) \xrightarrow{P_{train}, d, N \to \infty} \mathsf{Acc}(\psi_1, \psi_2, \eta)$$

   *Here the function $\mathsf{Acc}(\cdot)$ is defined as*

$$\mathsf{Acc}(\psi_1, \psi_2, \eta) = \mathop{\mathbb{E}}_{(G,Y) \sim \mathcal{D}_F} \left[ \Phi \left( \frac{\eta \gamma_1^2 \theta_3}{\sqrt{\frac{\eta^2 \gamma_1^4}{\psi_2} + \gamma_1^2 + \gamma_*^2}} YG \right) \right]$$

   *In particular, there exists an increasing and invertible function $g_{\psi_1, \psi_2} : [0, 1] \to \mathbb{R}_+$ such that*

$$\mathsf{Acc}(\psi_1, \psi_2, \eta) = g_{\psi_1, \psi_2}(\alpha(\psi_1, \psi_2, \eta)).$$

## C.2 Proof for Theorem 2

**Step 1: Rank-1 approximation of gradient descent in 2-layer networks by ref. (Ba et al., 2022).**
When the learning rate is constant, i.e., $\eta = O(1)$, ref. (Ba et al., 2022) shows that the gradient update matrix can be approximated by a rank-1 matrix. In particular, the following is a restatement of Proposition 2 in (Ba et al., 2022).

**Proposition 1** (Proposition 2 in (Ba et al., 2022)). *Given Assumption 1 and Setting 1, there exist some constants $c, C > 0$ such that for all large $P_{train}, N, d$, the following holds*

$$\left\| G_0 - \gamma_1 \mathbf{a} \left( \frac{\sum_i y_i \mathbf{x}_i^\top}{P_{train}} \right) \right\| \leq \frac{C \log^2 P_{train}}{\sqrt{P_{train}}} \cdot \|G_0\|$$

*with probability at least $1 - P_{train} e^{-c \log^2 P_{train}}$ and $\| \cdot \|$ denotes the operator norm.*

**Step 2: A formula for the storage capacity of a Gaussian model by ref. (Montanari et al., 2019).**
The storage capacity of a Gaussian model is proven in (Montanari et al., 2019). In particular, the following is a restatement of the Proposition 5.1 in (Montanari et al., 2019).

**Definition 4** (Gaussian model). *Let $\theta_* \in \mathbb{R}^N$ be some latent vector. A sample $(\mathbf{x}_i, y_i) \in \mathbb{R}^N \times \{\pm 1\}$ is i.i.d. sampled as follows. First, sample $\mathbf{x}_i$ from $\mathcal{N}(0, \Sigma)$ where $\Sigma$ is a covariance matrix satisfying certain technical condition as defined in Assumption 1-2 in (Montanari et al., 2019). Next, let $y_i = +1$ with probability $f(\langle \theta_*, \mathbf{x}_i \rangle)$ for some function $f$ satisfying Assumption 3 in (Montanari et al., 2019).*

**Proposition 2** (Theorem 3 in (Montanari et al., 2019)). *Consider a Gaussian model satisfying Definition 4. As $P_{train}, N, d \to \infty$, the storage capacity converges to*

$$\alpha^* = \left( \min_{c \in \mathbb{R}} \mathbb{E}_{(Z,G,Y) \sim \mathcal{D}_f} \left[ (-cYG - Z)_+^2 \right] \right)^{-1}$$

*where $(Z, G, Y) \sim \mathcal{D}_f$ is defined as the following sampling process*

$$Z \sim \mathcal{N}(0,1), \ G \sim \mathcal{N}(0,1), \ \Pr[Y=1] = 1 - \Pr[Y=-1] = f(\rho \cdot G).$$

*where $\rho$ is some scalar related to the Gaussian model as defined in Assumption 2 of (Montanari et al., 2019).*

Note that the capacity only depends on the alignment between data and task (as encoded in $f$) and does not depend on the covariance structure. The dependence on the covariance structure will appear when one considers the non-zero margin version of capacity.

**Step 3: A Gaussian equivalent model for 2-layer NNs after one gradient step.** Next, we combine a Gaussian equivalent model for random feature 2-layer NNs in (Montanari et al., 2019) (Theorem 3) and the rank-1 approximation of gradient step in Proposition 1 to get a Gaussian equivalent model for 2-layer NNs after one gradient step.

**Proposition 3.** *Given Assumption 1 and $0 < \psi_1, \psi_2, \eta < \infty$. Let $d \in \mathbb{N}$ and $(W_1, \beta_*, F)$ be the weight matrix, hidden vector, and label function from Setting 1. Let $\alpha^{GM}_{P_{train},d,N}(\psi_1, \psi_2, \eta)$ be the capacity of the following Gaussian model:*

$$
\begin{aligned}
\Sigma_{d,\eta} &= \gamma_1^2 W_1 W_1^\top + \gamma_*^2 I \\
\theta_{*,d,\eta} &= \alpha_{d,\eta}^{-1} \gamma_1 (\gamma_1^2 W_1 W_1^\top + \gamma_*^2 I)^{-1} W_1 \beta_* \\
\alpha_{d,\eta}^2 &= \gamma_1^2 \beta_*^\top W_1^\top (\gamma_1^2 W_1 W_1^\top + \gamma_*^2 I)^{-1} W_1 \beta_* \\
\tau_{d,\eta}^2 &= 1 - \alpha_{d,\eta}^2 \\
f_{d,\eta}(x) &= \mathbb{E}_{G \sim \mathcal{N}(0,1)} [F(\alpha_{d,\eta} x + \tau_{d,\eta} G)].
\end{aligned}
\tag{7}
$$

*We have that*

$$\lim_{P_{train},d,N \to \infty} |\alpha_{P_{train},d,N}(\psi_1, \psi_2, \eta) - \alpha^{GM}_{P_{train},d,N}(\psi_1, \psi_2, \eta)| = 0$$

*and*

$$\alpha^{GM}_{P_{train},d,N}(\psi_1, \psi_2, \eta) \xrightarrow{P_{train},d,N\to\infty} \alpha(\psi_1, \psi_2, \eta).$$

*Here the function $\alpha(\cdot)$ is defined as*

$$\alpha(\psi_1, \psi_2, \eta) = \left(\min_{c\in\mathbb{R}} \mathbb{E}_{(Z,G,Y)\sim\mathcal{D}_{f_\tau(\psi_1,\psi_2,\eta)}} \left[(-cYG - Z)_+^2\right]\right)^{-1}$$

*where the scalar function $f_\tau(\cdot)$ and $\tau(\psi_1, \psi_2, \eta)$ are defined as*

$$f_\tau(G) = \mathbb{E}_{G'\sim\mathcal{N}(0,1)} \left[F(\sqrt{1-\tau^2}G + \tau G')\right]$$

*and*

$$\tau = \tau(\psi_1, \psi_2, \eta) = \lim_{d\to\infty} \tau_{d,\eta} = \sqrt{\tau_0(\psi_1,\psi_2)^2 - \tau_\Delta(\psi_1,\psi_2,\eta)^2}.$$

*where $\tau_0(\psi_1,\psi_2) = \lim_{d\to\infty}\tau_{d,0}$.*

To derive the Gaussian equivalent model in Proposition 3 of the random features model after one gradient step defined in Setting 1, we analyze the following random features and their associated labels:

$$\Phi_0(\mathbf{x}_i) = \sigma(W_1\mathbf{x}_i), \quad \Pr[y_i = 1|\mathbf{x}_i] = 1 - \Pr[y_i = -1|\mathbf{x}_i] = F(\langle\beta_*, \mathbf{x}_i\rangle), \quad \|\beta_*\|_2 = 1$$

where $\mathbf{x}_i \sim \mathcal{N}(0, I_d)$ and $W_1 = W_0 + \eta G_0$ while $G_0$ satisfies the bound given in Proposition 1. Given the assumptions in Assumption 1, we can decompose the nonlinear activation function $\sigma$ into Hermite polynomials. Following our parameters in Parameter 1, we define the Gaussian equivalent features of our model as the linearization of Equation C.2:

$$\mathbf{g}_i = \gamma_1 W_1\mathbf{x}_i + \gamma_2 \mathbf{h}_i$$

where $\mathbf{h}_i \sim \mathcal{N}(0, I_N)$ are independent from everything else. Now, we wish to find a similar linearized Gaussian model for the labels $y_i$ given the Gaussian equivalent features $\mathbf{g}_i$. It is easy to check that the Gaussian features has the following covariance:

$$\mathbf{g}_i \sim \mathcal{N}(0, \Sigma_{d,\eta}), \quad \Sigma_{d,\eta} = \gamma_1^2 W_1 W_1^\top + \gamma_*^2 I$$

By matching covariance through Equation C.2, we obtain

$$\mathbf{x}_i = \gamma_1 W_1^\top \Sigma_{d,\eta}^{-1}\mathbf{g}_i + Q^{1/2}\tilde{\mathbf{h}}_i$$

where $Q = \gamma_2^2(\gamma_2^2 I_N + \gamma_1^2 W_1^\top W_1)^{-1}$ and $\tilde{\mathbf{h}}_i \sim \mathcal{N}(0, I_N)$ are independent of $\mathbf{x}_i$. Therefore, we can rewrite the label function parameter as

$$\langle\beta_*, \mathbf{x}_i\rangle = \alpha_{d,\eta}\langle\theta_{*,d,\eta}, \mathbf{g}_i\rangle + \varepsilon_i$$

where $\varepsilon_i \sim \mathcal{N}(0, \tau_{d,\eta}^2)$ are independent of $\mathbf{g}_i$. Effectively, we obtain an equivalent label function

$$f_{d,\eta}(x) = \mathbb{E}_{G\sim\mathcal{N}(0,1)}[F(\alpha_{d,\eta}x + \tau_{d,\eta}G)]$$

such that $\Pr[y_i = 1|\mathbf{x}_i] = 1 - \Pr[y_i = -1|\mathbf{x}_i] = f_{d,\eta}(\langle\theta_{*,d,\eta}, \mathbf{g}_i\rangle)$. It is easy to verify that this Gaussian model satisfies the assumptions in Definition 4.

**Step 4: Analysis of $\tau$.** Finally, we combine Proposition 1 and Proposition 3 to get the formula for the right hand side of Equation 7. From Proposition 1, we approximate $W_1$ as $W_1 = W_0 + \mathbf{a}\mathbf{u}^\top$ where $\mathbf{u} = \eta\sum_i y_i\mathbf{x}_i^\top/P_{train}$. To rewrite the right hand side of Equation 7, we first deal with the matrix inverse term using the same trick as in ref. (Ba et al., 2022). Let $\Sigma_t = \gamma_1^2 W_t W_t^\top + \gamma_*^2 I$. Observe that

$$\Sigma_1 = \Sigma_0 + \gamma_1^2 \begin{bmatrix}\mathbf{a} & \mathbf{c}\end{bmatrix} \begin{bmatrix}L_1 & 1 \\ 1 & 0\end{bmatrix} \begin{bmatrix}\mathbf{a}^\top \\ \mathbf{c}^\top\end{bmatrix}$$

where $\mathbf{c} = W_0\mathbf{u}$. By Sherman-Morrison-Woodbury formula, we have

$$\Sigma_1^{-1} = \Sigma_0^{-1} - \gamma_1^2\Sigma_0^{-1}\begin{bmatrix}\mathbf{a} & \mathbf{c}\end{bmatrix}\left(\begin{bmatrix}L_1 & 1 \\ 1 & 0\end{bmatrix}^{-1} + \gamma_1^2\begin{bmatrix}\mathbf{a}^\top \\ \mathbf{c}^\top\end{bmatrix}\Sigma_0^{-1}\begin{bmatrix}\mathbf{a} & \mathbf{c}\end{bmatrix}\right)^{-1}\begin{bmatrix}\mathbf{a}^\top \\ \mathbf{c}^\top\end{bmatrix}\Sigma_0^{-1}$$

$$= \Sigma_0^{-1} - \Delta_{aa} - \Delta_{cc} + \Delta_{ac} + \Delta_{ca}$$

where

$$\Delta_{aa} = \gamma_1^2 \frac{L_4 - L_1}{D} \Sigma_0^{-1} \mathbf{a}\mathbf{a}^\top \Sigma_0^{-1}$$

$$\Delta_{cc} = \gamma_1^2 \frac{L_3}{D} \Sigma_0^{-1} \mathbf{c}\mathbf{c}^\top \Sigma_0^{-1}$$

$$\Delta_{ac} = \gamma_1^2 \frac{1 + L_6}{D} \Sigma_0^{-1} \mathbf{a}\mathbf{c}^\top \Sigma_0^{-1}$$

$$\Delta_{ca} = \gamma_1^2 \frac{1 + L_6}{D} \Sigma_0^{-1} \mathbf{c}\mathbf{a}^\top \Sigma_0^{-1}$$

and

$$L_0 = \gamma_1^2 \beta_*^\top W_0^\top \Sigma_0^{-1} W_0 \beta_*$$

$$L_1 = \mathbf{u}^\top \mathbf{u}$$

$$L_2 = \mathbf{u}^\top \beta_*$$

$$L_3 = \gamma_1^2 \mathbf{a}^\top \Sigma_0^{-1} \mathbf{a}$$

$$L_4 = \gamma_1^2 \mathbf{c}^\top \Sigma_0^{-1} \mathbf{c}$$

$$L_5 = \gamma_1^2 \mathbf{c}^\top \Sigma_0^{-1} W_0 \beta_*$$

$$L_6 = \gamma_1^2 \mathbf{a}^\top \Sigma_0^{-1} \mathbf{c}$$

$$L_7 = \mathbf{a}^\top \mathbf{c}$$

$$L_8 = \gamma_1^2 \mathbf{a}^\top \Sigma_0^{-1} W_0 \beta_*$$

$$D = L_3(L_4 - L_1) - (1 + L_6)^2$$

Thus, we can rewrite the right hand side of Equation 7 as follows.

$$
\begin{aligned}
\tau_{d,\eta} = {} & 1 - \gamma_1^2 \beta_*^\top (W_0 + \mathbf{a}\mathbf{u}^\top)^\top \Sigma_0^{-1} (W_0 + \mathbf{a}\mathbf{u}^\top)\beta_* \\
& + \gamma_1^2 \beta_*^\top (W_0 + \mathbf{a}\mathbf{u}^\top)^\top \Delta_{aa}(W_0 + \mathbf{a}\mathbf{u}^\top)\beta_* \\
& + \gamma_1^2 \beta_*^\top (W_0 + \mathbf{a}\mathbf{u}^\top)^\top \Delta_{cc}(W_0 + \mathbf{a}\mathbf{u}^\top)\beta_* \\
& - \gamma_1^2 \beta_*^\top (W_0 + \mathbf{a}\mathbf{u}^\top)^\top \Delta_{ac}(W_0 + \mathbf{a}\mathbf{u}^\top)\beta_* \\
& - \gamma_1^2 \beta_*^\top (W_0 + \mathbf{a}\mathbf{u}^\top)^\top \Delta_{ca}(W_0 + \mathbf{a}\mathbf{u}^\top)\beta_* \\
= {} & 1 - L_0 - L_2^2 L_3 - 2L_2 L_8 \\
& + \frac{L_4 - L_1}{D}(L_2 L_3 + L_8)^2 \\
& + \frac{L_3}{D}(L_5 + L_2 L_6)^2 \\
& - 2\frac{1 + L_6}{D}(L_2 L_3 + L_8)(L_5 + L_2 L_6) \,.
\end{aligned}
$$

Similar to Proposition 29 in (Ba et al., 2022), by Hanson-Wright inequality, we have that $L_6, L_8, L_7 \to 0$.

$$L_0 \to \theta_2$$

$$L_1 \to \eta^2 \theta_4$$

$$L_2 = \eta \theta_3$$

$$L_3 \to \gamma_1^2 \underset{X \sim \mu_{\psi_1}}{\mathbb{E}} \left[ \frac{1}{\gamma_1^2 X + \gamma_2^2} \right] = \theta_1$$

$$L_4 \to \gamma_1^2 \eta^2 \theta_4 \cdot \psi_1 \underset{X \sim \mu_{\psi_1}}{\mathbb{E}} \left[ \frac{X}{\gamma_1^2 X + \gamma_2^2} \right] = \eta^2 \theta_2 \theta_4$$

$$L_5 \to \gamma_1^2 \eta \theta_3 \cdot \psi_1 \underset{X \sim \mu_{\psi_1}}{\mathbb{E}} \left[ \frac{X}{\gamma_1^2 X + \gamma_2^2} \right] = \eta \theta_2 \theta_3$$

$$L_6, L_7, L_8 \to 0$$

$$D \to L_3(L_4 - L_1) - 1 \to \eta^2 \theta_1 (\theta_2 - 1)\theta_4 - 1$$

To sum up, we have

$$\begin{aligned}
\lim_{d \to \infty} \tau_{d,\eta} &= 1 - \theta_2 - \frac{\eta^2 \theta_1 \theta_3^2 (\eta^2 \theta_1 (\theta_2 - 1)\theta_4 - 1)}{\eta^2 \theta_1 (\theta_2 - 1)\theta_4 - 1} \\
&\quad + \frac{\eta^4 \theta_1^2 (\theta_2 - 1)\theta_3^2 \theta_4}{\eta^2 \theta_1 (\theta_2 - 1)\theta_4 - 1} \\
&\quad + \frac{\theta_1 \theta_2^2 \theta_3^2}{\eta^2 \theta_1 (\theta_2 - 1)\theta_4 - 1} \\
&\quad - 2 \frac{\eta^2 \theta_1 \theta_2 \theta_3^2}{\eta^2 \theta_1 (\theta_2 - 1)\theta_4 - 1} \\
&= 1 - \theta_2 - \frac{\eta^2 \theta_1 (1 - \theta_2)^2 \theta_3^2}{1 + \eta^2 \theta_1 (1 - \theta_2)\theta_4} .
\end{aligned}$$

This completes the proof for the first part of Theorem 2.

**Step 5: Analysis for prediction accuracy.** Recall from Setting 1 the definition of prediction accuracy of the network after a gradient step is $\Pr_{(\mathbf{x},y) \sim \mathcal{D}_F(\beta_*)} [y \mathbf{a}^\top \sigma(W_1 \mathbf{x}) \geq 0]$. By Gaussian equivalence and Proposition 1, we have that the following.

$$\mathsf{Acc}_{P_{\text{train}}, d, N}(\psi_1, \psi_2, \eta)$$
$$= \Pr_{\substack{(\mathbf{x},y) \sim \mathcal{D}_F(\beta_*) \\ \mathbf{a}, W_1}} [y \mathbf{a}^\top \sigma(W_1 \mathbf{x}) \geq 0] .$$

By Proposition 1, we can further approximate the equation as follows.

$$= \Pr_{\substack{(\mathbf{x},y) \sim \mathcal{D}_F(\beta_*) \\ \mathbf{a}, W_0, \mathbf{u}}} [y \mathbf{a}^\top \sigma((W_0 + \mathbf{a}\mathbf{u}^\top)\mathbf{x}) \geq 0] + o(1) .$$

By Gaussian equivalence, we can further approximate the equation as follows.

$$= \Pr_{\substack{(\mathbf{x},y) \sim \mathcal{D}_F(\beta_*) \\ \mathbf{a}, W_0, W_*, \mathbf{u}}} [y \mathbf{a}^\top (\gamma_1(W_0 + \mathbf{a}\mathbf{u}^\top) + \gamma_* W_*)\mathbf{x}) \geq 0] + o(1)$$

where $W_* \in \mathbb{R}^{N \times d}$ and $([W_*]_{kj} \sim \mathcal{N}(0, 1/N))$ for each $k \in [N], j \in [d]$. Note that as $\mathbf{a}, W_0, W_*$ are independent, we can further simplify the equation as follows.

$$= \Pr_{\substack{(\mathbf{x},y) \sim \mathcal{D}_F(\beta_*) \\ \mathbf{a}, W_*', \mathbf{u}}} [y \gamma_1 \mathbf{u}^\top \mathbf{x} + \sqrt{\gamma_1^2 + \gamma_*^2} \cdot y \mathbf{a}^\top W_*' \mathbf{x} + o(1) \geq 0] + o(1)$$

where $W'_* \in \mathbb{R}^{N \times d}$ and $([W'_*]_{kj} \sim \mathcal{N}(0, 1/N))$ for each $k \in [N], j \in [d]$. Note that as $\mathbf{a}, W'_*$ are independent, we can further simplify the equation as follows.

$$
= \Pr_{\substack{(\mathbf{x},y) \sim \mathcal{D}_F(\beta_*) \\ Z \sim \mathcal{N}(0,1)}} \left[ \eta \gamma_1^2 \mathbb{E}_{(\mathbf{x}',y') \sim \mathcal{D}_F(\beta_*)} [yy' \mathbf{x}'^\top \mathbf{x}] + \sqrt{\gamma_1^2 + \gamma_*^2} \cdot Z + o(1) \geq 0 \right] + o(1) .
$$

Note that by decomposing $\mathbf{x}$ and $\mathbf{x}'$ to direction that's parallel to $\beta_*$ and orthogonal to $\beta_*$, we can further simplify the equation as follows.

$$
= \Pr_{\substack{(G,Y) \sim \mathcal{D}_F \\ Z,Z' \sim \mathcal{N}(0,1)}} \left[ \eta \gamma_1^2 \left( \mathbb{E}_{(G',Y') \sim \mathcal{D}_F} [YY'GG'] + \sqrt{1/\psi_2} Z' \right) + \sqrt{\gamma_1^2 + \gamma_*^2} \cdot Z + o(1) \geq 0 \right] + o(1)
$$

$$
= \Pr_{\substack{(G,Y) \sim \mathcal{D}_F \\ Z \sim \mathcal{N}(0,1)}} \left[ \eta \gamma_1^2 \theta_3 YG + \sqrt{\frac{\eta^2 \gamma_1^4}{\psi_2} + \gamma_1^2 + \gamma_*^2} \cdot Z + o(1) \geq 0 \right] + o(1)
$$

$$
= \mathbb{E}_{(G,Y) \sim \mathcal{D}_F} \left[ \Phi \left( \frac{\eta \gamma_1^2 \theta_3}{\sqrt{\frac{\eta^2 \gamma_1^4}{\psi_2} + \gamma_1^2 + \gamma_*^2}} YG \right) \right] + o(1) .
$$

Note that when fixing $\psi_1, \psi_2$ and non-trivial $F$, both capacity formula and prediction accuracy formula are increasing and invertible with respect to $\eta$. As a consequence, the two quantities are also analytically connected by an increasing and invertible function. This completes the proof for the second part of Theorem 2. We also provide numeric checks for the formulas in Figure 12.

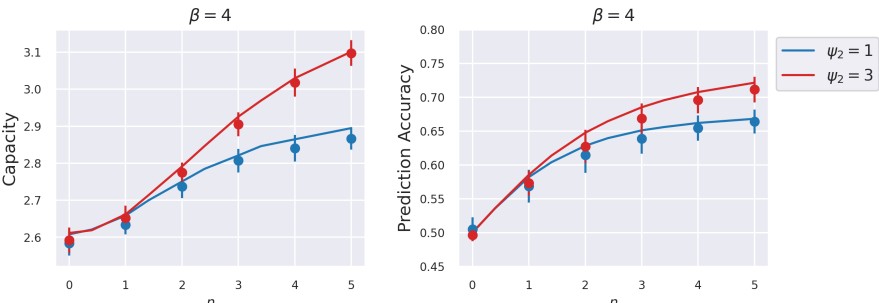

Figure 12: Numerical checks for the formulas in Theorem 2. We run the simulation with $d = 2000$, $\psi_1 = 1$, ReLU activation, and label function $f(x) = \frac{1}{1 + e^{-4x}}$ for 50 repetitions. Left: numerical checks for the capacity formula. Right: numerical checks for the prediction accuracy formula.

# D  2-LAYER NON-LINEAR NEURAL NETWORKS

In this paper, we use 2-layer non-linear neural networks and Gaussian mixture models (for input data generation) as a convenient experimental setup to systematically explore different regimes in feature learning. Moreover, given its medium level of complexity, it might be possible to have an analytical characterization of our numerical findings, and we leave it as an interesting future direction.

## D.1  EXPERIMENTAL SETUP

### D.1.1  SYNTHETIC DATA GENERATION

We focus on point manifold, which consists of data points associated with the same label. As discussed in the previous section, we are particularly interested in the effective radius, dimension, center alignment, axes alignment, and center-axes alignment of the representation manifolds. Therefore, we consider a synthetic model to generate training and test data with relevant geometric interpretations. Namely, construct $P \in \mathbb{N}$ synthetic data manifolds with radius $R \in \mathbb{R}_+$, intrinsic dimension

$D \in \mathbb{N}$, size $M \in \mathbb{N}$. The manifold layouts are further determined by center correlation strength $\rho_C \in [0, 1)$, axes correlation strength $\rho_A \in [0, 1)$, and center-axes correlation strength $\psi \in [0, 1)$, all of which we would detail in the following subsections.

**Isotropic spherical manifolds.** First, we consider the simplest case: manifolds with isotropic Gaussian center distribution and axes distribution with no correlations. This is the scenario considered in Section 3 and Section 4.

Let $d \in \mathbb{N}$ be the dimension of the data. We consider $P$ point manifolds $\{\mathcal{M}_i\}_{i \in [P]}$ with manifold size $M \in \mathbb{N}$ and radius $R$ that lies in a subspace of dimension $D$. Each manifold is defined as

$$\mathcal{M}_i = \{\mathbf{u}_0 + R \cdot \sum_{j=1}^{D} s_j^k \mathbf{u}_j + \epsilon \mathbf{v}_k\}_{k \in [M]}$$

where the axes $\mathbf{u}_j \sim N(0, I_d/d)$, the coordinates $s_j^k \sim N(0, 1)$, the noise vectors $\mathbf{v}_k \sim N(0, I_d/d)$, and $\epsilon = 10^{-2}$. The pre-scaled points in the manifolds $\{\sum_{j=1}^{D} s_j^k u_j\}_{k \in [M]}$ are well-normalized to unit norm.

Test manifolds share the same model except that the noise vectors $\mathbf{v}_j$ are sampled again in the same distribution.

**Isotropic Gaussian manifolds.** In certain experiments, we drop in the intrinsic dimension $D$ and directly consider manifolds defined as

$$\mathcal{M}_i = \{\mathbf{u}_0 + R \cdot \mathbf{v}_k\}_{k \in [M]}$$

where the noise vectors are $\mathbf{v}_k \sim N(0, I_d/d)$. Test manifolds share the same model except that the noise vectors $\mathbf{v}_k$ are sampled again in the same distribution.

**Correlated spherical manifolds.** To generated correlated manifolds, we consider an autoregressive model described by the covariance matrix $C = (\rho^{|i-j|})_{ij} \in \mathbb{R}^{P \times P}$, where $\rho \in [0, 1)$ is either the center correlation strength $\rho_C$ or axes correlation strength $\rho_A$. The center covariance $C_C$ is then mixed into the isotropic manifold centers $\{\mathbf{u}_0^j \sim N(0, I_d/d)\}_{j \in [M]}$. The axes covariance matrices $C_A^i$ is mixed into the isotropic axes $\{\mathbf{u}_i^j \sim N(0, I_d/d)\}_{j \in [M]}$ for each $i = 1, 2, \ldots, D$ respectively. The mixing is performed through multiplying the column matrix $M_C$ or $M_A^j \in \mathbb{R}^{P \times d}$ of centers or each axes with the Cholesky decomposition of $C_C$ or $C_A^i$. To incorporate center-axes correlation, we scale each center vector $\mathbf{u}_0$ by a factor of $(1 + \psi \cdot q)$ where $q \sim N(0, 1)$.

**Labels.** For $P$ manifolds with manifold size $M$, the $P$ labels are randomly sampled from a uniform distribution on $\{\pm 1\}$. Each label is associated with $M$ data points in the individual manifold. When learning with binary cross entropy, the labels are reassigned as $\{0, 1\}$ during loss and gradient computation.

### D.1.2 2-LAYER NEURAL NETWORK ARCHITECTURE

The model architecture we consider is similar to the architecture mentioned in Appendix C.

Let $d \in \mathbb{N}$ be the input data dimension, $N \in \mathbb{N}$ be the number of hidden neurons, $K \in \mathbb{N}$ be the number of linear readouts, $\alpha \in \mathbb{R}_+$ be the scaling factor of the readout weights.

Let $W = W_0 \in \mathbb{R}^{N \times d}$ be the initial weight matrix of a fully connected 2-layer neural network. Let $\{a_0^i\}_{i \in [K]}$ be a list of initial readout weights where $a_0^i \in \mathbb{R}^N$. Let $\sigma(\cdot) : \mathbb{R} \to \mathbb{R}$ be a non-linear activation function, e.g. ReLU or $\tanh$.

The feature of an input vector is defined as $\phi(\mathbf{x}) = \sigma(W\mathbf{x})$. The 2-layer neural network parameterized by $W$ and $a^i$ is defined as

$$f(W, a^i; \mathbf{x}) = \frac{\alpha}{\sqrt{N}} \mathbf{a}^\top \phi(\mathbf{x})$$

where the label prediction for data point $\mathbf{x}$ is $\mathsf{sgn}(f(\mathbf{x}))$ when learning with the mean squared error loss function. When learning with binary cross entropy loss function, we use $\{0, 1\}$ as labels and $\varsigma(f(\mathbf{x}))$ as prediction instead, where $\varsigma$ is the standard sigmoid function.

### D.1.3 LEARNING RULE

**Loss function and gradient update.** Let $\eta \in \mathbb{R}_+$ be the learning rate of the weight matrix, $c \in \mathbb{R}_+$ be the scaling factor of the readout learning rate, and let $\{(\mathbf{x}_i, y_i)\}_{i \in [PM]}$ be the collection of training data, where $P$ is the number of manifolds and $M$ is the manifold size.

We consider gradient descent over the loss function

$$\mathcal{L}(f) = \frac{1}{\alpha^2} \frac{1}{PM} \sum_{i \in [PM]} \ell(f(\mathbf{x}_i), y_i)$$

where $\ell : \mathbb{R} \times \{\pm 1\} \to \mathbb{R}$ is either the mean squared error (MSE)

$$\ell_{MSE}(z, y) = \frac{1}{2}(z - y)^2$$

or $l : \mathbb{R} \times \{0, 1\} \to \mathbb{R}$ is the binary cross entropy (BCE)

$$\ell_{BCE}(z, y) = y \cdot \log(1 + e^{-z}) + (1 - y) \cdot \log(1 + e^z)$$

**Mean squared error.** For the weight matrix, the gradient update with learning rate $\eta > 0$ is $W_{t+1} = W_t + \eta G_t$ where

$$G_t = \frac{1}{\alpha^2} \frac{1}{PM} \sum_{i \in [PM]} \frac{1}{K} \sum_{j \in [K]} \left[ \left( y_i - \frac{\alpha}{\sqrt{N}} \mathbf{a}_t^{j\top} \sigma(W_t \mathbf{x}_i) \right) \frac{\alpha}{\sqrt{N}} \mathbf{a}_t^j \odot \sigma'(W_t \mathbf{x}_i) \right] \mathbf{x}_i^\top$$

and $\sigma'(\cdot)$ denotes the first order derivative of $\sigma(\cdot)$. For each linear readout, the gradient update is $a_{t+1} = a_t + c\eta g_t$ where

$$g_t = \frac{1}{\alpha^2} \frac{1}{PM} \sum_{i \in [PM]} \left[ y_i - \frac{\alpha}{\sqrt{N}} \mathbf{a}_t^\top \sigma(W_t \mathbf{x}_i) \right] \frac{\alpha}{\sqrt{N}} \sigma(W_t \mathbf{x}_i)$$

Note that the $\alpha^{-2}$ multiplier on the loss function to ensure common convergence time when $\alpha \to \infty$ as mentioned in (Geiger et al., 2020).

**Binary cross entropy.** For the weight matrix, the gradient update with learning rate $\eta > 0$ is $W_{t+1} = W_t + \eta G_t$ where

$$G_t = \frac{1}{\alpha^2} \frac{1}{PM} \sum_{i \in [PM]} \frac{1}{K} \sum_{j \in [K]} \left[ \left( y_i - \varsigma\left[\frac{\alpha}{\sqrt{N}} \mathbf{a}_t^{j\top} \sigma(W_t \mathbf{x}_i)\right] \right) \frac{\alpha}{\sqrt{N}} \mathbf{a}_t^j \odot \sigma'(W_t \mathbf{x}_i) \right] \mathbf{x}_i^\top$$

where $\varsigma$ denotes the standard sigmoid function and $\sigma$ denotes the activation function. For each linear readout, the gradient update is $a_{t+1} = a_t + c\eta g_t$ where

$$g_t = \frac{1}{\alpha^2} \frac{1}{PM} \sum_{i \in [PM]} \left[ y_i - \varsigma\left[\frac{\alpha}{\sqrt{N}} \mathbf{a}_t^\top \sigma(W_t \mathbf{x}_i)\right] \right] \frac{\alpha}{\sqrt{N}} \sigma(W_t \mathbf{x}_i)$$

If not otherwise noted, we conduct experiments with the MSE loss function and ReLU activation function by default.

**A Note on Learning rate.** We define $\bar{\eta} = \eta \alpha^{-1}$ as the normalized effective learning rate. During training, We implicitly scale the learning rate $\eta$ by a factor of $\sqrt{N}$ in the experiments to enter the rich regime as mentioned in (Ba et al., 2022).

### D.1.4 TRAINING

For each 2-layer neural network experiment conducted in the paper, forty random seeds are chosen from 0 to 39000 with an interval of 1000 to train forty models in parallel for $10^5$ epochs. All training are conducted on the Flatiron Institute high performance computing clusters.

### D.1.5 FEATURE EXTRACTION

During analysis, fifty epochs are sampled uniformly in log-scale. For each model at checkpoint epoch $t$, we extract total $P$ size $M$ manifold representations $\{\Phi_t(\mathbf{x}_i)\}_{i \in [PM]}$ associated with labels $\{y_i\}_{i \in [PM]}$. We perform conventional analysis and manifold capacity analysis described in Appendix A and Appendix B respectively. We will present more details in the following experiment sections.

### D.2 CAPACITY IS A ROBUST MEASURE OF FEATURE LEARNING ACROSS ARCHITECTURE, DATA, AND LEARNING RULE VARIATIONS

The purpose of this section is to support Section 3 by showcasing that capacity is able to quantify feature learning even when model architecture, data distribution, and learning rule varies.

### D.2.1 FEATURE ANALYSIS METHODS

Here, we briefly present the conventional feature analysis methods and capacity analysis method and how they are computed in the experimental setup.

**Representation level analysis.** Activation stability is a representation level metric that intuitively captures how much neurons are activated in hidden units. Formally, we define it as

$$\frac{\sum_{i=1}^{PM} \sum_{j=1}^{N} \mathbf{1}_{>0}(\phi_j(\mathbf{x}_i))}{PMN}$$

Another conventional method to disentangle feature learning at representation level is tracking the norm of deviation from initial weights (Jacot et al., 2020)

$$\frac{\|W_t - W_0\|}{\|W_0\|}$$

On the other hand, the cosine similarity (Liu et al., 2024) can be used to study alignment at representation level

$$\frac{\Phi_t \Phi_0}{\|\Phi_t\|\|\Phi_0\|}$$

where $(\Phi_t)_{ij} = \phi_t(\mathbf{x}_i) \cdot \phi_t(\mathbf{x}_j) \in \mathbb{R}^{PM \times PM}$ is the gram matrix of features over the test data.

**Kernel methods.** The kernel methods for quantifying feature learning involves computing the Neural Tangent Kernel (NTK) (Jacot et al., 2020) for each pair of test data points:

$$\Theta_t(\mathbf{x}_1, \mathbf{x}_2) = \nabla_{w_t} f(\mathbf{x}_1) \cdot \nabla_w f(\mathbf{x}_2)$$

where $\nabla_{w_t} f$ denotes the total gradient of the neural network at epoch $t$ with respect to the hidden weights $W_t$ and readout weights $\{a_t^j\}$. Note that we scale the readout contribution to the total gradient by the readout learning rate factor $c \in \mathbb{R}_+$ aforementioned. Hence,

$$\nabla_w f(\mathbf{x}) = \nabla_{W_t} f(\mathbf{x}) + \frac{1}{K} \sum_{j=1}^{K} \nabla_{a_t^j} f(\mathbf{x})$$

After obtaining the gram matrix $\Theta_t = \Theta_t(\mathbf{x}_i, \mathbf{x}_j)_{ij} \in \mathbb{R}^{PM \times PM}$ from the test data, we can compute the *NTK change* defined as

$$\frac{\|\Theta_t - \Theta_0\|}{\|\Theta_0\|}$$

which can be interpreted as the relative deviation of the the kernel from initialization in the Frobenius norm metric. Conventionally studied, NTK change disentangles lazy and feature learning, as detailed in (Jacot et al., 2020). We present NTK change in Section 3 Figure 3 to compare it with capacity as the metric to track feature learning.

The *kernel alignment* can be similarly defined as the cosine similarity of initial and current NTK gram matrices:

$$\frac{\Theta_t \Theta_0}{\|\Theta_t\|\|\Theta_0\|}$$

which can be interpreted as the relative deviation of the kernel from initialization in terms of alignment. Kernel alignment is also studied in (Liu et al., 2024) to disentangle lazy and feature learning.

The *centered kernel alignment* (Kornblith et al., 2019) is another approximation method to study kernel evolution when the gram matrices is large:

$$\frac{HSIC(\Theta_t, \Theta_0)}{\sqrt{HSIC(\Theta_t, \Theta_t)HSIC(\Theta_0, \Theta_0)}}$$

where

$$HSIC = \frac{\text{Tr}(\Theta_t L \Theta_0 L)}{(n-1)^2}$$

These kernel metrics can be readily computed from the trained models and extracted features.

**Capacity and effective geometry.** For more details on data-driven manifold capacity analysis, please refer to Appendix B.

### D.2.2 EXTENDED DISCUSSION OF FIGURE 2A

**Setup.** In Figure 3a, we showcase that the degree of feature learning is controlled by the effective learning rate $\bar{\eta}$ with the following standard setup:

- Data: Isotropic Gaussian manifolds with $R = 0.5, M = 15$.

- Model: We set $\sigma = \text{ReLU}, N = 1500, d = 1000, P = 100, K = 1$.

- Learning rule: We set $\ell = \ell_{MSE}, \eta = 50, c = 0$ and

$$\alpha = 10/128, 10/112, 10/96, 10/80, 10/64, 10/16, 10/4, 10/1$$

so that the normalized effective learning rates are

$$\bar{\eta} = 128, 112, 96, 80, 64, 16, 4, 1$$

which is computed by $\bar{\eta} = \frac{\eta\alpha^{-1}}{5}$ where the division by 5 normalizes the smallest $\eta\alpha^{-1}$ to be 1.

- Training: We trained the models for 100000 epochs with 40 repetitions per parameter combination.

- Plotting: We use sample mean and 95% confidence interval for each data point.

**Result.** In Figure 13, we present the full range of conventional feature learning metrics, capacity, and effective manifold geometric measures of the same experiment presented in Figure 3a.

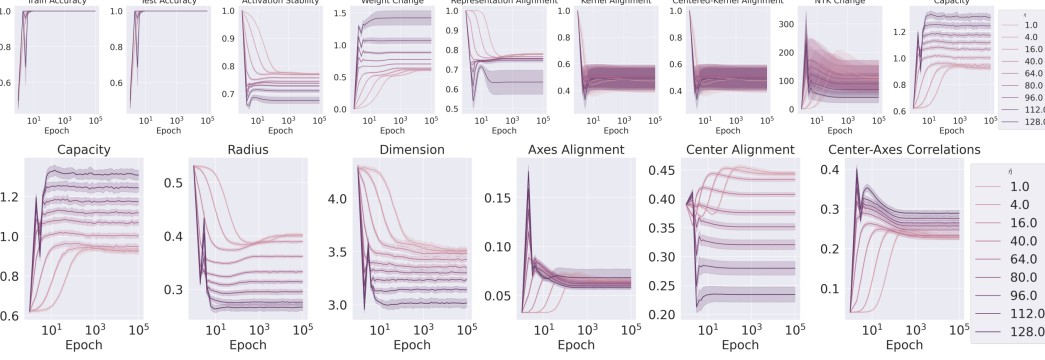

Figure 13: Conventional feature learning metrics, capacity, and effective geometric measures in a 2-layer neural network with same setup as in Figure 3a

### D.2.3 PROPORTIONAL LIMIT OF EXTENSIVE QUANTITIES

Finally, we study the degree of feature learning at the proportional limit of the 2-layer neural network in the sense that $N, P, d \to \infty$ where $P/N \to \psi_1, d/N \to \psi_2$. Here, we consider $\psi_1 = 1/15$ and $\psi_2 = 2/3$.

**Setup.** For each experiment, we only change $N, P, d$ together while keeping $P/N = 1/15, d/N = 2/3$. We keep the other parameters fixed:

- Data: Isotropic Gaussian manifolds with $R = 0.5, M = 15$.
- Model: We set $\sigma = \text{ReLU}, N = 300s, d = 200s, P = 20s, K = 1$ where the scaling factor $s = 1, 2, 3, 4, 5$.
- Learning rule: We set $\ell = \ell_{MSE}, \eta = 10, \alpha = 1, c = 0$ so that the normalized effective learning rates is $\bar{\eta} = 1$ which is computed by $\bar{\eta} = \frac{\eta \alpha^{-1}}{10}$ where the division by 10 normalizes the smallest $\eta \alpha^{-1}$ to be 1.
- Training: We trained the models for 100000 epochs with 40 repetitions per parameter combination.
- Plotting: We use sample mean and standard deviation for each data point.

**Result.** In Figure 14, we see that capacity is able to track the degree of feature learning at the proportional limit as $N, P, d$ scales up from $300, 20, 200$ to $1500, 100, 1000$. In particular, capacity and geometric measures saturate as we scale the extensive quantities.

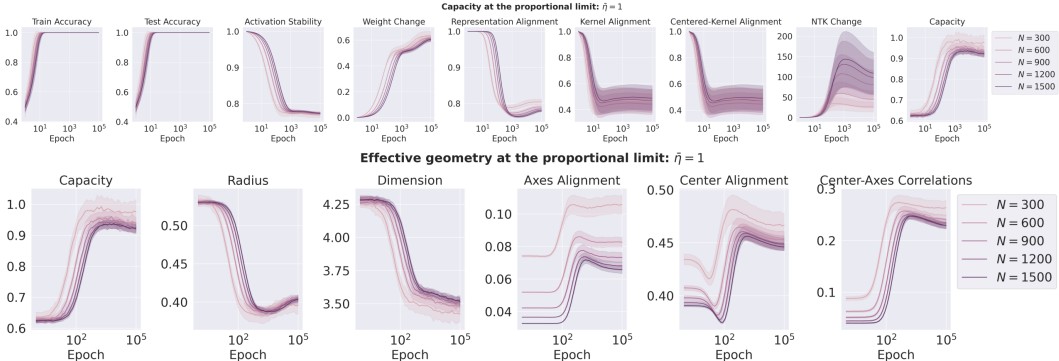

Figure 14: Conventional feature learning metrics, capacity, and effective geometric measures for different scaling factors $s = 1, 2, 3, 4, 5$ on $N, P, d$ in a 2-layer neural network.

### D.2.4 DIFFERENT ACTIVATION FUNCTIONS

In Figure 3a, we showed that capacity quantifies the degree of feature learning for the ReLU activation function. In this section, we show that capacity is a robust measure for different activation functions.

**Setup.** We consider a standard setup with the $\tanh$ activation replacing the ReLU activation:

- Data: Isotropic Gaussian manifolds with $R = 0.5, M = 15$.
- Model: We set $\sigma = \tanh, N = 300, d = 200, P = 20, K = 1$.
- Learning rule: We set $\ell = \ell_{MSE}, \eta = 100, c = 0$ and
$$\alpha = 10/16, 10/14, 10/12, 10/10, 10/8, 10/5, 10/4, 10/3, 10/2, 10/1$$
so that the normalized effective learning rates are
$$\bar{\eta} = 16, 14, 12, 10, 8, 5, 4, 3, 2, 1$$
which are computed by $\bar{\eta} = \frac{\eta \alpha^{-1}}{10}$ where the division by 10 normalizes the smallest $\eta \alpha^{-1}$ to be 1.

- Training: We trained the models for 100000 epochs with 40 repetitions per parameter combination.

- Plotting: We use sample mean and standard deviation for each data point.

**Result.** In Figure 15, we see that capacity is also able to track the degree of feature learning for $\tanh$ activation function.

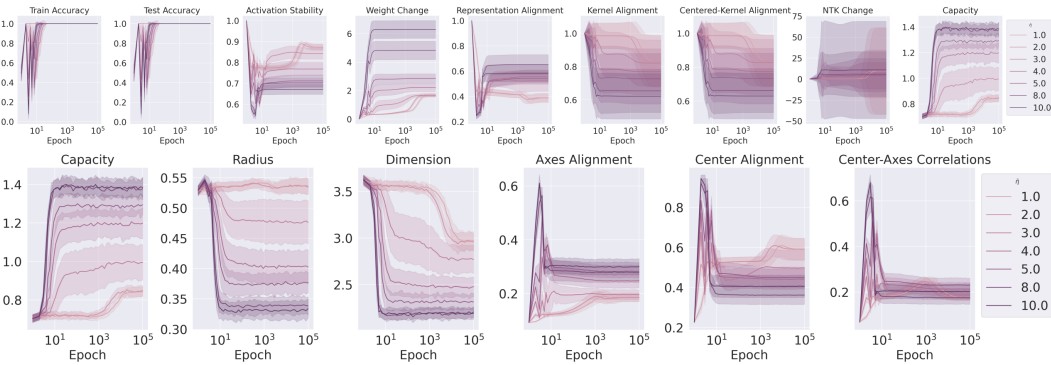

Figure 15: Conventional feature learning metrics, capacity, and effective geometric measures in a 2-layer neural network with $\tanh$ activation function.

### D.2.5 DIFFERENT DATA CORRELATIONS AND TASK DIFFICULTIES

In this section, we study the effect of data and task variations on the degree of feature learning and effective geometry. In particular, we are interested in data correlations parameterized by axes, center, and center-axes correlation strengths $\rho_A, \rho_C, \psi$; task difficulty parameterized by the number of linear readouts $K$, radius of data manifolds $R$, number of manifolds $P$, and the dimension of data $d$.

**Setup.** For each experiment, we only change one of the interested parameters (denoted in the legends of the plots) and keep the following set of default parameters fixed:

- Data: Correlated spherical manifolds with $R = 0.5, D = 8, M = 15, \rho_A = 0, \rho_C = 0, \psi = 0$.

- Model: We set $\sigma = \text{ReLU}, N = 300, d = 200, P = 20, K = 1$.

- Learning rule: we set $\ell = \ell_{MSE}, \alpha = 1, c = 0, \eta = 10$ so that the normalized effective learning rate is $\bar{\eta} = \frac{\eta \alpha^{-1}}{10}$ where the division normalizes the smallest $\eta \alpha^{-1}$ to be 1.

- Training: We trained the models for 100000 epochs with 40 repetitions per parameter combination.

- Plotting: We use sample mean and standard deviation for each data point.

**Result.** In Figure 16, we compare conventional measures of feature learning with capacity when varying one of the data or task parameter. We see that capacity is able to consistently reflect the degree of feature learning when there is data or task variations.

In Figure 17, we showcase the effective geometric measures of the same experiments. In particular, we note that data correlations are well-captured and disentangled by relevant effective alignment measures while not necessarily captured by conventional methods such as kernel or representation alignment. Meanwhile, we see that as number of tasks $K$ and data manifold radius $R$ increases, capacity and geometric measures saturate. Finally, we see that $P/N$ and $d/N$ ratios affects the learning strategy. For higher $P/N$ ratio, capacity increment is driven by lower effective dimension while effective radius increases. For higher $d/N$ ratio, capacity increment is driven by lower effective radius instead while effective dimension increases.

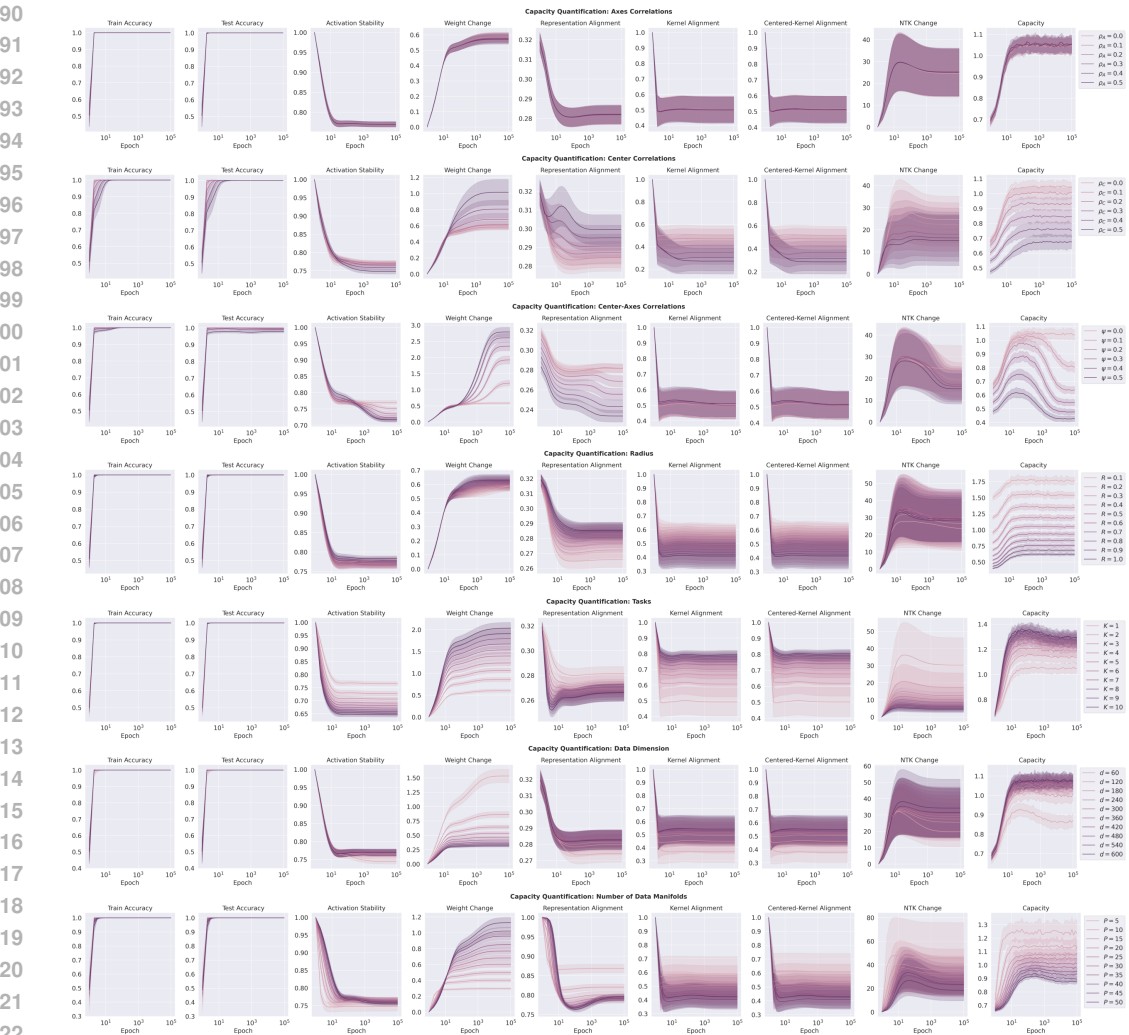

Figure 16: Capacity and conventional measures of feature learning for different data correlations and task difficulties in a 2-layer neural network.

### D.3 EFFECTIVE GEOMETRY REVEALS DISTINCT LEARNING DYNAMICS

#### D.3.1 LEARNING STRATEGIES

**Compression strategy setup** In Figure 4b where the networks performs the compression strategy, we use a difficult-task setup with higher data manifold radius and more readout tasks:

- Data: Isotropic spherical manifolds with $R = 1.0, D = 8, M = 15$.
- Model: We set $\sigma = \text{ReLU}, N = 300, d = 200, P = 20, K = 27$.
- Learning rule: we set $\ell = \ell_{MSE}, \alpha = 1, c = 0$ and

$$\eta = 1, 5, 10, 20, 30, 40, 50, 60, 70, 80, 90, 100, 110, 120, 130, 140, 150$$

so that the normalized effective learning rates are

$$\bar{\eta} = 1, 5, 10, 20, 30, 40, 50, 60, 70, 80, 90, 100, 110, 120, 130, 140, 150.$$

- Training: We trained the models for 100000 epochs with 40 repetitions per parameter combination.
- Plotting: We use sample mean for each data point.

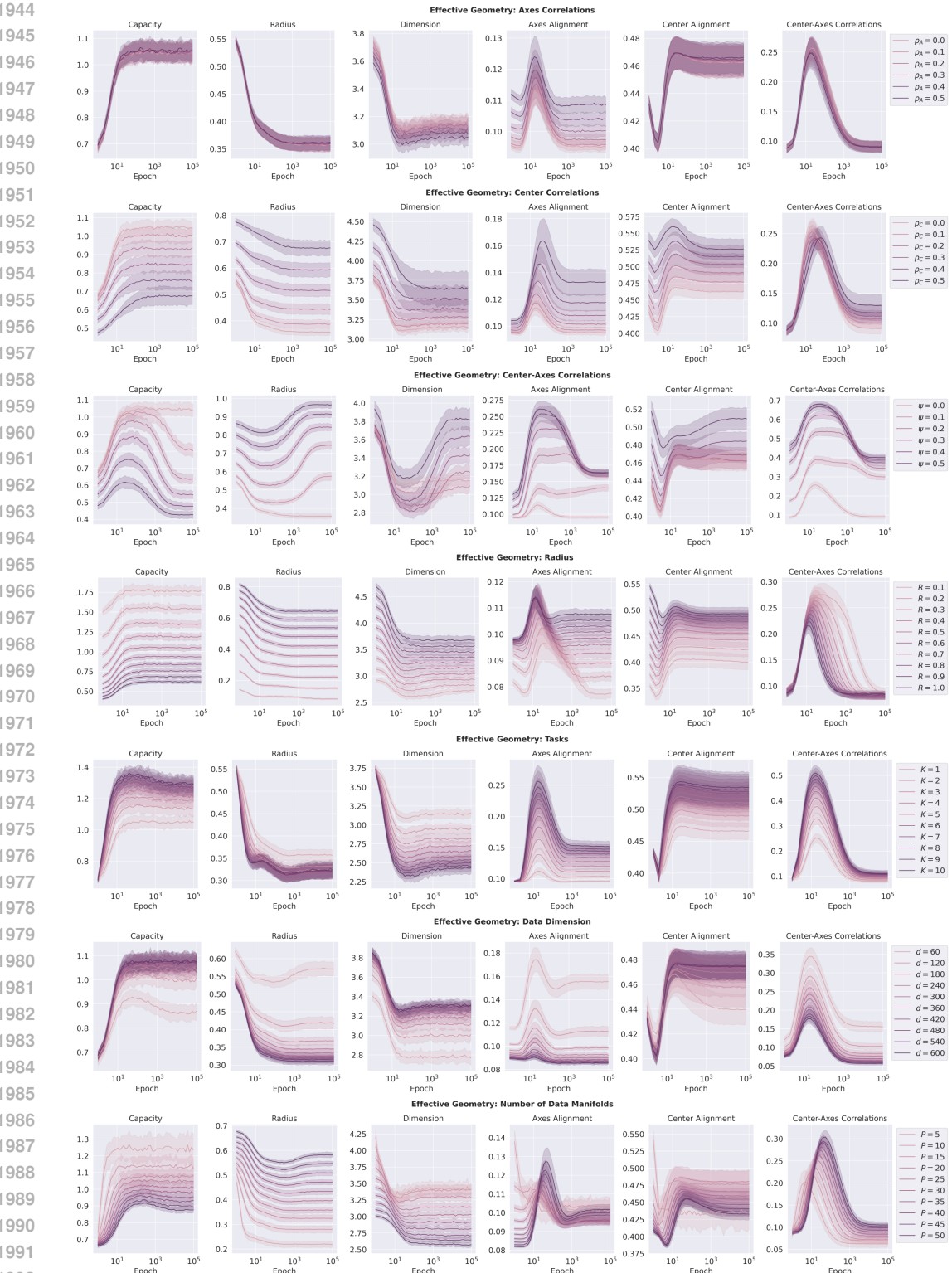

Figure 17: Capacity and effective geometric measures for different data correlations and task difficulties in a 2-layer neural network.

**Flattening strategy setup.** In Figure 4b where the networks performs the flattening strategy, we use an easy-task setup with smaller data manifold radius and very few readout tasks:

- Data: Isotropic spherical manifolds with $R = 0.5, D = 8, M = 15$.

- Model: We set $\sigma = \text{ReLU}, N = 300, d = 200, P = 20, K = 3$.

- Learning rule: we set $\ell = \ell_{MSE}, \alpha = 1, c = 0$ and

$$\eta = 80, 90, 100, 110, 120, 130, 140, 150, 160, 170$$

  so that the normalized effective learning rates are

$$\bar{\eta} = 80, 90, 100, 110, 120, 130, 140, 150, 160, 170.$$

- Training: We trained the models for 100000 epochs with 40 repetitions per parameter combination.

- Plotting: We use sample mean for each data point.

**Contour plot of learning strategies.** In Figure 4b and c, we use contour plots to visualize the different learning strategies adopted by the network. We use Equation 34 in (Chung et al., 2018) to approximate capacity using effective radius and dimension:

$$\alpha = \frac{1 + \left(\frac{1}{R_M^2}\right)}{D_M}$$

The scatter points with the same color correspond to a model trained with the same normalized effective learning rate $\bar{\eta}$ over different epochs.

### D.3.2 LEARNING STAGES

**Setup.** In Figure 4a, we adopt a setup with moderate radius and number of readout tasks that shows clean learning stages:

- Data: Isotropic spherical manifolds with $R = 1, D = 8, M = 15$.

- Model: We set $\sigma = \text{ReLU}, N = 300, d = 200, P = 20, K = 5$.

- Learning rule: we set $\ell = \ell_{MSE}, \eta = 10, \alpha = 1, c = 0$ so that the normalized effective learning rate is $\bar{\eta} = 10$.

- Training: We trained the models for 100000 epochs with 40 repetitions per parameter combination.

- Plotting: We use sample mean for each data point.

**Hidden Markov Model and learning stages plot.** In Figure 4a, we use heat map to visualize the learning stages of a particular model. In each row, we present the values of capacity, effective radius, dimension, axes alignment, center alignment, and center-axes alignment normalized between $[0, 1]$. In each column, we present one epoch sampled uniformly in log-scale from the training progress. The three vertical dotted lines separates four learning stages learned by a Hidden Markov Model (HMM).

Inspired by (Hu et al., 2024), we fit a Hidden Markov Model (Baum & Petrie, 1966) with the features obtained from manifold capacity and effective geometry. For each epoch, we concatenate these mesoscopic variables into feature vectors $\{z_{1:T}\}$, where $T$ is the number of sampled epochs. We then apply z-score normalization to ensure each feature have a zero mean and unit variance, as HMMs are sensitive to the scale of features:

$$z_t = \begin{bmatrix} f_1(\mathbf{w}_t) \\ \vdots \\ f_d(\mathbf{w}_t) \end{bmatrix}, \qquad \tilde{z}_t = \begin{bmatrix} \frac{f_1(\mathbf{w}_t) - \mu(f_1(\mathbf{w}_{1:T}))}{\sigma(f_1(\mathbf{w}_{1:T}))} \\ \vdots \\ \frac{f_d(\mathbf{w}_t) - \mu(f_d(\mathbf{w}_{1:T}))}{\sigma(f_d(\mathbf{w}_{1:T}))} \end{bmatrix}$$

Note that each $f_i$ here represents the function that takes the weights of the model at epoch $t$ and outputs the mesoscopic variable we obtained from manifold capacity analysis.

We collect $\{\tilde{z}_{1:T}\}_1^{N_{rep}}$ for different random seeds and fit the HMM using the Baum-Welch algorithm (Baum et al., 1970) with the hyperparameter being the number the hidden states over the sampled epochs. We choose the HMM with the number of hidden states with highest Bayesian information criterion (BIC), as it provides simpler and more interpretable models. Finally, we use the predicted hidden states to assign learning stages to each epoch.

# E DEEP NEURAL NETWORKS

## E.1 EXPERIMENTAL SETUP

In this section, we provide detailed information about the experimental setup for deep neural networks, including model architectures, datasets, training procedure, and manifold capacity measurements.

### E.1.1 MODELS

We use the VGG-11 models (Simonyan & Zisserman, 2014) for experimental results in the main paper. We also repeat these experiments on ResNet-18 (He et al., 2016). The specific implementation follows a similar setting in (Chizat et al., 2019) and is adapted from `https://github.com/edouardoyallon/lazy-training-CNN`.

**Output rescaling .** As previously studied in (Chizat et al., 2019), multiplying the model outputs by a large scaling factor $\beta$ can induce lazy learning (we use the notation $\beta$ instead of $\alpha$ in (Chizat et al., 2019) to avoid confusion with the notation $\alpha$ as capacity in Equation 2 ). In this section, we use the inverse scaling factor $\beta^{-1}$ as the parameter to control the degree of feature learning. We define the *normalized effective learning rate* $\overline{\eta} = \beta^{-1}$. We also note several adjustments to the common training framework to adapt to using the inverse scaling factor $\beta^{-1}$ as the parameter to control the degree of feature learning.

- Rescaled loss function: To adjust for using the scaling factor $\beta$, we use the rescaled loss function $L_\beta = \frac{L}{\beta^2}$ with $L$ denotes the loss function to accommodate for the time parameterization of the loss dynamic for large $\beta$ as previously indicated in (Chizat et al., 2019) and (Geiger et al., 2020).

- Model's initial outputs as 0: As mentioned in (Chizat et al., 2019), for the scaling factor $\beta$ to be able to control the rate of feature learning, the model output as initialization $f(W_0)$ must be equal 0. To ensure this condition, we set $f(W_t) = h(W_t) - h(W_0)$ with $W_t$ be the model's weight at training step $t$, $h$ be the output of the network, and $f$ be the final adjusted network output.

**Number of repetitions.** All model measurements (train accuracy, test accuracy, activation stability, etc.) are reported as the mean of 5 independently trained model (with different random seeds). The error bar indicates the bootstraped 95% confidence interval calculated using `seaborn.lineplot(errorbar=('ci', 95))`.

### E.1.2 DATASET

In this section, we list detailed information about the dataset used in the paper.

**CIFAR-10.** The CIFAR-10 dataset (Krizhevsky & Hinton, 2009) consists of 60000 32x32 colour images in 10 classes, with 6000 images per class. There are 50000 training images and 10000 test images.

**CIFAR-100.** The CIFAR-100 dataset (Krizhevsky & Hinton, 2009) is similar to CIFAR-10, except that it has 100 classes containing 600 images each. There are 500 training images and 100 testing images per class. Note that the images in CIFAR-10 and CIFAR-100 are mutually exclusive.

**CIFAR-10C.** The CIFAR-10C dataset (Hendrycks & Dietterich, 2018) includes images from the CIFAR-10 evaluation set with common corruptions such as Gaussian noise, fog, motion blur, etc. The dataset has 15 different common corruption types, and 5 different severity levels for each corruption type.

### E.1.3 TRAINING PROCEDURE

- Loss function: We follow the theoretical results and practice used in (Chizat et al., 2019) to use mean-squared error loss to train all DNNs mentioned in the paper.
- Optimizer: We use Stochastic Gradient Descent with momentum (implemented as `torch.optim.SGD(momentum=0.9)`) to train the models.
- Data augmentation: We apply the following data augmentation during training: `RandomCrop(32, padding=4),RandomHorizontalFlip`.
- Learning rate and learning schedule: We follow the practice in (Chizat et al., 2019) and set initial learning rate $\eta_0 = 1.0$ for VGG-11 and $\eta_0 = 0.2$ for ResNet-18. The learning rate schedule is defined as $\eta_t = \frac{\eta_0}{1 + \frac{1}{3}t}$.
- Initialization: We follow the practice in (Chizat et al., 2019) to initialize the model's weight using Xavier initialization (Glorot & Bengio, 2010) and the bias to be 0.
- Batch size: We use batch size of 128 during training and batch size of 100 during evaluation.

### E.1.4 MANIFOLD CAPACITY MEASUREMENTS

In this section, we provide detailed information about how we define object manifolds from the model's representations and measure the manifold capacity and geometric properties (Chung et al., 2018).

- Features extraction: For each image, we extract the object representation from the last linear layer (dimension $512$) before the classification layer (dimension $10$).
- Number of manifolds: We use 10 object manifolds for each measurement.
- Number of points per manifold: For each object manifold, we randomly sample 50 images from the interested class.
- Number of repetitions: Every capacity and geometry measurement is repeated 10 times per model instance (50 times if we have 5 model repetitions) and we report the mean and the error bar as the bootstraped 95% confidence interval calculated using `seaborn.lineplot(errorbar=('ci', 95))`.

### E.2 CAPACITY QUANTIFIES THE DEGREE OF FEATURE LEARNING IN DEEP NEURAL NETWORKS

**Capacity and manifold geometry for VGG-11 models.** In Figure 3, we show manifold capacity along with other common metrics used to identify feature learning such as train accuracy, test accuracy, relative weight norm change, and activation stability. In this section, we provide other manifold geometric measurements along with manifold capacity in Figure 18.

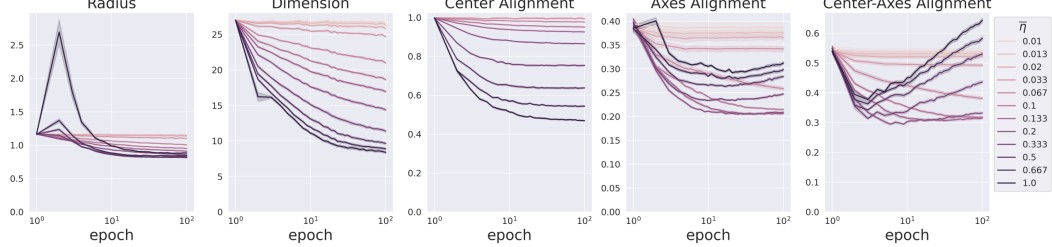

Figure 18: Manifold capacity and geometry for VGG-11 models trained with different $\overline{\eta}$

**Capacity quantifies the degree of feature learning in ResNet-18 models.** In section Section 3, we show that manifold capacity can capture the degree of feature learning in DNNs, specifically in VGG models. In this section, we empirically show this statement can also be extended to other model architectures, specifically ResNets, in Figure 19.

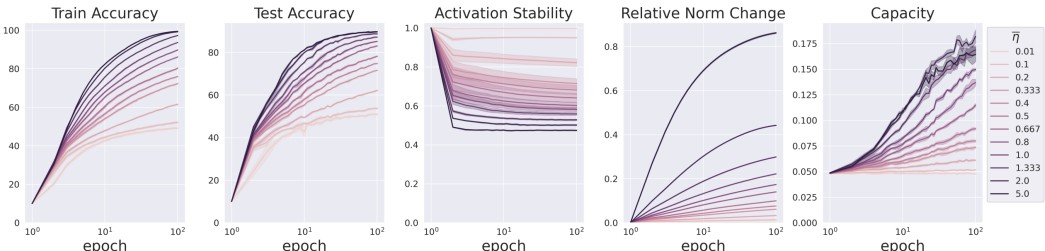

(a) Manifold capacity captures the degree of feature learning in ResNet-18

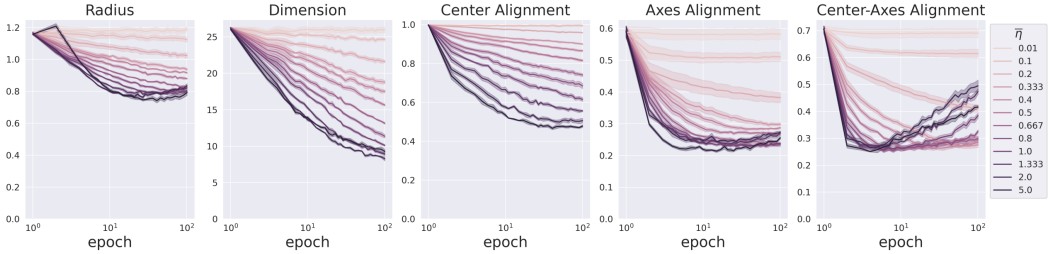

(b) Manifold geometry for ResNet-18 models trained with different $\overline{\eta}$

Figure 19: Manifold capacity and geometry of ResNet-18 models trained with different scale factor.

**Capacity quantifies the degree of feature learning in VGG-11 models trained with weight regularizer.** While most theoretical work in the lazy vs rich learning literature are formulated with vanilla mean squared error (MSE) loss (Jacot et al., 2020) (Chizat et al., 2019), in practice, MSE with weight regularizer (or weight decay) is used widely to prevent over-fitting and improve model generalization. In Figure 20, we explore the effect of weight decay to feature learning and demonstrate empirically that capacity can still quantify the degree of feature learning in models trained with L2-regularizer. We implemented L2-regularizer by setting `torch.optim.SGD(weight_decay=0.0002)`. We leave further study about the impact between the magnitude of weight regularizer and effective learning rate (and/or scaling factor) to the degree of feature learning as a potential future direction.

### E.3 MANIFOLD CAPACITY AND MANIFOLD GEOMETRY DELINEATE LEARNING STAGES IN DEEP NEURAL NETWORKS

In section Section 4.2, we have demonstrated the use of effective manifold geometry to uncover hidden learning stages in 2-layer neural networks. In this section, we showed that using similar technique, we can also discover geometric learning stages in deep neural networks as well.

**Experiment setup** We used similar setup mentioned in Section E.1. In this section, to give a higher resolution into the learning dynamic, we extracted the model checkpoint at each training step (after each training batch, with `batch_size=100`) instead of each training epoch (after a whole train dataset iteration).

### E.4 FEATURE LEARNING AND DOWNSTREAM TASK: OUT-OF-DISTRIBUTION GENERALIZATION

In this section, we measure the performance of the models trained with different degree of feature learning (quantified by effective learning rate $\overline{\eta}$) on the downstream tasks for OOD using CIFAR-100, a dataset with no overlap with CIFAR-10, the dataset used to train the model.

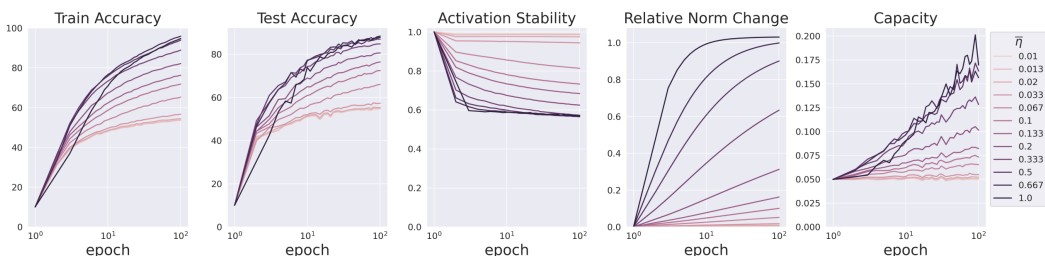

(a) Manifold capacity captures the degree of feature learning in VGG-11 models trained with L2-regularizer

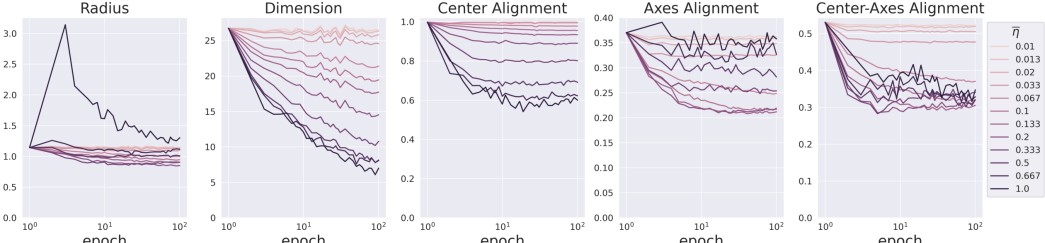

(b) Manifold geometry for VGG-11 models with L2-regularizer trained with different $\overline{\eta}$

Figure 20: Manifold capacity and geometry of VGG-11 models with L2-regularizer trained with different scale factor.

### E.4.1 EXPERIMENTAL SETUP

We use linear probe (Alain & Bengio, 2016) on representation from the last linear layer (dimension 512) to measure the performance of models trained on CIFAR-10 on the out-of-distribution dataset, CIFAR-100. Linear probes are linear classifiers trained on top of the representation to probe how much information the representations encode about a particular task or characteristic. This approach has been used widely in different fields including natural language processing (Belinkov et al., 2017) and computer vision (Raghu et al., 2021).

Here we provide detailed information about how we construct the linear probes.

**Optimizer.** We use *Adam* optimizer with initial learning rate $\eta_0 = 0.1$ and learning rate schedule is defined as $\eta_t = \frac{\eta_0}{1 + \frac{1}{3}t}$. Other parameters are default `Pytorch` parameters.

**Number of epochs.** The linear probe is trained for 50 epochs, unless it is stopped early, as described by the early stop method below.

**Early stop.** During training, if the validation loss is greater than the minimum validation loss so far for more than $N_{patience}$ epoch, then training is stopped. We set $N_{patience} = 3$.

### E.4.2 OOD PERFORMANCE FOR RESNET-18

In Section 5.2, we demonstrate how capacity and effective manifold geometry can be used to characterize the OOD performance of VGG-11 models trained with different effective learning rate $\overline{\eta}$. In this section, we show OOD performance and effective geometry of ResNet-18 models trained with different effective learning rate $\overline{\eta}$ in Figure 21. Interestingly, unlike VGG-11, for ResNet-18, the failure of models in the ultra-rich regime is characterized by the expansion of manifold dimension, not manifold radius.

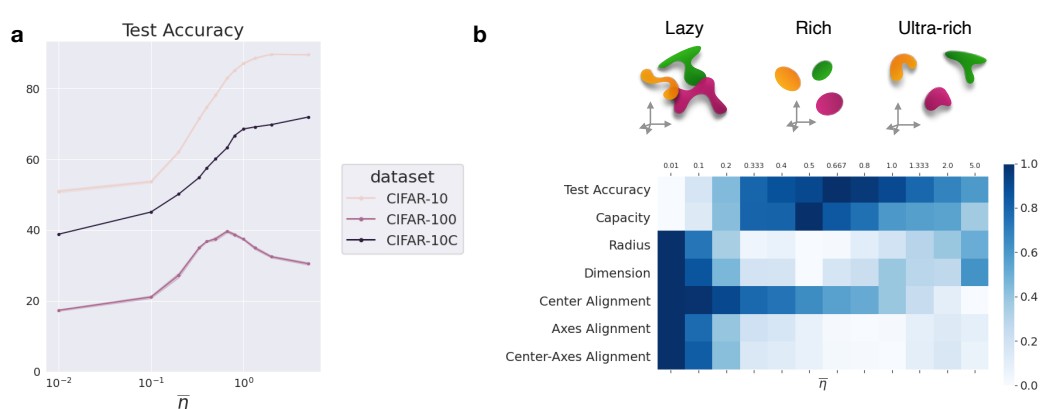

Figure 21: OOD performance and effective geometric measure of ResNet-18 models trained with different scale factor.

## F RECURRENT NEURAL NETWORKS

### F.1 EXPERIMENTAL SETUP

In this section, we provide detailed information about the experimental setup for recurrent neural network in 5.1, including model architectures, datasets, training procedure, and manifold capacity measurements.

#### F.1.1 DATASET

We used the package `neurogym` (Molano-Mazon et al., 2022) to simulate common cognitive tasks. In this paper, we trained recurrent neural networks to perform the following cognitive tasks: perceptual decision making, context decision making, and delay match sample. We followed the task configuration used in (Liu et al., 2024). We list detailed information of task configuration and descriptions below.

**Perceptual decision making (Britten et al., 1992) (documentation page)**

- Task description: In each trial, given two noisy stimulus, the agent needs to integrate the stimulus over time to determine which stimuli has stronger signal.

- Task configuration: We set up the task using the following parameters: {`timing:` {`fixation: 0, stimulus: 700, delay: 0, decision: 100`}, `dt: 100, seq_len: 8`}

**Context decision making (Mante et al., 2013) (documentation page)**

- Task description: In each trial, given two noisy stimulus, each has two modalities, the agent needs to integrate the stimulus in one specific modal while ignoring the other modal. The interested modal is given by the context.

- Task configuration: We set up the task using the following parameters: {`timing:` {`fixation: 0, stimulus: 200, delay: 500, decision: 100`}, `dt: 100, seq_len: 8`}

**Delay match sample (Miller et al., 1996) (documentation page)**

- Task description: In each trial, a sample stimulus is shown during the sample period, which followed by a delay period. Afterwards, the test stimulus is shown. The agent needs to determine whether the sample and the test stimuli are matched.

- Task configuration: We set up the task using the following parameters: `{timing: {fixation: 0, sample: 100, delay: 500, test: 100, decision: 100}, dt: 100, seq_len: 8}`

### F.1.2 MODELS

**Model architecture** We consider time-continuous recurrent neural networks (RNNs) architecture that are commonly used to model neural circuits (Liu et al., 2024; Ehrlich et al., 2021). Specifically, we consider RNNs with 1 hidden layer, ReLU activation, $N_{in}$ input units, $N_{hidden}$ hidden units, and $N_{out}$ output unit. Let $x_t \in \mathbb{R}^{N_{in}}$, $y_t \in \mathbb{R}^{N_{out}}$ be the corresponding input and output at time-step $t$. The model's hidden representation $h_t$ and outputs $\hat{y}_t$ at time step $t$ can be defined by the given equations:

$$h_{t+1} = \rho h_t + (1 - \rho)(W_h \sigma(h_t) + W_i x_t) \tag{8}$$

$$\hat{y}_t = W_o \sigma(h_t) \tag{9}$$

In the above equation, $W_i \in \mathbb{R}^{N_{in} \times N_{hidden}}$, $W_h \in \mathbb{R}^{N_{hidden} \times N_{hidden}}$, $W_o \in \mathbb{R}^{N_{hidden} \times N_{out}}$. $\sigma(.)$ is the non-linear activation function, in which we used ReLU, and $\rho$ is the decay factor which is defined by $\rho = e^{\frac{-dt}{\tau}}$ with time step $dt$ and time constant $\tau$. We use $N_{hidden} = 300$ for all RNNs models.

**Weight rank initialization** Following the practice in (Liu et al., 2024), we initialize the recurrence weight $W_h$ by initializing an initial full-ranked random Gaussian matrix, and then use Singular Value Decomposition to truncate the weight rank to the desired rank. The truncated weight matrix is then re-scaled to ensure that weight matrices with varying ranks have the same weight norm.

### F.1.3 TRAINING PROCEDURE

- Loss function: Since all three tasks that we consider are classification tasks, we use cross entropy loss.
- Optimizer: We use Stochastic Gradient Descent with momentum (implemented as `torch.optim.SGD(lr=0.003, momentum=0.9)`) to train the models.
- Batch size: We use batch size of 32 for each training step.

The models are trained for 10000 iterations and all models being compared achieved similar loss and accuracy after training (see Figure 22, 23, 24 for more details).

### F.1.4 MANIFOLD CAPACITY MEASUREMENTS

In this section, we provide detailed information about how we define object manifolds from the model's representations and measure the manifold capacity and geometric properties (Chung et al., 2018).

- Features extraction: We extract the representation $h_t$ (in Equation 8) from the hidden layer (dimension 300) with $t$ being the decision period of the trial.
- Number of manifolds: The number of possible choices in the decision period of all the three tasks that we consider is 2, so the number of manifolds are 2.
- Number of points per manifold: For each task-relevant manifold, we randomly sample 50 trials of the corresponding ground truth choices.
- Number of repetitions: Every capacity and geometry measurement is repeated 50 times and we report the mean and the error bar as the bootstraped 95% confidence interval calculated using `seaborn.lineplot(errorbar=('ci', 95))`.

### F.2 ADDITIONAL RESULTS ON OTHER COGNITIVE TASKS

In section 5.1, we present the results on how the initial structural connectivity bias (initialized by varying the rank of the weight matrix) affects the feature learning regime and representational geometry of a given model in the perceptual decision making task (also called the two-alternative forced choice task) (Britten et al., 1992). In this section, we show more detailed results (including

accuracy and loss) on the perceptual decision making task in Figure 22, along with two other cognitive tasks, which are context decision making task (Mante et al., 2013) in Figure 23 and delay match sample task (Miller et al., 1996) in Figure 24.

**Perceptual Decision Making Task**

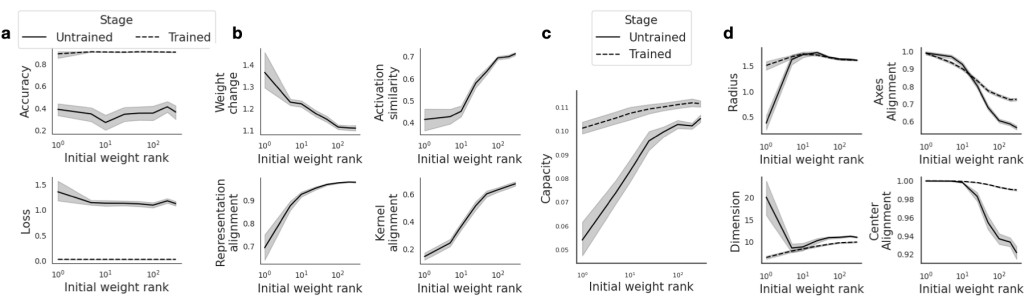

Figure 22: Structural connectivity bias in the two-alternative forced choice task. **a.** Model train and loss accuracy **b.** Weight change and alignment measurements **c.** Manifold capacity measurements **d.** Effective manifold geometry measurements.

**Context Decision Making Task**

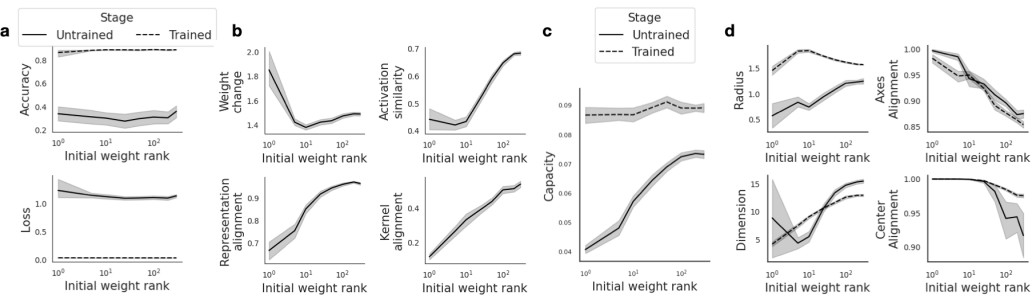

Figure 23: Structural connectivity bias in the context decision making task **a.** Model train and loss accuracy **b.** Weight change and alignment measurements **c.** Manifold capacity measurements **d.** Effective manifold geometry measurements.

**Delay Sample Matching Task**

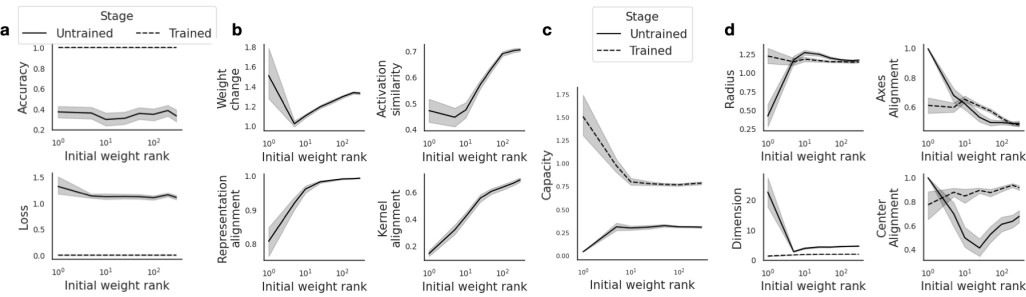

Figure 24: Structural connectivity bias in the delay mataching sample task. **a.** Model train and loss accuracy **b.** Weight change and alignment measurements **c.** Manifold capacity measurements **d.** Effective manifold geometry measurements.

