# OpenReview forum: "Beyond the Lazy versus Rich Dichotomy: Geometry Insights in Feature Learning from Task-Relevant Manifold Untangling"
_ICLR.cc/2025/Conference — Submitted to ICLR 2025_

### Official Review · Reviewer_D4ag · 2024-11-01

**Soundness:** 3
**Presentation:** 3
**Contribution:** 3
**Rating:** 6
**Confidence:** 3

**Summary:**

This paper proposes a novel framework for understanding neural network feature learning beyond the “lazy” versus “rich” dichotomy by focusing on task-relevant manifold untangling using representational geometry. The authors introduce manifold capacity as a key metric to quantify feature learning richness, and explore how manifold geometric measures reveal distinct learning stages and strategies. This framework is applied in contexts ranging from standard machine learning tasks to neuroscience, offering insights into structural inductive biases and challenges in out-of-distribution (OOD) generalization.

**Strengths:**

1. The authors present both theoretical analysis and empirical evidence demonstrating that manifold capacity effectively quantifies the degree of feature learning. Comparisons with other metrics, such as accuracy and weight changes, further underscore the efficacy of manifold capacity in this context.
2. The authors offer insightful empirical findings on the relationship between manifold geometry and learning stages, with applications in neuroscience and OOD detection. Robust experiments substantiate these findings, providing a solid foundation for their conclusions.

**Weaknesses:**

1. The structure needs refinement, as it currently feels like the authors have packed too much content into the main paper, resulting in diminished clarity. The main paper covers a wide range of topics—from manifold capacity to manifold geometry, from theory to experiments—and the relationship between manifold capacity and various manifold geometry measures is weakly explained, relying primarily on Figure 2c with minimal analysis. I suggest that the authors avoid treating manifold geometry as a separate section, even though some interesting findings are presented. An alternative approach would be to frame geometry as an extension of capacity (or to present capacity itself as a facet of manifold geometry) and to provide necessary analysis (I notice there is analysis in the appendix. It's better to consolidate relevant analysis from the appendix into a concise summary in the main paper).
2. The algorithm is not clearly presented, which also appears to be a side effect of the structural issue noted above. Given that the paper aims to provide practical insights into feature learning, with applications in neuroscience and machine learning, it is important to ensure readability for readers unfamiliar with the manifold background. Thus, a clear algorithm outlining the process for computing capacity during training would be essential.

**Questions:**

1. Could the authors provide an example of the lazy regime? Specifically, how can a neural network be trained without actually modifying its internal features?

---

> ### Author Response · Authors · 2024-11-19
> **Response Round 1**
>
> We thank **Reviewer D4ag** for your time and effort for providing detailed suggestions and questions to improve our manuscript.
>
> To recap, in this paper, we propose using manifold capacity and its geometric measures to investigate the lazy vs. rich dichotomy in feature learning. We appreciate that the reviewers found our approach using manifold capacity to measure feature learning to be "novel application" (**reviewer nCob, cJiT**), have "better alignment with the degree of feature learning compared to other metrics" (**reviewer nCob**), with "robust experiments substantiate these findings" (**reviewer D4ag**). **Reviewer v9qp, and D4ag** also noted that our findings in learning stages and OOD generalization being insightful.
>
> Before addressing each of your points in detail , we would like to draw your attention to our general responses, which provide context and address several common concerns raised across the reviews.
>
> * **Q1: The structure needs refinement, as it currently feels like the authors have packed too much content into the main paper, resulting in diminished clarity.**
>     * **A1:** We thank the reviewer for pointing out the structural issue of our manuscript. We completely agree with the suggestion and we’ve spent significant amount of effort in reorganizing the structure of the paper. In particular, we’ve increased the coverage on the definition and background information in Section 2 (on the introduction of manifold capacity and geometry) and Section 3.1 (the theoretical results). We invite the reviewer to skim through the changes we made according to your suggestions in the revised version. We look forward to hearing Reviewer D4ag’s thoughts and further advice on the revised version.
> * **Q2: The algorithm is not clearly presented, which also appears to be a side effect of the structural issue noted above.**
>     * **A2:** We thank the reviewer for the very good suggestion. We’ve added the results of the mean-field approximation for manifold capacity in Equation 1 (line 187), which can be empirically computed. We also added the pseudocode of estimating manifold capacity and its effective geometric measures in page 22. We look forward to the reviewer’s thoughts on the revised version and welcome any further suggestions and comments.
> * **Q3: Could the authors provide an example of the lazy regime? Specifically, how can a neural network be trained without actually modifying its internal features?**
>     * **A3:** We provide an example of lazy regime presented in Fig. 2b where the blue curve in the capacity panel doesn’t change at all across training while the text accuracy increases. We remark that this is the same setting used in [Chizat et al., NeurIPS 2019] and they have found that both the network weights and NTK are not changing during training. Namely, the network mainly learns via adjusting its final readout weights (and the features are hence the so called random features).
>
> We thank the reviewer for the thoughtful suggestions, which have significantly improved the paper’s clarity and presentation. We also appreciate the reviewer’s time and reevaluation of our revised work. Thank you again for your thoughtful feedback and time!
>
> ## References:
> * [Chizat et al., NeurIPS 2019] Chizat, Lenaic, Edouard Oyallon, and Francis Bach. "On lazy training in differentiable programming." Advances in neural information processing systems 32 (2019).

---

> > ### Comment · Reviewer_D4ag · 2024-11-27
> > **Clarity has been improved**
> >
> > I appreciate the authors’ efforts in refining the manuscript, which has significantly improved its flow and clarity. However, I still encourage the authors to provide more intuitive explanations of the manifold metrics used in this paper, such as manifold capacity. While the presentation has improved, it remains somewhat challenging for readers without a relevant background to fully understand the paper’s contributions. Overall, I find this to be an interesting and valuable study on manifold measures and feature learning richness. I have raised the presentation score to 3, and I would raising my overall rating to 7 if it is possible.

---

### Official Review · Reviewer_K6x2 · 2024-11-03

**Soundness:** 3
**Presentation:** 2
**Contribution:** 3
**Rating:** 5
**Confidence:** 3

**Summary:**

This paper advocates the use of manifold capacity as a way of classifying as a measure of the regime of neural networks. Manifold capacity is a measure from neuroscience literature that quantifies the performance of a linear classifier over features for classification as a measure of the richness of classifiers. The authors advocate this measure over other measures such as "lazy regimes" during which features show limited change in learning. They use this to analyze 2-layer RELU networks

**Strengths:**

I think the use and introduction of manifold capacity as a measure is interesting for the ML community as a metric for representation learning. They relate manifold capacity to test accuracy in 2-layer networks and deeper networks empirically. They also relate to other measures such as weight changes and alignment, though this is mostly in the appendix.

**Weaknesses:**

The paper is haphazardly written. The difference between previous work and their problem statement is not well delineated. Their goals arent succinctly mentioned. The main theorem statement being nearly vacuous in the main paper. "In 2-layer neural networks trained with gradient descent in the rich regime the changes in capacity track the underlying degree of richness in feature learning." Since richness itself is defined as capacity, this is either circular or vacuous.

A deeper read of the result in the appendix shows that they approximate the result of gradient descent and use a gaussian model to approximate a 2-layer network. This is perhaps the most interesting result in the paper but is not at all covered in the main paper. The authors should state that they prove that using SGD, models with high capacity converge to high accuracy or something less vacuous and offer the proof.

Other features such as manifold radius and "alignment" are also not measured. The connection to "untangling" is also not mentioned. Weight changes are used mainly as a strawman to invoke connections to NTK and need not be done.

Figure 2 is very confusing with a nonsensical caption, "Higher capacity means that a higher number of
manifolds per neuron can be packed in the neural state space." What does it mean to "pack manifolds into neurons." This kind of shoddy language obfuscates the message to the reader.

As the current writing stands, this seems to contribute no more (and likely less by way of confusion) to the work in 2018 by Chung et al. which describes the full intuition of manifold capacity and outlines its use both in neuronal and neural networks.

**Questions:**

What does "untangling" specifically mean?

What are the implications of the 2-layer results on deeper neural networks?

---

> ### Author Response · Authors · 2024-11-19
> **Response Round 1 (1/3)**
>
> We thank **Reviewer K6x2** for your time and effort for providing detailed suggestions and questions to improve our manuscript.
>
> To recap, in this paper, we propose using manifold capacity and its geometric measures to investigate the lazy vs. rich dichotomy in feature learning. We appreciate that the reviewers found our approach using manifold capacity to measure feature learning to be "novel application" (**reviewer nCob, cJiT**), have "better alignment with the degree of feature learning compared to other metrics" (**reviewer nCob**), with "robust experiments substantiate these findings" (**reviewer D4ag**). **Reviewer v9qp, and D4ag** also noted that our findings in learning stages and OOD generalization being insightful.
>
> Before addressing each of your points in detail , we would like to draw your attention to our general responses, which provide context and address several common concerns raised across the reviews.
>
> * **Q1:** The difference between previous work and their problem statement is not well delineated. Their goals aren’t succinctly mentioned.
>     * **A1:** The goal of this work is to provide a representation-based method to quantify task-relevant changes in neural representations, offering a novel approach to investigating the lazy vs. rich feature learning question. To the best of our knowledge, no previous work has proposed a representational-based method that can quantify such task-relevant changes in neural representations. For instance, the most common method involves tracking weight changes in a network. However, a network might change its weights while degrading its representations. Another common method, which is to track the change in the norm of the neural tangent kernel (NTK), requires the knowledge of the entire model weights to compute the NTK.
>     * About the distinction of our work from Chung et al 2018, while this previous work propose the use of mean-field approximation to compute manifold capacity (cited at Equation 1, L187 in our revised main text), our work introduces theoretical work to characterize manifold capacity after 1-step gradient update in 2-layer neural network, which is to our knowledge, a novel theoretical result. We look forward to the reviewer’s thoughts on the revised version and welcome any further suggestions.
> * **Q2:** The main theorem statement is nearly vacuous in the main paper. "In 2-layer neural networks trained with gradient descent in the rich regime, the changes in capacity track the underlying degree of richness in feature learning." Since richness itself is defined as capacity, this is either circular or vacuous.
>     * **A2a:** We apologize for the confusion in the initial version! In the revised version, we provide a more detailed statement of the theorem in lines 301–317, clarifying how the setting relates to rich learning. Specifically, richness is defined by how well the neural representations align with a hidden signal direction (in a teacher-student setting) [Ba et al., NeurIPS 2022]. In this setting, the alignment to signal is proportional to the learning rate, hence the learning rate can be considered the ground-truth for the degree of richness [Ba et al., NeurIPS 2022] (see lines 291–295).
>     * **A2b:** The main result of Theorem 1 is to establish the connection between capacity and learning rate, which is not circular and requires substantial technical development (lines 315–317). We welcome the reviewer’s further thoughts on the revised version.
>     * **A2c:** We interpret these results as a justification for using capacity to track the degree of richness in feature learning in a well-studied theoretical setting. We remark that our proof requires substantial technical improvements from previous work [Ba et al., NeurIPS 2022] due to the fundamental difference between regression and classification.

---

> > ### Author Response · Authors · 2024-11-19
> > **Response Round 1 (2/3)**
> >
> > * **Q3: A deeper read of the result in the appendix shows that they approximate the result of gradient descent and use a Gaussian model to approximate a 2-layer network. This is perhaps the most interesting result in the paper but is not at all covered in the main paper. The authors should state that they prove that using SGD, models with high capacity converge to high accuracy or something less vacuous and offer the proof.**
> >     * **A3a:** We agree with the reviewer that the theoretical result is the most technically involved contribution of this paper. At the same time, we highlight the novelty of our application of manifold capacity and its geometric measures as progress indicators for delineating feature learning (as noted by **Reviewers nCob, cJiT, v9qp, and D4ag**).
> >     * **A3b:** We have revised and clarified to add more details about the context and interpretation of the theoretical results in Section 3.1 and Appendix C. We welcome the reviewer’s feedback on these updates.
> > * **Q4: Other features such as manifold radius and "alignment" are not measured. The connection to "untangling" is also not mentioned.**
> >     * **A4:** We have added a detailed discussion of the connection between manifold capacity (quantifying the degree of “manifold untangling”) and geometric measures in lines 207–220, Fig. 2c, and Appendices B.4 and B.5. In Appendix B.5, we provide synthetic examples illustrating how changes in these geometric measures affect capacity.
> > * **Q5: Weight changes are used mainly as a strawman to invoke connections to NTK and need not be done.**
> >     * **A5:** Weight changes and NTK (or CKA) changes are the most prevalent methods in the literature for studying the rich vs. lazy problem in feature learning [Chizat et al., NeurIPS 2019]. In addition to weight changes, in Figure 3, we also have comparison to NTK-label alignment and representation-label alignment. As the main goal of this paper is to propose using manifold capacity to quantify richness, we believe it is reasonable to compare our method with these approaches. We welcome suggestions for alternative comparison methods.
> > * **Q6: Figure 2 is very confusing with a nonsensical caption, "Higher capacity means that a higher number of manifolds per neuron can be packed in the neural state space." What does it mean to "pack manifolds into neurons"? This kind of shoddy language obfuscates the message to the reader.**
> >     * We thank the reviewer to point out the concern about the vague language, however, we would like to note that "capacity" and "packability" are standard concepts in information theory, relating to the number of patterns a model can encode with a given number of bits. While we acknowledge that this concept may not be very common in the deep learning community, and we would try our best to explain the terminology, we want to emphasize that the terminology and language that we use is the results of a long line of established work in the information theory field, and should not be characterized as "shoddy" or "non-sensical".
> >     * The term "packing" is a standard notion in information theory and coding theory, commonly used to describe the efficiency of organizing or separating structures (e.g., signals or objects) in a given space. In our context, "packing manifolds" refers to the ability to represent and linearly separate a large number of manifolds within the neural representational space, quantified by the ratio P/N, where P is the number of manifolds and N is the dimensionality of the neural space. Higher capacity implies a greater number of manifolds can be "packed" into this space while maintaining separability.

---

> > ### Author Response · Authors · 2024-11-19
> > **Response Round 1 (3/3)**
> >
> > * **Q7: As the current writing stands, this seems to contribute no more (and likely less by way of confusion) to the work in 2018 by Chung et al., which describes the full intuition of manifold capacity and outlines its use in neuronal and neural networks.**
> >     * **A7:** We respectfully disagree with this assessment and apologize for any confusion caused. The main contribution of [Chung et al., PRX 2018] was the definition of manifold capacity and geometry, demonstrated only on mathematical examples. Neither real neural data nor deep network results were explored. While follow-up works applied manifold capacity in neuroscience and machine learning, to our knowledge, our work is the first to connect manifold capacity to feature learning. Furthermore, while this previous work propose the use of mean-field approximation to compute manifold capacity (cited at Equation 1, L187 in our revised main text), our work introduces theoretical work to characterize manifold capacity after 1-step gradient update in 2-layer neural network, which is to our knowledge, a novel theoretical result.
> > * **Q8: What does "untangling" specifically mean?**
> >     * **A8:** Untangling, a term from computational neuroscience, describes how object manifolds become progressively more separated along the visual hierarchy [Dicarlo and Cox, TiCS 2007]. In [Chung et al., PRX 2018], manifold capacity was introduced as a mathematical measure of untangling. Different definitions of untangling may apply depending on the task (e.g., classification in [Chung et al., PRX 2018]).
> > * **Q9: What are the implications of the 2-layer results on deeper neural networks?**
> >     * **A9:** These results justify using capacity to track the degree of richness in feature learning within a well-studied theoretical framework. We also remark the technical challenge of this result due to the fundamental differences between regression (as in [Ba et al., NeurIPS 2022]) and classification.
> >
> > In summary, we have clarified the manuscript’s goals and contributions, addressed the theoretical and presentation concerns, and expanded on key definitions and intuitions, particularly around manifold capacity, geometric measures, and their relevance to feature learning. These revisions strengthen the manuscript’s clarity and rigor, and we thank Reviewer K6x2 for their thoughtful feedback and suggestions, which have significantly improved the work. We also appreciate the reviewer’s time and reevaluation of our revised work.
> >
> > ## References
> > * [Ba et al., NeurIPS 2022] Ba, Jimmy, et al. "High-dimensional asymptotics of feature learning: How one gradient step improves the representation." Advances in Neural Information Processing Systems 35 (2022): 37932-37946.
> > * [Chizat et al., NeurIPS 2019] Chizat, Lenaic, Edouard Oyallon, and Francis Bach. "On lazy training in differentiable programming." Advances in neural information processing systems 32 (2019).
> > * [Chung et al., PRX 2018] Chung, SueYeon, Daniel D. Lee, and Haim Sompolinsky. "Classification and geometry of general perceptual manifolds." Physical Review X 8.3 (2018): 031003.
> > * [Dicarlo and Cox, TiCS 2007] DiCarlo, James J., and David D. Cox. "Untangling invariant object recognition." Trends in cognitive sciences 11.8 (2007): 333-341.
> > * MacKay, David (2003-09-25). Information Theory, Inference and Learning Algorithms. Cambridge University Press. p. 483. ISBN 9780521642989.
> > * Cover, Thomas M. (June 1965). "Geometrical and Statistical Properties of Systems of Linear Inequalities with Applications in Pattern Recognition". IEEE Transactions on Electronic Computers. EC-14 (3): 326–334. doi:10.1109/PGEC.1965.264137. ISSN 0367-7508.

---

> > > ### Comment · Reviewer_K6x2 · 2024-11-28
> > > **Paper improved during revision ---could improve ore**
> > >
> > > I appreciate the authors response and particularly the revision of their paper which now includes many of the definitions which were missing and more detailed statements of the theorem. However, it now seems to me that the most obvious "strawman" would be mutual information between learned features and classification tasks as in this paper https://ieeexplore.ieee.org/document/10480166, since these features are "untangled" they could be mutually informative individual to the task at hand.  Since weight changes, as they say do not imply richness of features.
> > >
> > >  I am glad that the authors specifically defined packability in this context. While packing and sphere packing are terms in coding theory it was not apparent how this was used previously.
> > >
> > > However, I still find the writing somewhat confusing and the paper could relate the task-relevant "untangling" and the manifold capacity to performance without the distracting strawmen of NTK. I think there is enough by way of definition here and translation from computational neuroscience to clarify it on its own merit. Therefore I will increase some aspects of my score and not others.

---

### Official Review · Reviewer_v9qp · 2024-11-04

**Soundness:** 3
**Presentation:** 2
**Contribution:** 3
**Rating:** 5
**Confidence:** 2

**Summary:**

This paper proposed that manifold capacity is a better way to identify the process of feature learning than commenly used rich-lazy dichotomy. This claim is theoretically proved in a limited setting, and empirically verified.

**Strengths:**

- The idea of catching the richness by manifold capacity is potentially useful.
- The experiments verifies that manifold capacity captures different stages of learning.

**Weaknesses:**

- The presentation is very confusing. For example, in the experiments, there are several critical measures, such as effective radius, dimension, center alignment etc. but they are not defined (at least not in the main paper).
- The main paper claimed that the Theorem 1 is proved for a 2-layer NN. However, the NN used in the proof is actually different from what people would expect to be a "2-layer NN", since based on Assumption 1: 1) the second layer is not trained but set to random; and 2) the choice of activation function is very limited as there is a rather strong condition on the activation function.

**Questions:**

In Section 2.1, what does "i-th input category" mean? In eq. (1), $P$ is the variable of the $\max$ operator, and for any $i \in [P]$, $\mathcal M_i$ is defined, and $\mathcal M_i$ is defined by $\mathcal X_i$. This somehow implies that $\mathcal X_i$ is also a part of the variable of the $\max$ operator, instead of pre-defined. Is it true?

---

> ### Author Response · Authors · 2024-11-19
> **Response Round 1 (1/2)**
>
> We thank **Reviewer v9qp** for your time and effort for providing detailed suggestions and questions to improve our manuscript.
>
> To recap, in this paper, we propose using manifold capacity and its geometric measures to investigate the lazy vs. rich dichotomy in feature learning. We appreciate that the reviewers found our approach using manifold capacity to measure feature learning to be "novel application" (**reviewer nCob, cJiT**), have "better alignment with the degree of feature learning compared to other metrics" (**reviewer nCob**), with "robust experiments substantiate these findings" (**reviewer D4ag**). **Reviewer v9qp, and D4ag** also noted that our findings in learning stages and OOD generalization being insightful.
>
> Before addressing each of your points in detail , we would like to draw your attention to our general responses, which provide context and address several common concerns raised across the reviews.
>
> * **Q1: There are several critical measures, such as effective radius, dimension, and center alignment, but they are not defined (at least not in the main paper).**
>     * **A1a:** We thank the reviewer for pointing out the confusion in the previous version of the manuscript. We have taken these suggestions seriously and significantly improved clarity in the revised version. In particular, we now provide detailed definitions and intuitive explanations of the geometric measures in lines 205–223 of the main paper. Additionally, Fig. 2c has been updated to include an intuitive depiction of how these geometric measures influence manifold capacity. We also kindly invite the reviewer to look at Issue 1 in general response which provides a detailed explanation on the definition of these metrics.
>     * **A1b:** We have also improved the presentation of the definition of manifold capacity in Section 2.1, now including concrete mathematical equations and definitions for clarity. Furthermore, Appendix B on manifold capacity theory has been significantly restructured for better readability. We look forward to the reviewer’s thoughts on the revised version and welcome any further suggestions or comments.
> * **Q2: The main paper claimed that Theorem 1 is proved for a 2-layer NN. However, the NN used in the proof is different from what people would expect to be a "2-layer NN," since, based on Assumption 1: (1) the second layer is not trained but set to random, and (2) the choice of activation function is very limited, as there is a strong condition on the activation function.**
>     * **A2a:** We now highlight this limitation in the main text and discuss its connection to the literature. Specifically, we explain the motivation to only update the first-layer weight W and fix the readout layer weight a in the main text (lines 287-289). Particularly, following the convention from prior work [Ba et al., NeurIPS 2022], the second-layer (read-out weight) is fixed to (1) avoid lazy learning, in which the network minimally adjust the hidden layer representation and focus on learning the readout weight, and (2) enable mathematically analysis focusing on the hidden layer representations.
>     * **A2b:** Regarding the choice of activation, we are using standard assumptions from the theory literature. The following is a quote from [Ba et al., NeurIPS 2022]: “Following [Hu and Lu, ToIT 2022], we assume smooth centered activation to simplify the computation; empirical evidence suggests that similar result holds beyond this condition (e.g. [Loureiro et al., NeurIPS 2021]). We also expect the Gaussian input assumption may be replaced by weaker orthogonality conditions as in [Fan and Wang, NeurIPS 2020].”
>     * **A2c:** We interpret these results as a justification for using capacity to track the degree of richness in feature learning in a well-studied theoretical setting. We remark that our proof requires substantial technical improvements from previous work [Ba et al., NeurIPS 2022] due to the fundamental difference between regression and classification.

---

> > ### Author Response · Authors · 2024-11-19
> > **Response Round 1 (2/2)**
> >
> > * **Q3: In Section 2.1, what does "i-th input category" mean?**
> >     * **A3:** For example, in CIFAR-10, there are 10 categories (i.e., classes) such as cats, dogs, etc. The "i-th input category" refers to the i-th class according to some canonical ordering.
> > * **Q4: In eq. (1), P is the variable of the max operator, and for any i∈[P], Mi is defined, and Mi is defined by Xi. This somehow implies that Xi is also a part of the variable of the max operator, instead of being predefined. Is this true?**
> >     * A4: We apologize for the confusion between Mi and Xi. In the manifold capacity definition, there should only be Mi, while Xi was simply an example of Mi (e.g., as point clouds). We have rewritten Section 2.1, and we hope the presentation is now clearer. We look forward to the reviewer’s feedback and further suggestions on improving clarity.
> >
> > We sincerely thank Reviewer v9qp for the thoughtful feedback, which has greatly enhanced the paper’s readability and precision. We also appreciate the reviewer’s time and reevaluation of our revised work.
> >
> > ## References
> > * [Ba et al., NeurIPS 2022] Ba, Jimmy, et al. "High-dimensional asymptotics of feature learning: How one gradient step improves the representation." Advances in Neural Information Processing Systems 35 (2022): 37932-37946.
> > * [Hu and Lu, ToIT 2022] Hu, Hong, and Yue M. Lu. "Universality laws for high-dimensional learning with random features." IEEE Transactions on Information Theory 69.3 (2022): 1932-1964.
> > * [Loureiro et al., NeurIPS 2021] Loureiro, Bruno, et al. "Learning curves of generic features maps for realistic datasets with a teacher-student model." Advances in Neural Information Processing Systems 34 (2021): 18137-18151.
> > * [Fan and Wang, NeurIPS 2020] Fan, Zhou, and Zhichao Wang. "Spectra of the conjugate kernel and neural tangent kernel for linear-width neural networks." Advances in neural information processing systems 33 (2020): 7710-7721.

---

### Official Review · Reviewer_cJiT · 2024-11-04

**Soundness:** 2
**Presentation:** 2
**Contribution:** 2
**Rating:** 6
**Confidence:** 3

**Summary:**

In this paper, the authors focus on the feature learning in deep learning. They use manifold capacity as a metric to quantify the degree of richness of feature learning. Experiment results show that such capacity can reveal different learning stages in different settings. They also apply this to problems in neuroscience and out-of-distribution tasks.

**Strengths:**

-	Understanding feature learning in deep learning is an important and interesting research problem.
-	The paper proposes a metric called manifold capacity to measure the feature learning progress, which seems to be new.
-	Several experiments in different domains are presented to support the claim.

**Weaknesses:**

See questions section below.

**Questions:**

-	There seems to be no explanation for Figure 2c. I don’t know what does those operations mean.
-	I’m a bit confused about eq (1), the definition of model capacity. Are the manifold $\{\mathcal{M}_i\}$ predefined or they are changing when $N$ is increasing. If they are changing, how do you choose them? Also, what values should $y_i$ take?
-	When looking at Appendix C for the way to compute manifold capacity, several notions seem to be not defined, such as $T$, $\lambda$.

-	I believe the way of computing manifold capacity should be mentioned in the main text, or at least mentioned under what conditions are those values computed (I believe they cannot be exactly computed unless some conditions are assumed).

-	For Theorem 1, when looking at the actual statement in appendix, these results are proved only under the setting that with one gradient step update. Though it is understandable that going beyond this is technically difficult, I feel the statement in the main text gives the reader a wrong impression.

-	In section 3.2/figure 3b, it’s not clear to me what the definition of wealthy and poor regime are and how they are related to the input dimension. Also, I’m not sure why at initialization there will be task-relevant features (and I don’t know what are task-relevant features in this setting). Moreover, the purple line in capacity in figure 3b changes only a little bit throughout the training, I’m not sure how to interpret this (feature learning only changes a little?).

-	It’s a bit hard for me to follow section 4, presumably because the definition of these metric (e.g., dimension, radius,…) are missing.


Typo:

-	Line 173, $o_N(1) \to 1$ -> $o_N(1) \to 0$

---

> ### Author Response · Authors · 2024-11-19
> **Response Round 1 (1/2)**
>
> We thank **Reviewer cJiT** for your time and effort for providing detailed suggestions and questions to improve our manuscript.
>
> To recap, in this paper, we propose using manifold capacity and its geometric measures to investigate the lazy vs. rich dichotomy in feature learning. We appreciate that the reviewers found our approach using manifold capacity to measure feature learning to be "novel application" (**reviewer nCob, cJiT**), have "better alignment with the degree of feature learning compared to other metrics" (**reviewer nCob**), with "robust experiments substantiate these findings" (**reviewer D4ag**). **Reviewer v9qp, and D4ag** also noted that our findings in learning stages and OOD generalization being insightful.
>
> Before addressing each of your points in detail , we would like to draw your attention to our general responses, which provide context and address several common concerns raised across the reviews.
>
> * **Q1: There seems to be no explanation for Figure 2c. I don’t know what those operations mean.**
>     * **A1a:** In Figure 2c, we presented how the changes of geometric measures would affect the capacity value as shown in [Chou et al., 2024]. For example, smaller manifold radius and smaller manifold dimension would make the capacity value higher as the manifolds become more separable. See the revised version for pictorial intuition and more examples.
>     * **A1b:** In the revised version (lines 225–237), we have added more explanations for Figure 2. We have also included the formula and interpretation for all of the geometric measurements in Section 2.2. In Appendix B.4 and B.5, we provide examples illustrating how these geometric measures interact with capacity as shown in Figure 2c. We look forward to the reviewer’s thoughts on the revised version and welcome any further suggestions and comments.
> * **Q2: I’m a bit confused about eq (1), the definition of model capacity. Are the manifolds Mi predefined, or do they change as N increases? If they change, how do you choose them? Also, what values should it take?**
>     * **A2:** We apologize for the confusion in the previous version. In the revised version first introduces (1) simulated capacity in line 168-177 first, which is more intuitive and can be empirically computed, but cannot be analytically analyzed. Next, we present (2) the capacity formula of the mean-field approximation (line 187-190), which is analytically trackable, more efficient for empirical computation, and link to manifold geometric measures (as delineated later). The definition of mean-field approximation (which is the one we put in the previous version) is now deferred to the Appendix B.3 (line 979-1020). We remark that these are results from [Chung et al., PRX 2018] and [Chou et al., 2024]. We also kindly invite the reviewer to look at Issue 1 in general response which provides a detailed explanation on the definition of manifold capacity.
> * **Q3: In Appendix C, several notions for computing manifold capacity, such as T and λ, are not defined.**
>     * **A3:** Thank you for pointing out this oversight. We have addressed the issue in Appendix B (the previous Appendix C has been reorganized) and double-check that all our notations are consistent.
> * **Q4: The way of computing manifold capacity should be mentioned in the main text, or at least under what conditions these values are computed. (I believe they cannot be exactly computed unless certain conditions are assumed.)**
>     * **A4a:** We appreciate the reviewer’s insightful point. Indeed, manifold capacity cannot be computed exactly; instead, we use a mean-field approximation formula.
>     * **A4b:** We have carefully addressed this concern by adding more explanations about the connection between manifold capacity and its mean-field approximation in lines 180–200. Specifically, we've included in the main text Equation (1) at line 187 the results of the mean-field approximation, which can be empirically computed. For completeness, we also provide pseudocode for estimating manifold capacity via the mean-field formula on page 21. We look forward to the reviewer’s thoughts on the revised version and welcome further suggestions or comments.

---

> > ### Author Response · Authors · 2024-11-19
> > **Response Round 1 (2/2)**
> >
> > * **Q5: For Theorem 1, the results are proved only under the assumption of one gradient step update. Although going beyond this is technically difficult, I feel the statement in the main text gives the reader a misleading impression.**
> >     * **A5a:** We now highlight this limitation in the main text and discuss its connection to the literature. Specifically, we provide more details in the updated statement of Theorem 1 (lines 301–311) and footnote 6 regarding the limitation of one-step gradient descent.
> >     * **A5b:** We follow the convention of prior work [Ba et al., NeurIPS 2022], which studied the prediction risk of ridge regression in the same 2-layer NN setting in the feature learning regime. This work focused only on presenting results for a one-step gradient update. Similarly, for simplicity, we present the result of one-step updates. Extending beyond one gradient step could break the key Gaussian equivalence step (as noted in footnote 2 of [Ba et al., NeurIPS 2022]) and poses significant mathematical challenges. We addressed this limitation in footnote 6 on page 6 of the revised version.
> >     * **A5c:** We interpret these results as a justification for using capacity to track the degree of richness in feature learning in a well-studied theoretical setting. We remark that our proof requires substantial technical improvements from previous work [Ba et al., NeurIPS 2022] due to the fundamental difference between regression and classification.
> > * **Q6: In Section 3.2/Figure 3b, it’s unclear what the definitions of "wealthy" and "poor" regimes are and how they relate to input dimension. Additionally, why are there task-relevant features at initialization? Furthermore, the purple line in Figure 3b changes very little during training. How should this be interpreted?**
> >     * **A6a:** This is an excellent question! Task-relevant features refer to structures in the neural representations that help discriminate task conditions (e.g., classifying different categories).
> >     * **A6b:** Task-relevant features can exist at initialization because the data distribution inherently contains separable structures. For example, in the extreme case where each category corresponds to a single point (i.e., no in-class variation, with manifold radius and dimension equal to zero), the direction between two points serves as a feature that can separate the categories. This structure can be preserved in random initialization, forming the so-called random features.
> >     * **A6c:** Regarding the little change in capacity for wealthy initialization (i.e., the purple line in Fig. 3b), this occurs because random features at initialization already provide strong representations for separating manifolds. Consequently, training has limited impact on further improving capacity.
> > * **Q7: It’s difficult to follow Section 4, presumably because the definitions of metrics (e.g., dimension, radius) are missing.**
> >     * **A7a:** We apologize for previously omitting the concrete definitions of geometric measures in the main text. In the revised version, we provide detailed definitions in lines 205–220 and update Figure 2c to include intuitive explanations of these measures. We also kindly invite the reviewer to look at Issue 1 in general response which provides a detailed explanation on the definition of these metrics. We look forward to the reviewer’s feedback on the revised version and welcome further suggestions.
> >
> > In summary, we have addressed Reviewer cJiT’s valuable feedback by clarifying key definitions, improving the presentation of manifold capacity and geometric measures, and expanding on theoretical results and their limitations. Additionally, we provided detailed explanations for task-relevant features, initialization regimes, and capacity dynamics. We sincerely thank Reviewer cJiT for the thoughtful feedback, which has greatly enhanced the paper’s readability and precision. We also appreciate the reviewer’s time and reevaluation of our revised work.
> >
> > ## References
> > * [Chung et al., PRX 2018] Chung, SueYeon, Daniel D. Lee, and Haim Sompolinsky. "Classification and geometry of general perceptual manifolds." Physical Review X 8.3 (2018): 031003.
> > * [Ba et al., NeurIPS 2022] Ba, Jimmy, et al. "High-dimensional asymptotics of feature learning: How one gradient step improves the representation." Advances in Neural Information Processing Systems 35 (2022): 37932-37946.
> > * [Chou et al., 2024] Chou, Chi-Ning, et al. “Neural Manifold Capacity Captures Representation Geometry, Correlations, and Task-Efficiency Across Species and Behaviors” bioRxiv 2024.02.26.582157; doi: https://doi.org/10.1101/2024.02.26.582157

---

> ### Comment · Reviewer_cJiT · 2024-12-02
>
> I appreciate authors' response to address my concerns. The revised version improves the clarity. I therefore increase my score.

---

### Official Review · Reviewer_nCob · 2024-11-09

**Soundness:** 3
**Presentation:** 3
**Contribution:** 3
**Rating:** 6
**Confidence:** 3

**Summary:**

This paper presents a geometric framework to examine the manifold of neural network representations, providing insights into the distinctions between lazy and rich training regimes in feature learning. Specifically, it revisits the manifold capacity concept introduced in Chung et al. (2018), theoretically demonstrating that manifold capacity can serve as an indicator of the underlying richness in feature learning. Based on empirical studies using synthetic data and two-layer neural networks, observations are made regarding the relationship between manifold capacity and the degree of feature learning, as well as the stages of feature evolution. The proposed geometric measures are further applied to neural networks in neuroscience and out-of-distribution generalization tasks to explore the broader implications of this approach.

**Strengths:**

- The novel application of manifold capacity and other effective geometric measures to investigate the lazy-vs-rich dichotomy is intriguing. It shows better alignment with the degree of feature learning compared to other metrics, such as weight changes or NTK-label alignments.
- Most of the derivations seem correct, though I could not verify every detail.

**Weaknesses:**

- The theoretical derivation relies on a one-step gradient argument, but the fact is not mentioned in the manuscript. Moreover, the link between Theorem 2’s results and the increase in feature learning degree is not entirely clear, and additional commentary could enhance clarity.
- It would be helpful to discuss the generalizability of the observations, such as those in Figure 4. Various hyperparameters (e.g., the choice of optimization algorithms, weight initialization methods, batch size, learning rate, and scheduling) could influence implicit biases in the algorithm, affecting neural representations, geometric metrics, and even the stages of learning.

**Questions:**

- Regarding the proposed effective geometric measures to explain capacity changes, are there standard reference lengths compared to the radius? A simple manifold radius magnitude may not accurately capture the problem's complexity when the differences between manifold means scale identically.
- Additionally, the definitions and implications of axes alignment, center alignment, and center-axes alignment are less discussed compared to radius and dimensionality (e.g., in Figures 5b, 6c).
- When using geometric measures, which layer(s) should be analyzed? Are the results consistent across different layers?

---

> ### Author Response · Authors · 2024-11-19
> **Response Round 1 (1/2)**
>
> We thank **Reviewer nCob** for your time and effort for providing detailed suggestions and questions to improve our manuscript.
>
> To recap, in this paper, we propose using manifold capacity and its geometric measures to investigate the lazy vs. rich dichotomy in feature learning. We appreciate that the reviewers found our approach using manifold capacity to measure feature learning to be "novel application" (**reviewer nCob, cJiT**), have "better alignment with the degree of feature learning compared to other metrics" (**reviewer nCob**), with "robust experiments substantiate these findings" (**reviewer D4ag**). **Reviewer v9qp, and D4ag** also noted that our findings in learning stages and OOD generalization being insightful.
>
> Before addressing each of your points in detail , we would like to draw your attention to our general responses, which provide context and address several common concerns raised across the reviews.
>
> * **Q1: The theoretical derivation relies on a one-step gradient argument, but this fact is not mentioned in the manuscript.**
>     * **A1a:** We follow the convention of prior work [Ba et al., NeurIPS 2022], which studied the prediction risk of ridge regression in the same 2-layer NN setting in the feature learning regime. Specifically, they focused only on presenting results for a 1-step gradient update. For simplicity, we also presented results for a 1-step update. Extending beyond one gradient step could break the key Gaussian equivalence step (as noted in footnote 2 of [Ba et al., NeurIPS 2022]) and presents significant mathematical challenges in this setting.
>     * **A1b:** We addressed this limitation in footnote 6 on page 6 of the revised version.
>     * **A1c:** We interpret these results as a justification for using capacity to track the degree of richness in feature learning in a well-studied theoretical setting. We remark that our proof requires substantial technical improvements from previous work [Ba et al., NeurIPS 2022] due to the fundamental difference between regression and classification.
> * **Q2: The link between Theorem 2’s results and the increase in the degree of feature learning is not entirely clear, and additional commentary could enhance clarity.**
>     * **A2a:** First, we clarify that we are working with a teacher-student setting where the data labels are correlated with a hidden signal direction (see Setting 1, lines 1164–1168).
>     * **A2b:** In this context, the first-step gradient update can be approximated by a rank-1 matrix containing label information, aligning the updated weights with the hidden signal (Proposition 1, lines 1236–1241; see also Proposition 2 in [Ba et al., NeurIPS 2022]). Thus, in this setting, the learning rate serves as the ground truth to measure the amount of task-relevant information (i.e., richness in learning) within the model representation after gradient updates. We have also added this explanation to the main text (section 3.1, L290-295).
> * **Q3: It would be helpful to discuss the generalizability of the observations, such as those in Figure 4. Various hyperparameters (e.g., the choice of optimization algorithms, weight initialization methods, batch size, learning rate, and scheduling) could influence implicit biases in the algorithm, affecting neural representations, geometric metrics, and even the stages of learning.**
>     * **A3a:** This is an excellent question. In general, we aim for geometric measures that strike a balance between (i) detecting differences in manifold organization and (ii) being robust to minor changes that do not significantly affect performance.
>     * **A3b:** For 2-layer NNs, we found the results to be robust across different hyperparameters (see Appendix D for settings varying network parameters, data distributions, and optimization algorithms). We’ve added remarks about this robustness in the main text.
>     * **A3c:** For DNNs, the results were generally robust within the same architecture but could vary across architectures. For instance, in the OOD experiments, we observed geometric differences in VGG-11 (Fig. 6) and ResNet-18 (Fig. 22).
> * **Q4: Regarding the proposed effective geometric measures to explain capacity changes, are there standard reference lengths compared to the radius? A simple manifold radius magnitude may not accurately capture the problem's complexity when the differences between manifold means scale identically.**
>     * **A4:** This is a very insightful question! The radius is normalized by the norm of the manifold center, allowing it to accurately (and monotonically) capture the packability of manifolds. We've added the formula for all the geometric measures, including the radius, in the main text (line 207-line 220).

---

> > ### Author Response · Authors · 2024-11-19
> > **Response Round 1 (2/2)**
> >
> > * **Q5: The definitions and implications of axes alignment, center alignment, and center-axes alignment are less discussed compared to radius and dimensionality (e.g., in Figures 5b, 6c).**
> >     * **A5:** We appreciate the reviewer’s interest in the alignment measures. Due to space constraints, we did not emphasize them in the main text. However, in the OOD experiment, we observed that center-axis alignment, along with radius, serves as a geometric signature for the ultra-rich regime (see Figure 6c and lines 527–528).
> > * **Q6: When using geometric measures, which layer(s) should be analyzed? Are the results consistent across different layers?**
> >     * **A6:** In this paper, we focused our analysis on the last layer, which is traditionally considered to contain the most high-level features. Results from earlier layers are expected to exhibit weaker signals, as fewer categorical features are represented there.
> >
> > We appreciate the reviewer’s thoughtful suggestions and re-evaluation of our revised work. Thank you again for your thoughtful feedback and time!
> >
> >
> > ## References
> > * [Chung et al., PRX 2018] Chung, SueYeon, Daniel D. Lee, and Haim Sompolinsky. "Classification and geometry of general perceptual manifolds." Physical Review X 8.3 (2018): 031003.
> > * [Ba et al., NeurIPS 2022] Ba, Jimmy, et al. "High-dimensional asymptotics of feature learning: How one gradient step improves the representation." Advances in Neural Information Processing Systems 35 (2022): 37932-37946.
> > * [Chou et al., 2024] Chou, Chi-Ning, et al. “Neural Manifold Capacity Captures Representation Geometry, Correlations, and Task-Efficiency Across Species and Behaviors” bioRxiv 2024.02.26.582157; doi: https://doi.org/10.1101/2024.02.26.582157

---

> > > ### Comment · Reviewer_nCob · 2024-12-02
> > >
> > > I appreciate the authors' efforts to enhance the clarity of the paper through their elaborations.
> > >
> > > However, I still find the theoretical contribution challenging to interpret, particularly the connection between the capacities defined in Sections 2.1–2.2 of the revised manuscript and those discussed in Theorems 1 and 2. It would improve the manuscript significantly if these connections were made more explicit.
> > >
> > > Based on the authors' overall responses, I have slightly increased my score.
> > >
> > > A minor comment: In Lines 170–171, the variable $\mathbf{s}$ is not properly defined in the text.

---

### Author Response · Authors · 2024-11-19
**General Responses Round 1 (1/4)**

We thank all the reviewers for their time and effort. Your valuable feedback has greatly helped us improve the clarity and quality of our paper.

In this paper, we propose using manifold capacity and its geometric measures to investigate the lazy vs. rich dichotomy in feature learning. We appreciate that the reviewers found our approach using manifold capacity to measure feature learning to be "novel application" (**reviewer nCob, cJiT**), have "better alignment with the degree of feature learning compared to other metrics" (**reviewer nCob**), with "robust experiments substantiate these findings" (**reviewer D4ag**). **Reviewer v9qp, and D4ag** also noted that our findings in learning stages and OOD generalization being insightful.

The main concerns and questions to our paper surrounded the clarity of presentation, specifically due to the lack of (a) definition of manifold capacity and manifold geometric measurements, (b) algorithm to empirically compute manifold capacity, and (c) interpretation of the theoretical results . We have taken all the reviewers’ suggestions seriously and incorporated them into the revised manuscript.

Specifically, in the revision for the main text, we have incorporated (a) formal definition for manifold capacity and all geometric measurements, (b) formula and pseudocode to empirically compute manifold capacity, (c) more detailed context, interpretation and a more precise version of our theoretical results. Below in the thread we provide a more detailed response for each of the above general concern. Detailed responses to individual reviews will be replied separately in the threads.

The following is a summary of the changes we made in our updated manuscript to address these issues:
* Definition of manifold capacity:
    * Simulated version (more intuitive but not analytically tractable): Definition 1 in line 168-177.
    * Mean-field version (more complicated but analytically tractable): Definition 2 and 3 in line 979-1020.
    * Capacity formula: equation (1) in line 186-190.
* Definition of manifold geometric measures:
    * Intuition: Section 2.2, line 198-206.
    * Definitions: Section 2.2, line 207-220 . Appendix B.4.
    * How effective geometric measures connect to the changes in capacity value: Fig 2c and Appendix B.5.
* Overview of our theoretical results and interpretations: Section 3.1, line 277-314.
* An updated informal theorem statement: Theorem 1, line 299-309.

We believe these revisions significantly enhance the paper's accessibility and contributions, and we appreciate the reviewers’ re-evaluation of our work. Thanks again for the thoughtful feedback and time!

## References
* [Chung et al., PRX 2018] Chung, SueYeon, Daniel D. Lee, and Haim Sompolinsky. "Classification and geometry of general perceptual manifolds." Physical Review X 8.3 (2018): 031003.
* [Ba et al., NeurIPS 2022] Ba, Jimmy, et al. "High-dimensional asymptotics of feature learning: How one gradient step improves the representation." Advances in Neural Information Processing Systems 35 (2022): 37932-37946.
* [Chou et al., 2024] Chou, Chi-Ning, et al. “Neural Manifold Capacity Captures Representation Geometry, Correlations, and Task-Efficiency Across Species and Behaviors” bioRxiv 2024.02.26.582157; doi: https://doi.org/10.1101/2024.02.26.582157
* [Montanari et al., 2019] Montanari, Andrea, et al. "The generalization error of max-margin linear classifiers: High-dimensional asymptotics in the overparametrized regime." arXiv preprint arXiv:1911.01544 7 (2019).

---

> ### Author Response · Authors · 2024-11-19
> **General Responses Round 1 (2/4)**
>
> **Issue 1: Reviewers asked for formal introduction to manifold capacity, the algorithm to compute manifold capacity, and its connection to effective geometric measures.**
>
> We briefly recall that manifold capacity is a recently proposed metric [Chung et al., PRX 2018; Chou et al., 2024] for measuring how separable neural manifolds are in an average-case setting (i.e., the degree of manifold untangling). It has been widely applied in the neuroscience community to quantify the progression of neural computation in the brain. The main contribution of our work is to use manifold capacity as a representation-based method to quantify the degree of richness in feature learning. To the best of our knowledge, no previous work has proposed a representational-based method that can quantify task-relevant changes in neural representations. For instance, the most common method involves tracking weight changes in a network. However, a network might change its weights while degrading its representations. Another common method, which is to track the change in the norm of the neural tangent kernel (NTK), requires the knowledge of the entire model weights to compute the NTK.
>
> The following is a list of questions regarding the clarity of manifold capacity and relevant geometric measures raised by the reviewers and our responses:
>
> **Q1-1:** **Reviewer cJiT, v9qp, K6x2** asked for a clearer definition of manifold capacity. In the initial version, we included the mean-field definition of manifold capacity in the proportional limit but did not sufficiently explain how it is empirically computed. Also, reviewers cJiT, D4ag raised concerns about the practicality of calculating this value.
>
> **A1-1:** To address these issues, the revised version first introduces **simulated capacity** in line 168-177 first, which is more intuitive and can be empirically computed, but cannot be analytically analyzed. Next, we present the **capacity formula** of the mean-field approximation (line 187-190), which is analytically trackable, more efficient for empirical computation, and link to manifold geometric measures (as delineated later). The definition of mean-field approximation (which is the one we put in the previous version) is now deferred to the Appendix B.3 (line 979-1020).  We remark that these are results from [Chung et al., PRX 2018]  and [Chou et al., 2024].
>
> - **(Simulated capacity)** The following is the definition of simulation capacity. See also Section 2.1 in the revised version for more explanation.
>
> Definition: (Simulated capacity.)
> Let $P,N\in\mathbb{N}$ and $\mathcal{M}_i\subseteq\mathbb{R}^N$ be a convex set for each $i\in[P]=\{1,\dots,P\}$. For each $n\in[N]$, define
>
> $$
> p_n := \Pr_{\mathbf{y},\Pi_n}[\exists \theta\in\mathbb{R}^n\\, :\\, y_i\langle\theta,\mathbf{s}\_i\rangle\geq0,\\, \forall i\in[P],\\, \mathbf{s}\_i\in\mathcal{M}\_i)]
> $$
> where $\mathbf{y}$ is a random dichotomy sampled from $\\{\pm1\\}^P$ and $\Pi_n$ is a random projection operator from $\mathbb{R}^N$ to $\mathbb{R}^n$.
> Suppose $p_N=1$, the simulated capacity of $\\{\mathcal{M}_i\\}\_{i\in[P]}$ is defined as
>
> $$
> \alpha_{\text{sim}} := \frac{P}{\min_{n\\, :\\, p_n\geq0.5}\\{n\\}} \\, .
> $$
>
> - **(Capacity formula)** The following is the mean-field approximation formula to manifold capacity (note that this will be very important for the definition of geometric measures):
>
> $$
> \alpha^{-1}_{\text{mf}} = \frac{1}{P} \mathbb{E}\_{\mathbf{y},T} \left[\max\_{\mathbf{s}_i\in\mathcal{M}i} \left\\{ \\| \text{proj}\_{\text{cone}(\left\\{y_i,\mathbf{s}_i\right\\})} T \\|_2^2  \right\\} \right]
> $$
>
> where $\mathbf{y}$ is randomly sampled from $\\{\pm1\\}^P$ and $T$ is randomly sampled from $\mathcal{N}(0,I_N)$, the multivariate Gaussian distribution with mean $0$ and covariance $I_N$. $\text{cone}(\cdot)$ is the convex cone spanned by the vectors, i.e., $\text{cone}(\\{y_i\mathbf{s}_i\\})=\\{\sum_i\lambda_iy_i\mathbf{s}_i\\, :\\, \lambda_i\geq0\\}$.
>
> **Remark:** [Chou et al., 2024] showed that $|\alpha_{\text{sim}}-\alpha_{\text{mf}}|<O(1/N)$.

---

> ### Author Response · Authors · 2024-11-19
> **General Responses Round 1 (3/4)**
>
> **Q1-2:** **Reviewer nCob, cJiT, v9qp, D4ag** noted the lack of mathematical definitions for geometric measures in the main text, which makes it difficult to follow and interpret the geometric results.
>
> **A1-2:** In the revised version, after presenting the mean-field approximation, we introduce the concept of anchor points and define all relevant geometric measures in detail (line 205-220). Examples in the Appendix B.5 further demonstrate the relationship between capacity and these geometric measures, making the connections clear and intuitive. We remark that these are results from [Chung et al., PRX 2018] and [Chou et al., 2024].
>
> The following is a list of formal definition of effective geometric measures (please read them along with the capacity formula (2) above):
>
> * **Anchor points:** for each $\mathbf{y},T$ and $I\in[P]$, define $\{\mathbf{s}\_i(\mathbf{y},T)\} = y_i\cdot\arg\max\_{\{\mathbf{s}\_i\}}\\|\text{proj}_\{\text{cone}(\{y_i\mathbf{s}_i\})}T\\|_2^2$ as the *anchor points* with respect to $\mathbf{y}$ and $T$.
>     * Center and axis part of anchor points: For each $i\in[P]$, define $\mathbf{s}_i^0:=\mathbb{E}\_\{\mathbf{y},T}[\mathbf{s}_i(\mathbf{y},T)]$ as the *center* of the $i$-th manifold and define $\mathbf{s}_i^1(\mathbf{y},T):=\mathbf{s}_i(\mathbf{y},T)-\mathbf{s}_i^0$ to be the *axis* part of $\mathbf{s}_i(\mathbf{y},T)$ for each pair of $(\mathbf{y},T)$.
> * **Manifold geometric measures:**
>     * *Manifold dimension* captures the degree of freedom of the noises/variations within the manifolds. Formally, it is defined as $D_\text{mf}:=\mathbb{E}_\{\mathbf{y},T}[\\|\text{proj}\_\{\text{cone}(\{\mathbf{s}_i^1(\mathbf{y},T)\})}T\\|_2^2]$.
>     * *Manifold radius* captures the noise-to-signal ratio of the manifolds. Formally, it is defined as $R_\text{mf} := \sqrt{\mathbb{E}\_\{\mathbf{y},T}\left[\frac{\\|\text{proj}\_\{\text{cone}(\{\mathbf{s}\_i(\mathbf{y},T)\})}T\\|^2}{\\|\text{proj}\_\{\text{cone}(\{\mathbf{s}^1_i(\mathbf{y},T)\})}T\\|^2-\\|\text{proj}\_\{\text{cone}(\{\mathbf{s}\_\{i}(\mathbf{y},T)\})}T\\|^2}\right]}$.
>     * *Center alignment* captures the correlation between the center of different manifolds. Formally, it is defined as $\rho^c_\text{mf}:=\frac{1}{P(P-1)}\sum_{i\neq j}|\langle\mathbf{s}_i^0,\mathbf{s}_j^0\rangle|$.
>     * *Axis alignment* captures the correlation between the axis of different manifolds. Formally, it is defined as $\rho^a_\text{mf}:=\frac{1}{P(P-1)}\sum_{i\neq j}\mathbb{E}\_{\mathbf{y},T}[|\langle\mathbf{s}_i^1(\mathbf{y},T),\mathbf{s}_j^1(\mathbf{y},T)\rangle|]$.
>     * *Center-axis alignment* captures the correlation between the center and axis of different manifolds. Formally, it is defined as $\psi_\text{mf}:=\frac{1}{P(P-1)}\sum_{i\neq j}\mathbb{E}\_{\mathbf{y},T}[|\langle\mathbf{s}_i^0,\mathbf{s}_j^1(\mathbf{y},T)\rangle|]$.

---

> ### Author Response · Authors · 2024-11-19
> **General Responses Round 1 (4/4)**
>
> **Issue 2: Reviewers highlighted the confusion about the concrete setting of our theoretical results and how to interpret them, specifically how the results directly relate to the degree of feature learning.**
>
> We built on previous work from [Ba et al. NeurIPS 2022] and [Montanari et al., 2019] to analytically characterize the connection between capacity, prediction error, and the effective degree of richness in a well-studied theoretical model. We interpret our theoretical results as justifications for using capacity to track the degree of richness in feature learning. We remark that our proof requires substantial technical improvements from previous work [Ba et al., NeurIPS 2022] due to the fundamental difference between regression and classification. The following is a list of questions regarding our theoretical results raised by the reviewers and our responses:
>
> * **Reviewer nCob, cJiT** asked for a clear explanation of the assumptions and precise results in the theoretical framework, particularly the use of a 1-step gradient update.
>     * We agree that the precise scope of the theoretical results, particularly about capacity after one-step gradient update, is important to the result interpretation, and have added this assumption in the main text. We provide more context on this assumption below.
>     * Here we follow the convention of a previous work [Ba et al., NeurIPS 2022] which studied the prediction risk of ridge regression in the same 2-layer NN setting in the feature learning regime. In particular, they only focused on presenting the result of 1-step gradient update. So for the simplicity of presentation, we also just presented the result of 1-step update. Indeed, going beyond one gradient steps could break the key Gaussian equivalence step (as remarked in footnote 2 in [Ba et al., NeurIPS 2022]) and present a significant mathematical challenge in this setting.
>     * We addressed this issue in footnote 6 at the bottom of page 6 in the revised version.
> * **Reviewer v9qp** asked for why the readout weights have been fixed throughout training.
>     * The use of a 2-layer network with fixed readout weights is standard in related work (e.g., [Ba et al., NeurIPS 2022]), allowing for mathematical characterization and control over the interpolation between lazy and rich learning by adjusting the learning rate.
>     * Our results are technically non-trivial, as previous work focused on regression settings, while we address classification tasks, requiring different mathematical treatments.
> * **Reviewer nCob, cJiT, v9qp, K6x2** asked for the interpretations of our theoretical results, the relevance of the result to feature learning, as well as the position in the literature.
>     * We interpret these results as justifications for using capacity to track the degree of richness in feature learning in a well-studied theoretical setting.
>     * About the link between Theorem 1 result (which states capacity after 1-step gradient update is a monotonically increasing function of the learning rate) to the degree of feature learning, in our setting, first-step gradient update can be approximated by a rank-1 matrix that contains label information, resulting in the updated weight to be more aligned with the hidden signal β*. Hence, in this setting, the learning rate η can be used as the ground-truth to measure the amount of task-relevant information (i.e., richness in learning) in the model representation after gradient updates. We have also added this explanation in the main text for better clarification (see line 298-300 and footnote 6 at the bottom of page 6).
>     * We remark that our proof requires substantial technical improvements from previous work [Ba et al., NeurIPS 2022] due to the fundamental difference between regression and classification.

---

### Meta-Review · Area_Chair_qAKN · 2024-12-20

**Metareview:**

This paper introduces the concept of manifold capacity (as defined in Chung et al. 2018) for quantifying the untangling of task-relevant neural manifolds, providing a framework for understanding lazy vs rich feature learning. Theoretically, the paper establishes connections between manifold capacity and feature learning in two-layer neural networks trained under gradient descent, supported by empirical evidence on real settings such as VGG / ResNet on CIFAR-10. This connection revealed new geometric insights into subtypes of feature learning. Additionally, the paper demonstrates the framework’s utility in both neuroscience tasks and out-of-distribution generalization studies.

The reviewers generally acknowledged the novelty of employing manifold capacity for the study of feature learning. However, several reviewers highlighted a significant clarity issue concerning several key technical components. While the authors made significant efforts during the rebuttal phase to address these issues through extensive rewriting, the limited discussion period may not have been sufficient for the reviewers to fully reassess the manuscript’s technical quality. Therefore, I recommend reconsideration for future submissions after a careful rewriting to improve clarity of the presentation.

**Additional Comments On Reviewer Discussion:**

Several reviewers, including cJiT and v9qp, criticized the lack of clear definitions for critical measures (e.g., effective radius, alignment) in the original submission. While revisions addressed some issues, residual confusion remained, particularly in geometric interpretations and experimental setups.

The reliance on a one-step gradient update and the assumption of fixed readout weights were noted by cJiT and v9qp as limiting the generality of the theoretical results. The authors justified their choice as consistent with prior work and highlighted the mathematical challenges of extending beyond one step. However, the issue remains unresolved for some reviewers.

---

### Decision · Program_Chairs · 2025-01-22

Reject